# Spurious transcription causing innate immune responses is prevented by 5-hydroxymethylcytosine

Fan Wu[1], Xiang Li[1,9], Mario Looso[1,9], Hang Liu[1,9], Dong Ding[1], Stefan Günther[1], Carsten Kuenne[1], Shuya Liu[2,3], Norbert Weissmann[4,5], Thomas Boettger[1], Ann Atzberger[1], Saeed Kolahian[6], Harald Renz[6], Stefan Offermanns[2], Ulrich Gärtner[7], Michael Potente[8], Yonggang Zhou[1], Xuejun Yuan[1]✉ & Thomas Braun[1,5]✉

Generation of functional transcripts requires transcriptional initiation at regular start sites, avoiding production of aberrant and potentially hazardous aberrant RNAs. The mechanisms maintaining transcriptional fidelity and the impact of spurious transcripts on cellular physiology and organ function have not been fully elucidated. Here we show that TET3, which successively oxidizes 5-methylcytosine to 5-hydroxymethylcytosine (5hmC) and other derivatives, prevents aberrant intragenic entry of RNA polymerase II pSer5 into highly expressed genes of airway smooth muscle cells, assuring faithful transcriptional initiation at canonical start sites. Loss of TET3-dependent 5hmC production in SMCs results in accumulation of spurious transcripts, which stimulate the endosomal nucleic-acid-sensing TLR7/8 signaling pathway, thereby provoking massive inflammation and airway remodeling resembling human bronchial asthma. Furthermore, we found that 5hmC levels are substantially lower in human asthma airways compared with control samples. Suppression of spurious transcription might be important to prevent chronic inflammation in asthma.

DNA methylation on cytosine (5mC) is a crucial mechanism in the epigenetic modulation of cell-type specific transcription required for maintaining cell identity. Methylation of CpG islands located in gene promoters often results in transcriptional repression whereas, paradoxically, intragenic 5mC correlates frequently with transcriptional strength, although strong CpG methylation (at least 90%) slows down elongation[1,2].

A recent study demonstrated that the presence of intragenic DNA methylation correlates with prevention of spurious transcription within gene bodies, thereby ensuring transcriptional fidelity[3] and eliminating the threat of aberrant potentially hazardous aberrant RNAs[4-6].

Although 5mC-mediated gene repression is relatively stable, substantial phenotypical or functional changes such as cell differentiation

[1]Department of Cardiac Development and Remodeling, Max Planck Institute for Heart and Lung Research, Bad Nauheim, Germany. [2]Department of Pharmacology, Max Planck Institute for Heart and Lung Research, Bad Nauheim, Germany. [3]Department of Medicine, University Medical Center Hamburg-Eppendorf, Hamburg, Germany. [4]Cardiopulmonary Institute (CPI), Universities of Giessen and Marburg Lung Center (UGMLC), Giessen, Germany. [5]Member of the German Center for Lung Research (DZL), Justus-Liebig-University, Giessen, Germany. [6]Philipps University of Marburg - Medical Faculty, Center for Tumor- and Immunobiology (ZTI), Institute of Laboratory Medicine and Pathobiochemistry, Molecular Diagnostics, Marburg, Germany. [7]Institute for Anatomy und Cell Biology, Giessen, Germany. [8]Angiogenesis and Metabolism Laboratory, Max-Planck Institute for Heart and Lung Research, Bad Nauheim, Germany. [9]These authors contributed equally: Xiang Li, Mario Looso, Hang Liu. ✉e-mail: Xuejun.Yuan@mpi-bn.mpg.de; Thomas.Braun@mpi-bn.mpg.de

but also pathological processes require demethylation. The biochemistry of demethylation remained enigmatic for decades until the discovery of TET enzymes, which successively convert 5mC to 5hmC, 5-formylcytosine (5fC) and 5-carboxycytosine (5caC)[7]. Thymine DNA glycosylase (TDG)-mediated excision of 5fC and 5caC coupled with base excision repair (BER) will eventually result in demethylation[8]. Three family members exist in mammals, TET1–3, which share similar enzymatic activities and cofactor requirements (that is, α-ketoglutarate, oxygen and $Fe^{2+}$) but differ in expression profiles and target preferences[9]. For example, expression of mouse *Tet3* is low in embryonic stem cells (ESC) but increases substantially in some differentiated cell types[10]. Generation of 5hmC by TETs has been viewed mostly as a transition state, required for removal of 5mC and subsequent alleviation of gene repression, but the substantial amount of 5hmC in several somatic cell types makes it unlikely that 5hmC exclusively represents a nonfunctional intermediate of demethylation. Further support of this idea comes from the enrichment of 5hmC in gene bodies of highly expressed genes and at active enhancers[11]. Despite remarkable progress in the field, the true function of 5hmC formation at gene bodies is still incompletely understood, which is caused in part by missing knowledge about the physiological roles of putative 5hmC interactors[12].

Changes in DNA methylation contribute to profound and reversible phenotype changes from contractile to synthetic states of smooth muscle cells (SMCs) in response to external cues[13,14]. SMCs are found not only in the medial layer of muscularized vessels but also in the airways. Phenotype switching of SMCs in airways contributes to diseases of the lung such as asthma and chronic obstructive pulmonary disease (COPD)[15–17]. *Tet2* was reported to act as a master regulator of murine SMC plasticity, since its knockdown in vitro inhibits expression of key procontractile genes and its overexpression elicits SMC gene expression in fibroblasts[18]. In contrast, the function of *Tet3* in SMCs remains unclear.

Here we describe a pivotal role of TET3 in regulating the fidelity of gene transcription that is required for maintaining the identity of SMC and balancing immune responses in the lung. Our study reveals that spurious transcripts in *Tet3*-deficient mouse SMCs lead to activation of TLR7/8 signaling-dependent innate immune responses and massive lung inflammation, resembling human asthma, offering perspectives to treat various lung diseases.

## Results

### Loss of *Tet3* reduces 5hmc levels in SMC

To explore the role of 5hmC in the regulation of gene expression in a physiological context, we first searched for cells strongly expressing *Tet3*, assuming that high expression may be indicative of a decisive function of TET3. To this end, we introduced a *LacZ* reporter gene cassette into the endogenous *Tet3* gene and visualized *LacZ* expression by 5-bromo-4-chloro-3-indolyl-β-D-galactoside (X-gal) staining. Expression of *Tet3-LacZ*, which was present in virtually all cells of the developing mouse embryo at E9.5, was enriched in SMCs of adult animals (Fig. 1a and Extended Data Fig. 1a,b). Likewise, we observed increased expression of *Tet3* in contractile SMCs derived from mouse embryonic stem cell (mESC-SMCs), while expression of *Tet1* and *Tet2* was lower, suggesting a dominant function of TET3 in SMCs (Extended Data Fig. 1c–e). To address the function of TET3 in SMCs in vivo, we generated SMC-specific *Tet3* knockout mice (*Tet3[smKO]*) using an inducible *α-SMA[ERT2Cre]* strain[19] (Extended Data Fig. 1f–h). *Tet3[smKO]* mice were viable and fertile but failed to gain body weight 15 weeks after tamoxifen administration (Extended Data Fig. 1i). Phenotyping of *Tet3[smKO]* mice, 8 weeks after tamoxifen administration, showed clear morphological changes in the lung, indicated by a shift from a columnar to a cuboidal epithelium, whereas no obvious structural abnormalities were detected in other SMC-containing organs (Fig. 1b and Extended Data Fig. 1j,k). Thus, we decided to focus on the lung for further studies. Introduction of a *tdTomato* reporter allele into *Tet3[smKO]* mice (referred

to as *Tet3[smKO:T]*) allowed FACS-based isolation of lung SMCs, revealing a profound reduction of 5hmC levels in SMCs after loss of *Tet3* (Fig. 1c–f and Extended Data Fig. 1l–p), while global levels of 5fC and 5caC were unchanged. Since 5fC and 5caC are effectively removed by TDG and base excision repair, we assume that 5fC/5caC levels do not reflect dynamic changes in the oxidation of 5hmC in *Tet3* mutant SMC but rather represent stable remnants of previous oxidation events, probably acquired during SMC differentiation (Fig. 1e,f). Furthermore, no obvious changes in the global 5mC content in *Tet3* mutant SMCs were observed (Fig. 1e,f). Although we cannot exclude that hypermethylation of active genomic regions is levelled out by a paradoxical loss of DNA methylation in heterochromatin[20], this finding might indicate that TET3 serves an additional function in SMCs, independent of dynamic 5mC changes required for transcriptional activation. Loss of *Tet3* did not lead to compensatory upregulation of TET2 and germline inactivation of *Tet2* using two different mouse strains did not change 5hmC levels in SMCs of the aorta and lung (Extended Data Fig. 2a–h,j). Furthermore, we did not detect any obvious morphological abnormalities in SMC-containing organs of *Tet2* mutants, although a critical role of *Tet2* was reported for maintaining the differentiated state of SMC in human coronary arteries[18] (Extended Data Fig. 2g–k). Notably, *Tet2* inactivation in *Tet3*-deficient SMCs (*Tet2/Tet3[smKO]*) did not lead to a further decline of 5hmC in bronchial SMCs (BSMCs) or aggravated the airway remodeling phenotype of *Tet3[smKO]* mice. In pulmonary vascular SMCs (VSMCs) of *Tet2/Tet3[smKO]*, the 5hmC levels were lower than in *Tet3[smKO]* VSMCs, but no serious vascular abnormalities were evident (Extended Data Fig. 2l,m). We conclude that TET3 is the main enzyme for maintaining normal 5hmC levels in BSMCs and that the functions of TET2 and TET3 overlap in pulmonary VSMCs.

### 5hmC prevents spurious entry of RNA polymerase II

Next, we determined the distribution of 5hmC in lung SMCs by Nano-5hmC-seal (Nano-seal), a nonantibody-based technique. Bioinformatics analysis disclosed genome-wide accumulation of 5hmC in gene bodies with an enrichment at proximal 5′-upstream regulatory regions and a sharp decline at the transcriptional start sites (TSS) (Extended Data Fig. 3a). Highly transcribed genes showed the strongest accumulation of 5hmC within gene bodies, while weakly expressed genes had much lower 5hmC levels (Fig. 2a). Inactivation of *Tet3* led to global reduction of 5hmC levels (Fig. 2b and Extended Data Fig. 3a). In contrast, genes expressed at very low levels (bottom 5%) showed no enrichment in gene bodies and were less affected by the loss of *Tet3* (Fig. 2a). These data suggest that TET3-mediated generation of intragenic 5hmC depends primarily on transcriptional activity.

Next, we determined the binding profiles of RNA polymerase II (Pol II) phosphorylated at Ser5 (Pol II pSer5) by chromatin immunoprecipitation with sequencing (ChIP–seq) after DRB-induced block of transcriptional elongation (Fig. 2c). Importantly, loss of *Tet3* increased binding of Pol II pSer5 to TSS and gene bodies (Extended Data Fig. 3b). Grouping of genes into quartiles (group a–d) based on Pol II pSer5 ChIP–seq data (log$_2$ *Tet3[smKO:T]*/Ctrl) yielded an even clearer view. Increased intragenic binding of Pol II pSer5 in *Tet3* mutant SMCs was correlated positively with transcriptional activity measured by higher levels of Pol II pSer5 binding at TSSs (Fig. 2d), RNA-seq reads (Fig. 2e) and enhanced 5hmC content within gene bodies of control SMCs (Fig. 2f and Extended Data Fig. 3c). These data indicate that *Tet3*-mediated formation of 5hmC at gene bodies prevents intragenic entry of Pol II into highly transcribed genes.

To dissect the mechanisms leading to preferential accumulation of 5hmC at gene bodies of highly transcribed genes, we investigated whether TET3 associates with the transcription elongation machinery. Coimmunoprecipitation (Co-IP) revealed that wildtype (WT) but not catalytically inactive TET3 interacts with pan-RNA Pol II and elongating Pol II (Pol II pSer2), which was further verified by in situ proximity ligation assays (PLA) (Fig. 3a,b and Extended Data Fig. 3d). We also

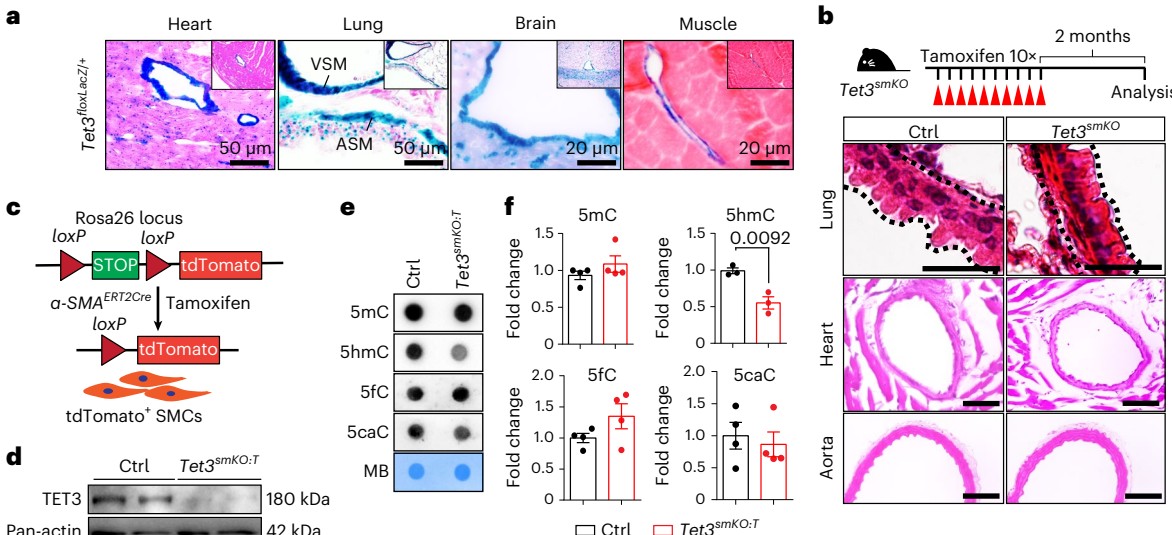

**Fig. 1 | TET3 is required to maintain 5hmC levels in SMCs. a**, LacZ staining of cryosections derived from different organs of *Tet3[floxLacZ/+]* mice (*n* = 3). Scale bars, 50 μm (heart and lung) or 20 μm (brain and muscle). **b**, H&E staining of paraffin sections from control or *Tet3[smKO]* lungs (*n* = 5). Scale bar, 200 μm (upper panel); 50 μm (middle panel); 10 μm (lower panel). The experimental design is depicted in the upper panel. **c**, Outline of the strategy to isolate tdTomato-labeled SMCs.

**d**, Western blot analysis of TET3 in sorted lung control and *Tet3[smKO:T]* SMCs. **e**, Dot blot analysis of 5mC (*n* = 4), 5hmC (*n* = 3), 5fC and 5caC (*n* = 4) levels in sorted lung SMCs from control and *Tet3[smKO:T]* mice. Methylene blue (MB) staining served as loading control. **f**, Quantification of 5mC, 5hmC, 5fC and 5caC levels from the dot blot analysis of SMCs (two-tailed unpaired *t*-test). *n* represents independent animals. Data are presented as mean values ± s.e.m.

found interactions of WT but not catalytically inactive TET3 with the H3K36 trimethyltransferase SETD2 and colocalization of 5hmC with SETD2 but not with the H3K36 dimethyltransferase NSD3 (Fig. 3a,b). Importantly, increased expression of WT but not catalytically inactive TET3 enhanced binding of SETD2 to Pol II (Fig. 3c), suggesting that TET3-mediated 5hmC formation stabilizes interactions of SETD2 with the RNA Pol II-containing elongation machinery, although SETD2 is able to interact directly with the carboxy-terminal domain of Pol II at pSer2[21,22]. The dramatic reduction of H3K36me3 within gene bodies in SMCs after inactivation of *Tet3* indicates a failure of SETD2 or H3K36me3-dependent repressive chromatin formation required to prevent entry of Pol II (refs. [3,4]) (Extended Data Fig. 3e,f). Integrated analysis of H3K36me3 ChIP–seq and RNA-seq or Pol II pSer5 ChIP–seq data revealed a strong decline in H3K36me3 levels in highly transcribed genes concomitant with a strong increase of intragenic Pol II entry following loss of *Tet3* (Fig. 3d,e). ChIP–qPCR further validated substantial reduction of H3K36me3 in intragenic regions of highly expressed genes such as *Acta2*, *Cnn1*, *Myh11*, *Dbn1*, *Arhgap18* and *Lpxn* in *Tet3* mutant SMCs, which was not observed in low-expressed genes (Extended Data Fig. 3g). Taken together, our findings indicate that TET3 and/or 5hmC facilitate recruitment of SETD2 and subsequent H3K36me3 deposition within transcribed gene bodies, preventing ectopic entry of Pol II pSer5 to gene bodies in SMCs.

### Aberrant transcripts in *Tet3* mutant SMCs

To identify genes with spurious intragenic transcription, we analyzed RNA-seq data of SMCs isolated from *Tet3[smKO:T]* mice by calculating the ratio between the RPKM (reads per kilobase per million mapped reads) of intermediate and first exons (Extended Data Fig. 4a,b). Of all genes containing more than four exons, 7,761 had a log₂ ratio greater than one of all intermediate exons from second exon onwards versus the first exon in *Tet3*-deficient SMCs (Fig. 4a). To detect bona fide cryptic transcription initiation events, we performed Cap-analysis gene expression-sequencing (CAGE–seq), which identifies transcription start sites (TSSs) at single-base pair resolution[23,24]. Importantly, the number of intragenic CTSS (defined as TSS with CAGE tag greater than eight, the average value of each single-base TSS on annotated

TSSs) increased significantly in *Tet3*-deficient SMCs (Extended Data Fig. 4c). The frequency of ectopic intragenic transcriptional initiation correlated positively with transcription activity indicated by CAGE signals at canonical TSSs, which corresponds well to Pol II ChIP–seq data (Fig. 2d and Extended Data Fig. 4d). Since localization of 5hmC is strongly asymmetric, with significantly higher levels on the sense strand[25], we focused on 2,114 genes that contain intragenic CTSS on the sense-strand-specific for *Tet3*-deficient SMC (Fig. 4a). Out of the 2,114 genes, 515 (24%) showed increased ratios of RNA-seq reads between downstream and first exons as well as enhanced Pol II intragenic entry in mutant SMCs, and were therefore designated spuriously expressed genes (Fig. 4a). Genes with high transcriptional activity and epigenetic signatures, including increased DNA accessibility and H3K4me2/3 deposition generated more spurious transcripts than other genes (Fig. 4b,c and Extended Data Fig. 4e–g). Such genes code for contractile actin filament bundle and actomyosin structure organization (for example, *Acta2*, *Cnn1*, *Myh11*, *Dbn1*) and pathways important for sarcoplasmic reticulum function and SMC contraction such as 'Focal adhesion,' 'Calcium signaling pathway' and 'Inositol phosphate metabolism'[26] (Fig. 4d,e and Extended Data Fig. 4h,i). As a consequence of enhanced spurious transcription, more RNA-seq reads of highly expressed contractile SMCs genes, for example, *Acta2*, *Arhgap18* and *Cnn1*, were recorded after *Tet3* inactivation, albeit concentrations of functional full-length mRNAs detected by semiquantitative PCR dropped (Fig. 4e and Extended Data Fig. 4j). We reason that the reduced presence of full-length mRNAs for contractile functions is not caused by reduced transcriptional activity, since (1) transcriptional activity at respective loci is not diminished and (2) expression of key transcription factors driving SMC gene expression (for example, *Klf4* and myocardin (*Myocd*)) remained unchanged (Extended Data Fig. 4k).

The drop in 5hmC accumulation after inactivation of *Tet3* was particularly evident within gene bodies, where 5hmC levels are high compared with proximal 5′-upstream regulatory regions, indicating a more important role of TET3 in transcriptional elongation than transcriptional initiation in this subset of genes (Extended Data Fig. 5a,b). hMeDIP–qPCR confirmed a marked reduction of 5hmC levels specifically at intragenic but not promoter regions of highly expressed

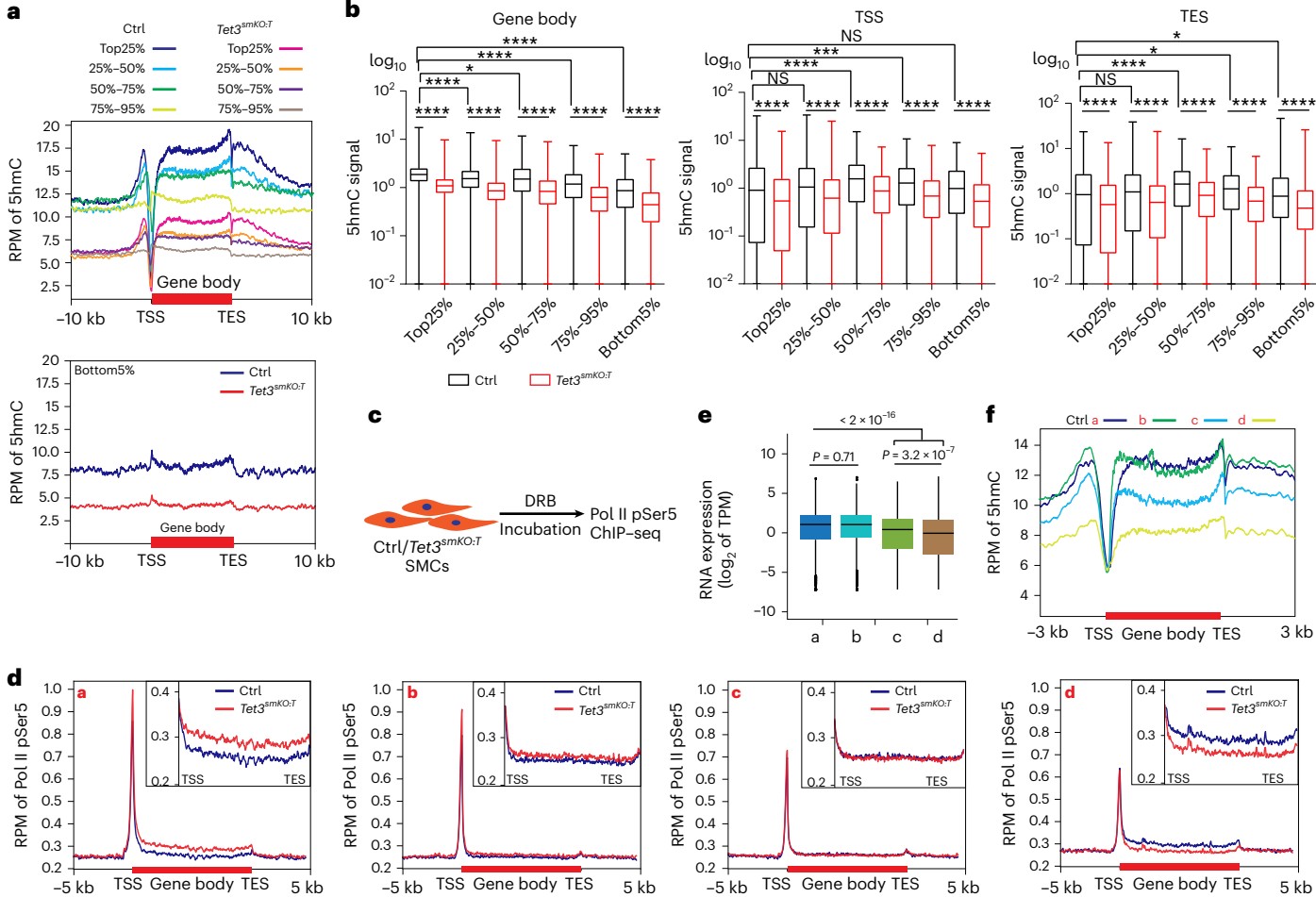

**Fig. 2 | *Tet3* inhibits aberrant entry of Pol II into gene bodies with high 5hmC content. a**, Overlap of Nano-seal-seq and RNA-seq datasets from sorted lung SMCs of control and *Tet3^smKO:T^* mice demonstrating a positive correlation of 5hmC enrichment and gene expression levels. Genes were divided into five groups based on RNA-seq expression levels in control lung SMCs. The distribution of 5hmC at TSS, gene body and TES in each group is shown. **b**, Boxplots represent 5hmC Nano-seal-seq signals within gene body, TSS and TES region of distinct gene groups as in **f** (*n* = 2 independent animals; one-way ANOVA with Tukey's post hoc test: *\**P* < 0.05, *\*\*\**P* < 0.001, *\*\*\*\**P* < 0.0001). The boxplot displays the median with min (bottom value) to max (top value). **c**, Depiction of the experimental

strategy. Sorted lung SMCs were treated with DRB before Pol II pSer5 ChIP. **d**, ChIP–seq analysis of intragenic Pol II pSer5 entry in control and *Tet3*-deficient SMCs in genes grouped into quartiles (**a**–**d**) according to Pol II pSer5 ChIP–seq data (log₂ *Tet3^smKO:T^* per Ctrl). Insets, zoomed-in view of gene bodies. **e**, RNA-seq analysis of transcriptional activity in quartiles defined by the degree of Pol II pSer5 binding (*n* = 2). *P* values were calculated with the Kruskal–Wallis test followed by the Wilcoxon rank sum test. Data in **e** are presented as mean values ± s.e.m. **f**, Analysis of 5hmC accumulation in quartiles defined by the extent of Pol II pSer5 binding.

spurious genes coding for contractile proteins after *Tet3* depletion. In contrast, 5hmC levels of low-expressed nonspurious genes were not affected (Extended Data Fig. 5c). Intriguingly, 5mC levels were not significantly altered within either intragenic or promoter regions of highly expressed contractile and low-expressed synthetic genes after loss of *Tet3* (Extended Data Fig. 5d). This finding indicates that dynamic formation of 5hmC and deposition of H3K36me3 rather than the mere presence of 5mC alone plays a decisive role in preventing spurious transcription.

Of note, we found a significant enrichment of CpG dinucleotides and transcription factor binding motifs containing CpG sequences, including motifs related to *Sp2* and members of the Ets family within 50 base pairs (bp) of TET3-dependent intragenic CTSSs in spuriously expressed genes (Fig. 4f). CAGE–seq analysis confirmed that ectopic transcriptional initiation occurred specifically at intragenic binding motifs of contractile genes such as *Acta2* and *Myh11* in *Tet3*-deficient SMC (Fig. 4g). Importantly, 5hmC and H3K36me3 levels were both reduced in the vicinity of intragenic CTSSs of *Acta2* and *Myh11* genes in *Tet3*-deficient SMCs (Fig. 4g).

**Spurious transcripts activate TLR7 signaling**

To determine the functional impact of spurious transcripts, Kyoto Encyclopedia of Genes and Genomes (KEGG) analysis of RNA-seq data from *Tet3* mutant and control lung SMCs was performed. Intriguingly, the top 15 upregulated pathways were associated mainly with inflammatory responses (Fig. 5a). In particular, *Tet3* inactivation resulted in upregulation of genes involved in endosomal TLR7/8 signaling (that is, *Tlr7*, *Myd88, Ccl5*, *Il1b* and so on), which is normally activated by single-stranded RNA of viral origin causing production of cytokines and chemokines, and expression of a set of macrophage-enriched genes such as *Cd68*, *Adgre1* and *Lgals3* in SMC[27] (Fig. 5b and Extended Data Fig. 6a). Moreover, we detected enhanced levels of EEA1 and RAB7–proteins regulating endosome trafficking that colocalize with TLR7 in *Tet3*-deficient SMCs (Extended Data Fig. 6b). Recruitment of the adapter molecule MYD88 by TLR7 was increased substantially in *Tet3*-deficient SMCs (Fig. 5c), suggesting that aberrant spurious transcripts provoke activation of nucleic-acid-sensing TLRs, which bestow *Tet3*-deficient SMCs with macrophage-like properties–as seen during phenotype switching under pathological conditions[28].

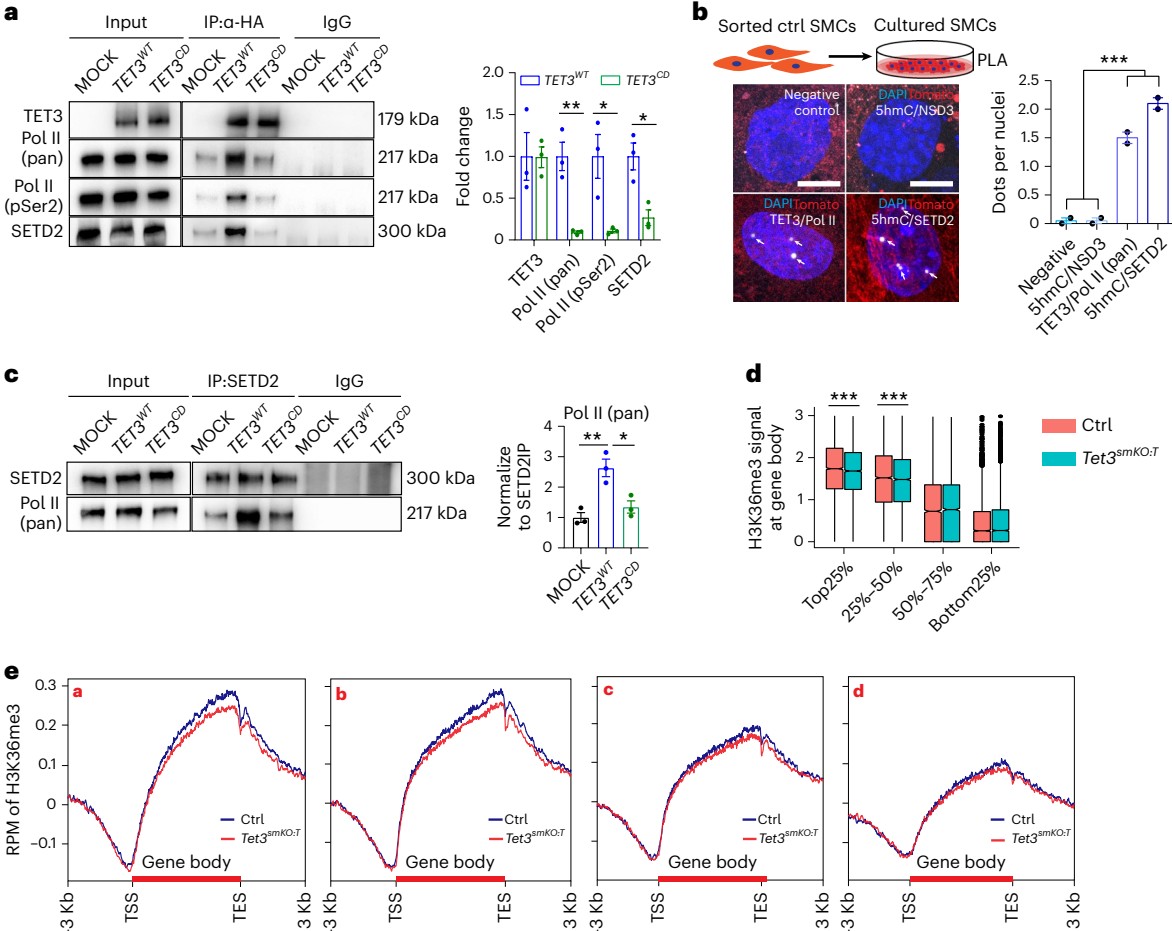

**Fig. 3 | TET3-dependent 5hmC formation stabilizes SETD2-Pol II recruitment, facilitating intragenic H3K36me3 deposition within highly transcribed genes. a**, Co-IP of WT (*TET3^WT^*) or catalytically inactive (*TET3^CD^*) HA-tagged human *TET3* after expression in HEK293T cells followed by western blot analysis (*n* = 3 independent experiments). Quantification of coprecipitated Pol II (pan), Pol II (pSer2) and SETD2 is shown on the right (*n* = 3; two-tailed unpaired *t*-test: **P = 0.0059, *P = 0.0276, *P = 0.0172). **b**, PLA to visualize interactions of 5hmC and NSD3, TET3 and Pol II (pan), and 5hmC and SETD2 in control SMCs (*n* = 2 independent animals). Positive PLA signals are indicated by white arrows. Quantification of average PLA signal per nuclei (%) in control SMCs is shown in

the right panel (*n* = 2). One-way ANOVA with Tukey's post hoc test: ***P < 0.001. Scale bar, 50 μm. **c**, Co-IP to detect interactions of Pol II with SETD2 in HEK293T cells following mock, *TET3^WT^* and *TET3^CD^* transfections (*n* = 3 independent experiments). Quantification of coprecipitated Pol II is shown on the right (*n* = 3, one-way ANOVA with Tukey's post hoc test: *P = 0.0156, **P = 0.005). **d**, Analysis of H3K36me3 signals within gene bodies of quartiles defined by RNA-seq analysis of transcriptional activity (*n* = 2). *P* values were calculated with the one-tailed likelihood-ratio test: ***P < 0.001. **e**, Distribution of H3K36me3 ChIP–seq signals within gene bodies of different subgroups of genes as defined in Fig. 2b (*n* = 2). Data in **a–c** are presented as mean values ± s.e.m.

To analyze whether spurious transcripts indeed activate TLR7 signaling, we transfected HEK293 and HeLa cells with whole cellular RNA extracted from control and *Tet3*-deficient SMCs. Expression levels of endosomal TLR7 downstream genes including *IRF7*, *IL1b*, *CCL5*, *CD86*, *IFNb*, *CXCL9* but not of *TLR7/MYD88* or the nonendosomal target *CCR5* were significantly elevated in HeLa cells by RNA from *Tet3*-deficient SMCs compared with control SMC RNA (Fig. 5d and Extended Data Fig. 6c). Absence of TLR7 signaling, as in HEK293 cells, or E6446-mediated TLR7 inhibition of HeLa cells, prevented such an increase, indicating that induction of innate immune responses by spurious transcripts depends on TLR7 (Fig. 5d and Extended Data Fig. 6d,e).

To confirm the hypothesis that TET3-mediated 5hmC formation prevents spurious transcription and subsequent inflammatory responses, we expressed either WT or catalytically inactive human TET3 in mESCs-derived SMCs after *Tet3* knockdown (*Tet3^KD^*) (Fig. 6a and Extended Data Fig. 6f). We verified that knockdown of *Tet3* reduces 5hmC and H3K36me3 formation (Fig. 6a and Extended Data Fig. 6g) and enhances TLR7-dependent expression of cytokine/chemokine genes, similar to *Tet3*-deficient primary SMCs (Extended Data Fig. 6h). Expression of WT but not catalytically inactive human TET3 normalized

expression of cytokine/chemokine genes in *Tet3^KD^* SMCs (Fig. 6b). Transfection of cellular RNA collected from *Tet3^KD^* SMCs stimulated cytokine/chemokine gene expression in recipient cells, which was blocked by TLR7 inhibition. Likewise, cellular RNA from *Tet3^KD^* SMCs lost this stimulatory effect when WT, but not catalytically inactive, human TET3 was expressed in the donor cells (Fig. 6c).

### Loss of *Tet3* causes airway inflammation

Characterization of the pathological responses in airways uncovered a switch from the contractile (spindle-shape with actin filament and dense bodies) to the synthetic (rhomboid-shape with rough endoplasmic reticulum state of SMCs, 2 months after *Tet3* inactivation (Fig. 7a and Extended Data Fig. 7a–c). In line with ultrastructural changes in SMCs, expression levels of miR-145a (a master regulator of SMC contractility[29]) and contractile markers such as α-SMA and MYH11 were decreased substantially in FACS-sorted *Tet3^smKO:T^* and in vitro differentiated *Tet3^KD^* SMCs (Fig. 7b,c and Extended Data Fig. 7d–f). In contrast, protein levels of synthetic marker genes such as TPM4 and VIM (vimentin) were elevated (Fig. 7b,c and Extended Data Fig. 7f). Moreover, we observed enhanced binding of Pol II pSer5 at intragenic CTSSs within

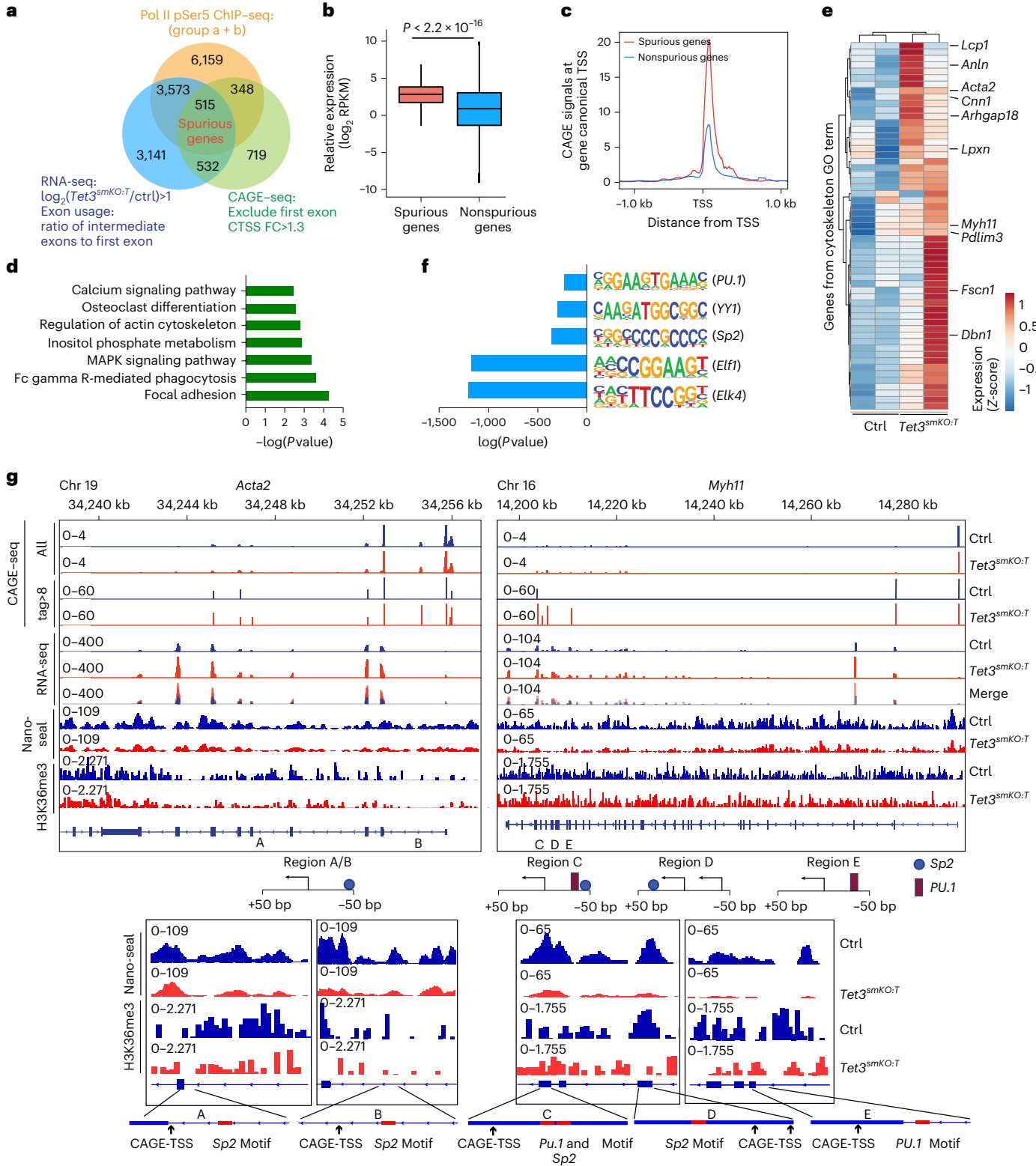

**Fig. 4 | *Tet3* inhibits aberrant intragenic initiation of transcription within highly expressed SMC genes. a**, Venn diagram based on integrated analysis of ChIP–seq, RNA–seq and CAGE–seq datasets to identify spuriously transcribed genes. **b**, Transcriptional activity of spurious and nonspurious genes assessed by RNA-seq. *P* values was as calculated using the one-tailed likelihood-ratio test. **c**, CAGE–seq signals at canonical TSSs of spuriously and nonspuriously transcribed genes. **d**, KEGG pathway analysis of spuriously transcribed genes shown in **a** (*n* = 2 independent animals). *P* value was calculated with two-sided, Fisher's exact test. **e**, Heatmap reflecting normalized counts transformed by *z* score from RNA-seq of sorted SMCs (*n* = 2 independent animals). Genes involved in cytoskeleton formation are shown. Genes related to contractile actin filament bundle and actomyosin structure organization are indicated. GO, gene ontology. **f**, Histogram of *Tet3*-deficiency-dependent enrichment of CpG-containing motifs in a region ± 50 nucleotides of intragenic TSSs (CAGE tag >8) in SMCs (*n* = 2 independent animals). *P* value was calculated using Pearson's correlation coefficient test. **g**, IGV tracks displaying the first single nucleotide of CAGE–seq capture sequences (CAGE tag >8) and RNA-seq peaks, Nano-seal and H3K36me3 ChIP–seq signals in *Acta2* and *Myh11* genes. Bottom, schematic representation of CAGE-TSS, putative transcription factor binding sites and gene tracks views of Nano-seal, H3K36me3 ChIP–seq signals within genomic regions containing putative transcription factor binding sites.

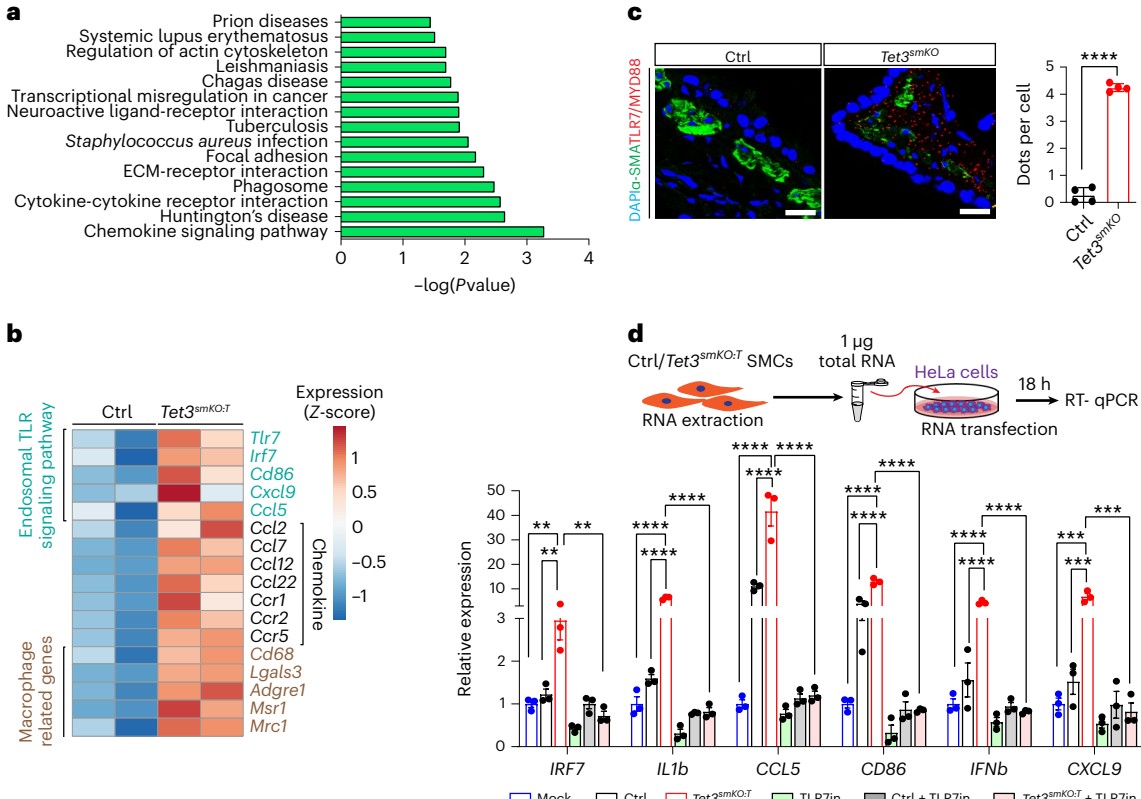

**Fig. 5 | Spurious transcripts activate the endosomal TLR7 signaling pathway in SMCs. a**, KEGG pathway analysis of RNA-seq data from SMCs of control and *Tet3^{smKO:T}* lungs (*n* = 2 independent animals). *P* value was calculated using the two-sided, Fisher's exact test. **b**, The heatmap represents normalized counts transformed by the *z* score of differentially expressed genes involved in TLR signaling, chemokine signaling and selected macrophage related genes (*n* = 2; log₂(fold change) >0.585, Wald-test with Benjamini–Hochberg correction: *P* ≤ 0.05). **c**, In situ PLA to visualize the interaction between TLR7 and MYD88 in airway SMCs. Nuclei were counterstained with DAPI. Quantification of PLA signals is shown in the right panel (*n* = 3 independent animals; two-tailed, unpaired *t*-test: ****P* < 0.0001). Scale bar, 50 μm. **d**, RT-qPCR analysis of HeLa cells transfected with total RNAs from control SMCs, *Tet3^{smKO:T}* SMCs and after mock transfection with or without TLR7 inhibitor (E6446) treatment (*n* = 3 independent experiments; one-way ANOVA with Tukey's post hoc test: ***P* < 0.01; ****P* < 0.001; *****P* < 0.0001). The experimental strategy for mRNA transfection is depicted in the upper panel. Data in **c** and **d** are presented as mean values ± s.e.m.

contractile but not synthetic marker genes, resulting in elevated transcription of intermediate exons and impaired production of full-length mRNA transcripts of contractile, but not synthetic, genes (Fig. 7d–f). Expression of WT but not a catalytically inactive human TET3 prevented phenotypic changes induced by *Tet3* suppression in mESC-derived SMCs (mESC-SMCs), conclusively demonstrating that the phenotype switch of SMCs relies on the reduction of 5hmC (Fig. 7d–f).

The phenotype switch after *Tet3* inactivation did not enhance proliferation of SMCs as indicated by unchanged numbers of Ki67+ SMCs (Extended Data Fig. 7g). Instead, we detected more SA-β-Gal-positive cells in the bronchial smooth muscle layer and elevated expression of senescence marker genes, for example, *p16* (*Cdkn2a*) and *p21* (*Cdkn1a*) in *Tet3*-deficient SMCs (Extended Data Fig. 7h,i). Acquisition of a senescence-associated secretory phenotype (SASP) by SMCs might enhance paracrine effects on neighboring cells in the lung. In fact, we detected concomitant upregulation of interferon response-related genes in lung SMCs and epithelial but not in endothelial cells, although the switch from the contractile to synthetic state was also observed in VSMCs of the lung by electron microscopy (EM) (Extended Data Fig. 7b,c and Extended Data Fig. 8a,b). Consequences of putative paracrine effects of *Tet3*-null SMCs were further examined by culturing mouse epithelial lung cells (MLE12 cells) with conditioned medium from control and *Tet3^{KD}* SMCs. Conditioned medium from *Tet3^{KD}* SMCs increased expression of pro-inflammatory genes (*Il6*, *Il1b* and *Ifnb*) and EMT related genes (*Fn1*, *Cdh1*, *Vim*). Expression of WT but not of catalytically inactive

human TET3 in *Tet3^{KD}* SMCs abolished this effect (Extended Data Fig. 8c). We conclude that bronchial SMCs are particularly susceptible to innate immune responses, eliciting adverse effects on neighboring epithelial cells, whereas vascular SMCs may require additional noxae to induce pathological vascular responses.

Furthermore, we noted a pronounced metaplasia of Club (CCSP+) but not ciliated cells (α-tubulin+) to mucus-producing goblet cells (Mucin5AC+, AGR2+ or PAS+) and excessive extracellular matrix deposition (Collagen I+) 2 months after SMC-specific *Tet3* inactivation (Fig. 8a and Extended Data Fig. 8d,f). Of note, *Tet3* inactivation did not lead to elevated expression of Mucin5AC in the intestinal epithelium, suggesting tissue-specific reactions (Extended Data Fig. 8g).

FACS analysis and immunofluorescence staining revealed substantially increased numbers of neutrophils, CD3+ T cells and interstitial macrophages but not of B cells and eosinophils in whole lung tissues, 2 months after *Tet3* inactivation (Fig. 8b and Extended Data Fig. 8h). At 6 months after *Tet3* inactivation, the lung phenotype had progressed further. We found massive peribronchiolar fibrosis and lesions in *Tet3^{smKO}* lungs, composed primarily of proliferative CD45R+ B cells (Fig. 8c and Extended Data Fig. 8i,j). In addition, the number of eosinophils was increased, while the rise of CD3+ T cells was no longer significant, although a moderate but significant increase in the number of Th2 cells was detected (Fig. 8d and Extended Data Fig. 8k). In line with this finding, production of the Th2 cytokines interleukin 4 (IL4), interleukin 13 (IL13) and interleukin 17a (IL17a), inducing Club cell metaplasia and enhanced mucin secretion[30], was strongly upregulated in the CD3+

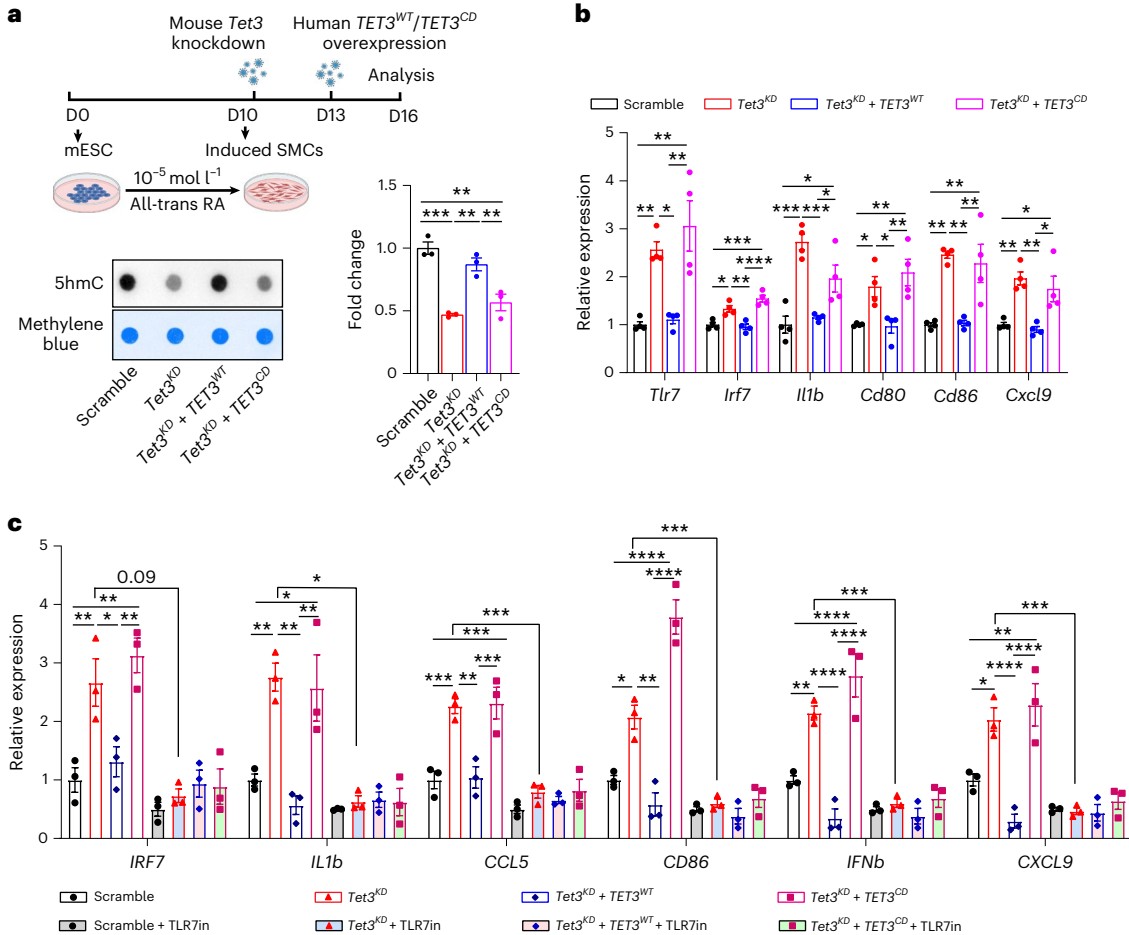

**Fig. 6 | TET3 prevents TLR7-dependent inflammatory responses by formation of 5hmC. a**, Dot blot analysis of 5hmC levels in mESC-SMCs after *Tet3* knockdown (*Tet3^KD^*) with or without expression of human *TET3^WT^* or *TET3^CD^* (*n* = 3 independent experiments). Quantification of 5hmC levels is shown on the right (*n* = 3; one-way ANOVA with Tukey's post hoc test: **P < 0.01, ***P < 0.001). The experimental strategy for manipulation of mESC-SMCs is depicted in the upper panel. **b**, RT-qPCR analysis of *Tlr7*, *Irf7*, *Il1b Cd80*, *Cd86* and *Cxcl9* in mESC-SMCs after transduction of scramble, *Tet3^KD^*, *Tet3^KD^ + TET3^WT^* and *Tet3^KD^ + TET3^CD^* lentiviruses

(*n* = 4 independent experiments; one-way ANOVA with Tukey's post hoc test: *P < 0.05; **P < 0.01; ***P < 0.001; ****P < 0.0001). **c**, RT-qPCR analysis of HeLa cells transfected with RNA isolated from mESC-SMCs after transduction with scramble, *Tet3^KD^*, *Tet3^KD^ + TET3^WT^* and *Tet3^KD^ + TET3^CD^* lentiviruses with or without TLR7 inhibitor (TLR7in) E6446 (*n* = 3 independent experiments; one-way ANOVA with Tukey's post hoc test: *P < 0.05; **P < 0.01; ***P < 0.001; ****P < 0.0001). Data in **a–c** are presented as mean values ± s.e.m.

T cell fraction (Extended Data Fig. 8l), indicating secondary immune responses mediated mainly by B cells, eosinophils and Th2 cells.

The phenotype of *Tet3^smKO:T^* mice strongly resembles the clinical appearance of adult human asthma, in which innate immune responses cause phenotype switching of SMCs but also bears some similarities to COPD[16,17,31]. Notably, we detected a strong reduction of 5hmC levels within bronchial SMC layer in human samples from asthma patients and in two distinct mouse models, which rely either on the use of house dust mites or *Aspergillus fumigatus* to induce asthma (Fig. 8e–h and Extended Data Fig. 9a–d). In contrast, we did not detect a decline of 5hmC in the bronchial SMC of human COPD and cystic fibrosis patients and in mouse lung after exposure to hypoxia for 28 days (10% O₂) (Extended Data Fig. 9e–h).

## Discussion

Previous studies unveiled that DNMT3B-dependent 5mC formation is critical to prevent inappropriate transcription at gene bodies[3,32]. Here, we propose that TET3-mediated 5hmC formation stabilizes interactions of SETD2 with the Pol II-containing elongation machinery, thereby facilitating H3K36me3 chromatin modifications, which, after passage of Pol II, prevents its re-entry (Extended Data Fig. 10a). Such a model is fully compatible with an essential role of DNMT3B but indicates that

5hmC is indispensable not only for allowing transcriptional elongation but also for preventing aberrant transcription initiation within gene bodies.

CAGE−seq data analysis revealed that spurious transcription initiation sites in *Tet3*-deficient SMC are enriched at CpG dinucleotides. This observation suggests that TET3-mediated oxidation of 5mC occurs primarily at heavily methylated cryptic intragenic TSSs or transposon elements to prevent aberrant transcriptional initiation. Since high density of CpG methylation (at least 90%) slows down elongation rates[1], TET3, which interacts with Pol II, will have more time to demethylate genomic loci with high 5mC density. Although the role of 5hmC after formation of H3K36me3 within the intragenic cryptic TSSs is still enigmatic, it is possible that 5hmC not only enhances recruitment of SETD2 but also serves as an intermediate towards DNA demethylation to support the elongating Pol II complex for overcoming obstacles imposed by DNA methylation. We also demonstrate that reduced 5hmC formation is associated with reduced H3K36me3 deposition in spuriously transcribed genes after *Tet3* inactivation. In contrast, the 5mC content of such genes did not change, further supporting the decisive function of 5hmC and H3K36me3 to prevent spurious transcription by favoring closed chromatin structures in gene bodies of SMCs[4,33].

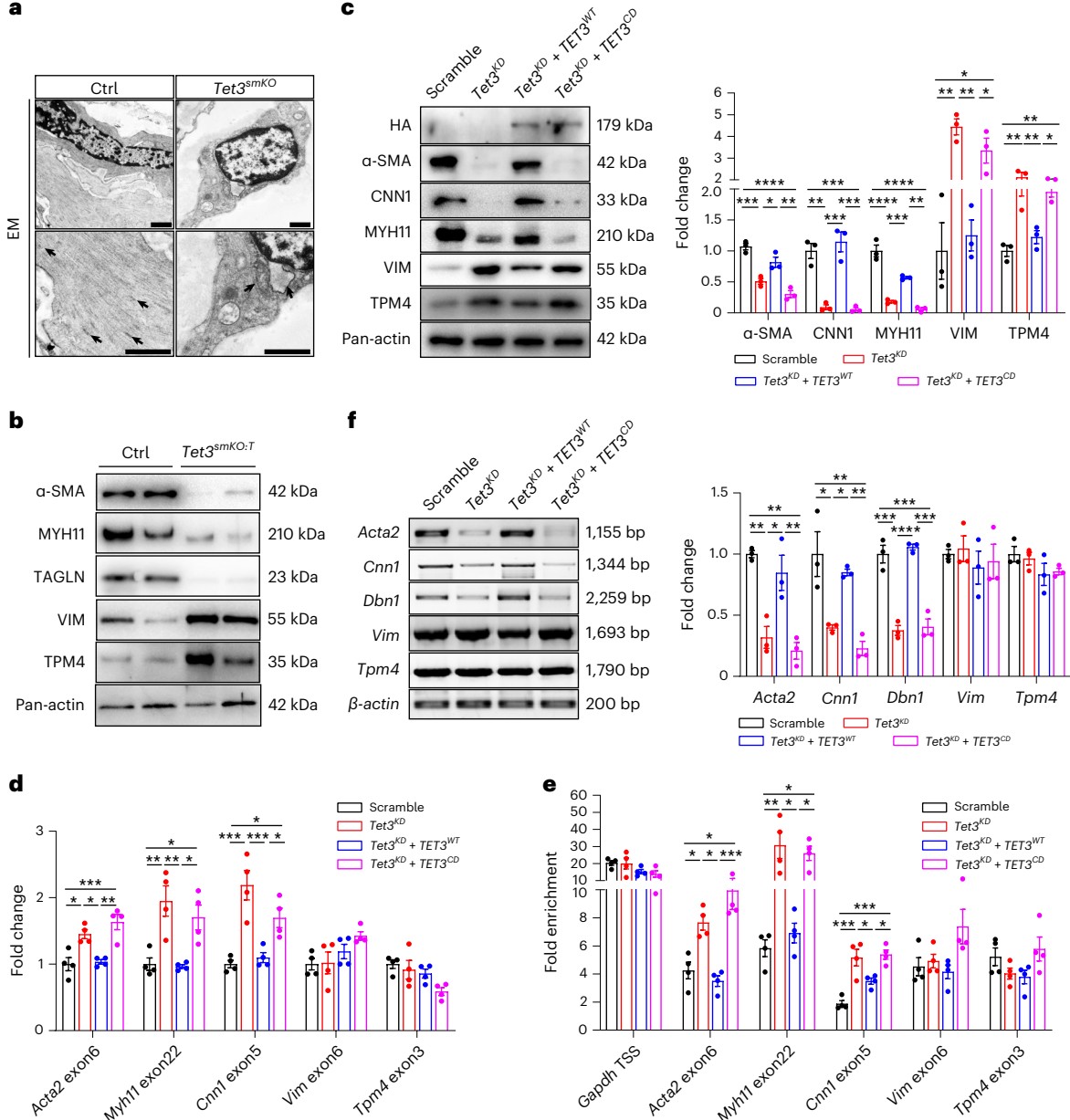

**Fig. 7 | Reduction of 5hmC causes a phenotype switch of BSMCs. a**, EM images of control and *Tet3^smKO^* bronchial SMCs (*n* = 4). Arrows point to dense bodies (control) and rough endoplasmic reticulum (*Tet3^smKO^*). Scale bar, 1,000 nm. **b**, Western blot analysis of sorted lung SMCs. Pan-actin was used as loading control. **c**, Western blot analysis of mESC-SMCs after transduction with scramble, *Tet3^KD^*, *Tet3^KD^* + TET3^WT^ and *Tet3^KD^* + TET3^CD^ lentiviruses. Pan-actin was used as loading control. Quantification of protein levels is shown on the right (*n* = 3 independent experiments; one-way ANOVA with Tukey's post hoc test: *$P < 0.05$; **$P < 0.01$; ***$P < 0.001$; ****$P < 0.0001$). **d**, RT-qPCR analysis of indicated genes in mESC-SMCs after transduction with scramble, *Tet3^KD^*, *Tet3^KD^* + TET3^WT^, *Tet3^KD^* + TET3^CD^ lentiviruses (*n* = 4 independent experiments; one-way ANOVA with Tukey's post hoc test: *$P < 0.05$; **$P < 0.01$; ***$P < 0.001$).

**e**, ChIP–qPCR to monitor Pol II pSer5 enrichment within gene bodies of indicated genes in mESC-SMCs after transduction with scramble, *Tet3^KD^*, *Tet3^KD^* + TET3^WT^, *Tet3^KD^* + TET3^CD^ lentiviruses with DRB treatment (*n* = 4 independent experiments; one-way ANOVA with Tukey's post hoc test: *$P < 0.05$; **$P < 0.01$; ***$P < 0.001$). **f**, Semiquantitative RT-PCR analysis of *Acta2*, *Cnn1*, *Dbn1*, *Vim* and *Tpm4* full-length mRNA in mESC-SMCs after transduction with scramble, *Tet3^KD^*, *Tet3^KD^* + TET3^WT^ and *Tet3^KD^* + TET3^CD^ lentiviruses. *β-actin* was used for normalization. Quantification was performed by Image J and is shown on the right (*n* = 3 independent experiments; one-way ANOVA with Tukey's post hoc test: *$P < 0.05$; **$P < 0.01$; ***$P < 0.001$; ****$P < 0.0001$). Data in **c–f** are presented as mean values ± s.e.m.

In addition to TET3, SMCs also express TET2 but not TET1. A recent study reported that TET2 acts as a master regulator of SMC plasticity by increasing chromatin accessibility at promoters of key procontractile genes[18]. Notably, we found that TET2 depletion neither attenuates 5hmC levels nor causes lung abnormalities in vivo, suggesting that TET3 compensates for the absence of TET2 in several differentiated tissues in vivo, at least under baseline conditions. This observation is consistent with upregulation of TET3 in various differentiated organs

of *Tet1* and *Tet2* double knockout mice[10]. Concomitant inactivation of *Tet2* and *Tet3* in SMCs further reduced 5hmC in pulmonary VSMCs but not in BSMCs, although the gross morphology of vessels was unaffected in *Tet2/Tet3* compound mutants. We assume that the function of *Tet2* and *Tet3* to generate 5hmC partially overlaps in VSMCs. The question remains why BSMCs are more vulnerable to the loss of 5hmC than VSMCs, which might be due to a lower rate of spurious transcription in VSMCs compared with BSMCs or a lower threshold of BSMCs to activate

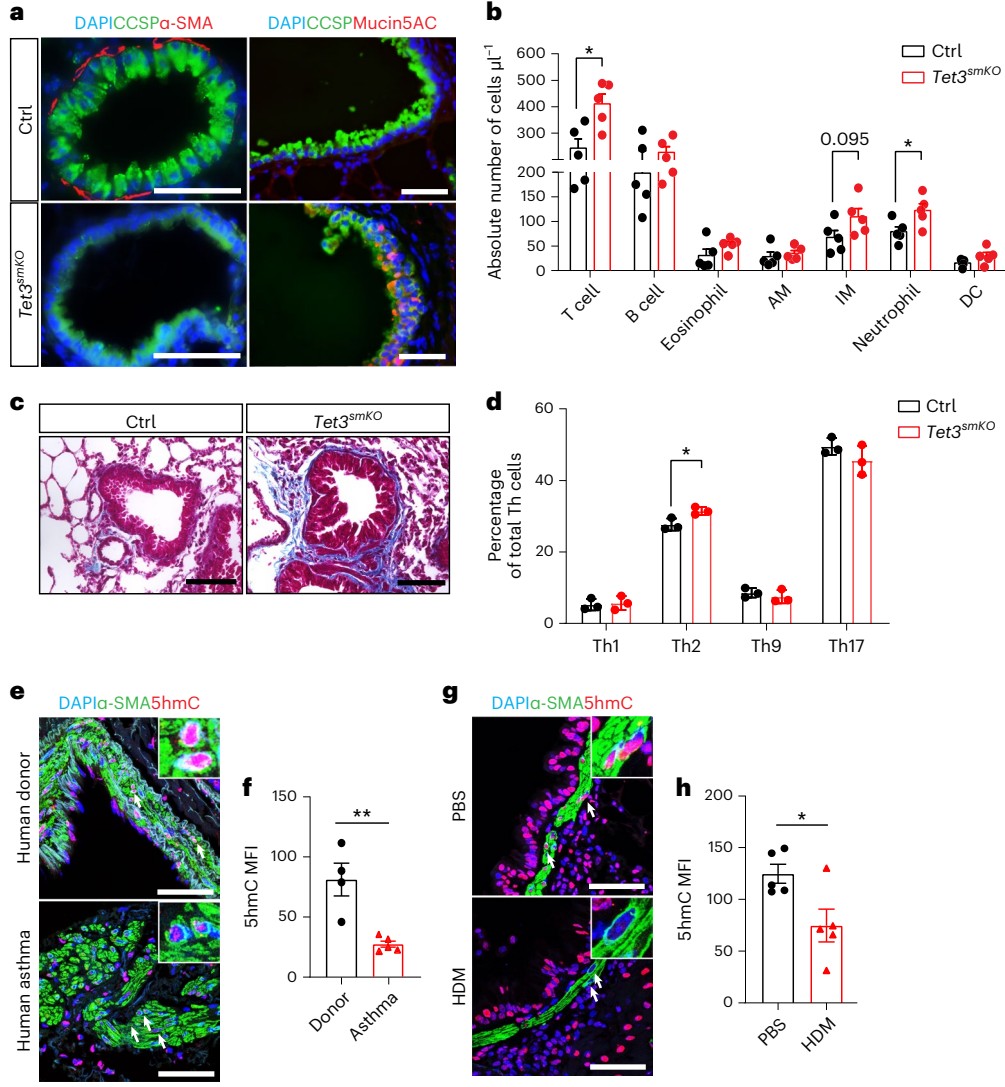

**Fig. 8 | Inactivation of *Tet3* in SMCs causes an asthma-like pathology.**
**a**, Immunofluorescence analysis of CCSP and α-SMA (upper panel), and CCSP and Mucin5AC (lower panel) using paraffin sections prepared from control and *Tet3smKO* lungs 8 weeks after tamoxifen injection (*n* = 3). Scale bar, 50 μm. DNA was stained by DAPI. **b**, FACS analysis of absolute numbers of different immune cells in whole lungs from control and *Tet3smKO* mice 8 weeks after tamoxifen injection (*n* = 5 independent animals; two-tailed unpaired *t*-test: *\*P* = 0.0113, *\*P* = 0.0359). AM, alveolar macrophages; IM, interstitial macrophages. **c**, Trichrome staining of cryosections prepared from control and *Tet3smKO* lungs, 6 months after tamoxifen injection (*n* = 5). Scale bar, 50 μm. **d**, FACS analysis of the percentage of different T helper cells (Th) within the CD3+ T cell fraction in the lung, 6 months after tamoxifen injection (*n* = 3 independent animals; two-tailed unpaired *t*-test: *\*P* = 0.0298). **e**, Immunofluorescence staining for

α-SMA and 5hmC on lung paraffin sections from donor (*n* = 4) and asthma patients (*n* = 5). Scale bar, 50 μm. DNA was stained by DAPI. Arrows indicate 5hmC-positive SMCs in control samples. Insets, enlarged images of 5hmC-stained SMCs from the respective panels. **f**, Quantification of MFI of 5hmC was performed by Image J (*n* = 4 independent samples; two-tailed unpaired *t*-test: *\*\*P* = 0.004). **g**, Immunofluorescence staining for α-SMA and 5hmC on lung paraffin sections from PBS- and house dust mites (HDM) -treated mice (*n* = 5). Scale bar, 50 μm. DNA was stained by DAPI. Arrows indicate 5hmC-positive SMCs in control samples. Insets, enlarged images of 5hmC-stained SMCs from the respective panels. **h**, Quantification of MFI of 5hmC was performed by Image J (*n* = 5 independent animals; two-tailed unpaired *t*-test: *\*P* = 0.0259). The enlarged images in **e** and **g** were scaled to 150 × 150 pixels. Data in **b**, **d**, **f** and **h** are presented as mean value ± s.e.m.

innate immune responses. We assume that the switch from a contractile to a synthetic phenotype in *Tet3*-deficient pulmonary VSMCs is the consequence of massive lung inflammation caused by *Tet3*-deficient BSMCs, since *Tet3* mutant VSMCs in the aorta do not show a reduction in the expression of contractile genes. Alternatively, a differential responsiveness of neighboring cells to activate innate immune responses may contribute, which is supported by the upregulation of interferon response-related genes in epithelial but not in endothelial cells via conditioned medium from *Tet3* mutant SMCs.

Our study unveils the biological consequences of inappropriate cryptic transcription in mammals, which has not previously been adequately addressed[5,6]. We discovered that accretion of aberrant

intragenic transcripts in *Tet3*-deficient SMCs activates the TLR7 nucleic-acid-sensing system, subsequently provoking immune responses and lung pathogenesis (Extended Data Fig. 10b). Generation of spurious transcript might interfere with proper modification of self-RNAs, such as 2′-O-methylation, pseudouridine (Ψ), 5-methylcytidine (m5C), 2-thio-uridine (s2U) or N6-methyladenosine (m6A)[34], or provoke removal of such modifications, thus misleading cells to recognize them as foreign. Spurious transcripts might also resemble ssRNA degradation products or become processed into products recognized by TLR7/8. Currently, we do not know which specific properties of spurious transcripts enable activation of the endosomal TLR7/8 signaling pathway, but it is evident that such products are

stable enough to elicit an innate immune response after transfection into other cells.

Our study highlights the central role of SMCs in lung disease, demonstrating that alterations of contractility and initiation of innate immune responses in SMCs are sufficient to stage a massive inflammatory reaction, which compromises airway function and results in an asthma-like phenotype. The observation of reduced 5hmC formation in bronchial SMCs in asthma patients and in two different mouse asthma models is intriguing. The discovery that loss of *Tet3* causes an asthma-like phenotype strongly suggests that the reduction of 5hmC in airways of human asthma patients is not an epiphenomenon but is causally involved in the pathogenesis of asthma. We speculate that modulation of TET3 activity and/or 5hmC formation might be a viable approach to interfere with chronic innate immune responses, initiated or maintained by inappropriate cryptic transcription.

## Online content

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

## Methods

### Study approval

Approval to use human samples from the BioMaterialBank Nord for research was granted by the Ethics Committee of the University of Lübeck (Az 12-220 and 14-225). Tissue donations from the DZL Biobank (Deutsches Zentrum für Lungenforschung) was approved by the Ethics Committee of the Department of Human Medicine of Justus Liebig University Hospital, in accordance with national law and with the 'Good Clinical Practice/International Conference on Harmonization' guidelines. Written informed consent was obtained from each patient or the patient's next of kin (Az. 58/15 and 111/08). All animal experiments were done in accordance with the Guide for the Care and Use of Laboratory Animals published by the US National Institutes of Health (NIH Publication No. 85-23, revised 1996) and were approved by the responsible Committee for Animal Rights Protection of the State of Hessen (Regierungspraesidium Darmstadt) with the project numbers B2/1125, B2/1137 and B2/1056.

### Animals

$Tet3^{floxLacZ/+}$ and $Tet3^{fl/fl}$ mice were generated in house by using a targeting vector purchased from the European Conditional Mouse Mutagenesis Program (EUCOMM). $\alpha$-$SMA^{ERT2Cre2}$ transgenic mice were provided by P. Chambon (IGCMB Strasbourg). $ROSA26^{tdTomato}$ mice were obtained from The Jackson Laboratory. C57BL/6 mice were obtained from Charles River. All mice were maintained in individually ventilated cages, at 22.5 °C ±1 °C and a relative humidity of 50% ±5% with controlled illumination (12 h dark/light cycle). Mice were given access to food and water ad libitum. All mouse strains were backcrossed and maintained on a C57BL/6 genetic background. Primers used for genotyping are listed in (Supplementary Table 1). All experiments were performed using approximately equal numbers of male and female mice, since preliminary data did not indicate significant differences between females and males in respect to changes in airway morphology. Tamoxifen (Sigma) was administered intraperitoneally at 75 mg kg$^{-1}$ body weight daily for 10 days starting from 8 weeks old. In all experiments, mice without the respective floxed allele but containing the Cre-recombinase expressing allele and/or the tdTomato reporter served as controls, unless indicated otherwise.

### Isolation of SMCs and epithelial and endothelial cells

After sacrificing experimental mice, blood was removed by perfusion with cold PBS through the right ventricle before lung dissection. Lung tissues were dissected and minced into small pieces before incubation in 3 ml digestion buffer (DPBS containing Collagenase type 2 (2 mg ml$^{-1}$, Worthington), Elastase (0.04 mg ml$^{-1}$, Worthington) and DNase (5U ml$^{-1}$, Roche) with frequent agitation at 37 °C for 10 min. Immediately afterwards, ten times the volume of cold DMEM supplemented with 10% fetal bovine serum (FBS) was added to single-cell suspensions. Cells were dissociated mechanically by passing four to five times through a 30 ml syringe and consecutive filtering through 100-, 70- and 40-μm cell strainers (BD Biosciences). The filtrate was centrifuged at 300 $g$ at room temperature (RT) for 10 min. Pellets were resuspended in 1 ml precooled MACS buffer (catalog no. A9576, Miltenyi Biotec) with 1% BSA. After 5 min centrifugation at 300 $g$, 4 °C, cell pellets were resuspended in 90 μl MACS buffer and incubated with 10 μl CD45 MicroBeads (catalog no. 130-052-301) and anti-Ter-119 MicroBeads (catalog no. 130-049-901) at 4 °C for 15 min to remove hematopoietic cells. After washing with MACS buffer, cells were loaded into preconditioned LS columns (Miltenyi Biotec) on a MACS separator and the flow-through containing unlabeled cells was collected. 4,6-Diamidino-2-phenylindole (DAPI)$^-$ and tdTomato$^+$ populations were sorted using a FACSAria III (BD Biosciences). (Extended Data Fig. 1m and Supplementary Figs. 1 and 2). Epithelial and endothelial cells were isolated using anti-EpCAM MicroBeads (catalog no. 130-105-958) and anti-CD31 Microbeads (catalog no. 130-097-418), respectively.

### In vitro differentiation of mouse embryonic stem cell-derived SMCs

The method for differentiation of SMC from mouse embryonic stem (ES) cells was based on a published protocol[35]. Briefly, resuspended single mouse ES cells were plated on plates coated with gelatin at a density of $4 \times 10^4$ cm$^{-2}$ at 37 °C, 5% $CO_2$ in differentiation medium (DMEM (Sigma) supplemented with 10% fetal calf serum FCS (Sigma), 1 mM L-glutamine, 0.1 mM l$^{-1}$ 2-mercaptoethanol, 0.1 mM l$^{-1}$ nonessential amino acids, 100 U penicillin and $10^{-5}$ mol l$^{-1}$ all-trans RA (Sigma)). Cells were cultured for 8–10 days with a daily change of fresh RA-containing differentiation medium.

### Lentivirus infection

HEK293T cells ($2 \times 10^6$ per 10 cm dish) were transfected with either 5 μg pLKO.1-$Tet3$-shRNA (shRNA sequence: sh1, 5′-CTGTTAG GCAGATTGTTCT; and sh2, 5′-TCCAACGAGAAGCTATTT), which does not target human $Tet3$, pLJM1-$TET3^{WT}$ or pLJM1-$TET3^{CD}$ (mutation at H1077Y and D1079A), together with 4.5 μg psPAX2 (Addgene, catalog no. 12260), and 0.5 μg pMD2.G (Addgene, catalog no. 12259) using the Turbofect transfection reagent and Opti-MEM for 6–8 h. The lentivirus-containing supernatant was collected at 48 and 72 h after transfection, pooled and filtered through a 0.45 μM cell strainer to remove HEK293T cells. Lentiviruses were concentrated with a Lenti-X concentrator according to the manufacturer's instructions (TaKaRa, catalog no. 631231). Differentiated SMCs were infected with the $Tet3$ shRNA lentivirus with Polybrene (8 μg ml$^{-1}$) for 6–8 h, followed by infection with either $TET3^{WT}$ and $TET3^{CD}$ lentiviruses 3 days later.

### Human asthma, donor, COPD and cystic fibrosis samples and mouse asthma and chronic hypoxia samples

Human asthma samples were received from the BioMaterialBank Nord, Clinical and Experimental Pathology Medicine, Research Center Borstel. Human donor samples, human COPD samples and human cystic fibrosis samples were provided by the DZL Biobank were obtained during lung transplantation of human COPD and cystic fibrosis patients[36]. Donor lung material was obtained as a result of atypical resections undertaken to adjust the donor organ to the recipient's thoracal cavity. Clinical characteristics of patients and donors are provided in Supplementary Table 2. Experimental asthma in mice was induced by intranasal (IN) application of house dust mite (HDM) allergen whole-body extracts (Greer Laboratories) derived from the common HDM species *Dermatophagoides pteronyssinus* (*Der p*) and *Dermatophagoides farinae* (*Der f*)[37,38]. In the second model, experimental asthma was induced by either intraperitoneal (IP) or subcutaneous (SC) injection of *A. fumigatus* (ASP) followed by IN challenges with ASP[39]. Control animals were treated with PBS. For chronic hypoxia experiments, mice were kept under normobaric hypoxia (10% $O_2$) or normobaric normoxia (21% $O_2$) in a ventilated chamber (Biospherix) for 28 days. All animal studies were reviewed and approved by the Federal Authorities for Animal Research of Regierungspräsidium Giessen, Hessen, Germany (animal protocols G61/2019 and G27/2020, Gi 09/2017 for the hypoxia mouse model) and were carried out according to the guidelines of the German Animal Welfare Act.

### Gene expression analysis

Total RNA was extracted using TRIzol reagent (Invitrogen), following the manufacturer's instructions. RNA was reverse-transcribed with Superscript II (Invitrogen) following standard procedures. Real-time PCR was performed with two technical replicates using the StepOne Real-time PCR system and KAPA SYBR FAST qPCR Master Mix (KAPA Biosystems). Relative quantitation of gene expression was performed using the ΔΔCT method. Ct values of the target genes were normalized to the β-actin gene using the equation $\Delta Ct = Ct_{reference} - Ct_{target}$ and expressed as ΔCt. Relative mRNA expressions are shown with the average from control samples set as 1. Primers and PCR conditions are listed in Supplementary Table 1.

## Immunohistochemistry, immunofluorescence and histological analysis

After perfusion with PBS, tissues were dissected and immediately fixed in 4% paraformaldehyde (PFA). For paraffin sections, samples were dehydrated following standard protocols and sectioned at 7 μm after paraffin embedding for immunofluorescence, hematoxylin/eosin (H&E) and trichrome staining using established techniques. For cryosections, fixed tissues were equilibrated in 30% sucrose/PBS at 4 °C overnight and frozen on dry ice. Sections (7 μm) were mounted on SuperFrost slides for immunofluorescence or periodic acid-Schiff (PAS) staining using a kit from Sigma. Immunofluorescence images were acquired with a Leica M205 FA and a ZEISS Imager Z1. Acquisition of immunohistochemistry and histological images was performed with a ZEISS Axioplan2. 5hmC signals were determined by quantifying the average mean fluorescence intensity (MFI) per nucleus of 100 randomly selected α-SMA⁺ cells in lung tissue section of individual mouse and human subjects using Image J. $N$ numbers refer to the number of individual mouse and human subjects. Antibodies for immunofluorescence staining are listed in Supplementary Table 3.

## Western blot and dot blot assays

Sorted SMCs were incubated in lysis buffer (20 mM Tris-HCl, pH 8.0, 200 mM NaCl, 1 mM EDTA, 1 mM EGTA, 1% Triton X-100) and resolved by SDS–PAGE before transfer to nitrocellulose filters. Dot blot assays were performed with 100 ng genomic DNA using a Bio-Dot Microfiltration apparatus (catalog nos. 170-6545 and 170-6547). Protein expression was visualized using an enhanced chemiluminescence detection system (GE Healthcare) and quantified using the ChemiDoc gel documentation system (Bio-Rad). Antibodies are listed in Supplementary Table 3.

## Electron microscopy

Lungs were isolated and fixed in 1.5% glutaraldehyde (v/v), 1.5% PFA (v/w) in 0.15 M HEPES (v/w), pH 8.0 at 4 °C for at least 24 h, and subsequently incubated with 1% osmium tetroxide for 2 h. Samples were stained en bloc with 50% saturated uranyl acetate, followed by sequential ethanol dehydration (30%, 50%, 75%, 95%), and embedded in Agar 100. Ultrathin sections were cut using an ultramicrotome and image acquisition was performed with a Philips CM10 electron microscope. All images were captured with a slow-scan 2K CCD camera.

## FACS analysis

Single-cell suspensions from lung were analyzed with different antibody panels: T cells were defined as CD3⁺; B cells were defined as B220⁺; eosinophils were defined as Siglec-F⁺CD11c⁻); alveolar macrophages (AMs) were defined as Siglec-F⁺ CD11c⁺ CD11B⁻F4/80⁺; interstitial macrophages (IMs) were defined as Siglec-F⁻ CD11c⁻CD11b⁺ F4/80⁺; neutrophils were defined as Siglec-F⁻ CD11c⁻CD11b⁺ Ly6G⁺; dendritic cells were defined as Siglec-F⁻ CD11cʰⁱ MHCIIʰⁱ⁴⁰. T helper type 1 (Th1) cells were defined as CD4⁺ CD183⁺; Th2 cells were defined as CD4⁺ CD194⁺ CD196⁻; Th9 cells were defined as CD4⁺ CD194⁻ CD196⁺; Th17 cells were defined as CD4⁺ CD194⁺ CD196⁺. CountBright Absolute Counting Beads (Thermo Fisher) was used to calculate absolute numbers of cells in the sample. Fluorescence compensation controls and fluorescence-minus-one (FMO) stain sets were used to identify cells within multicolor-stained samples. Flow cytometry was performed with the LSR Fortessa (BD Biosciences) analyzer. Data acquisition and analysis was done using BD FACS Diva v.8 software (Supplementary Fig. 2).

## RNA-seq

RNA was isolated from sorted SMC using the miRNeasy micro Kit (Qiagen) combined with on-column DNase digestion (DNase-Free DNase Set, Qiagen) to avoid contamination by genomic DNA. RNA and library preparation integrity were verified with BioAnalyzer 2100 (Agilent) or LabChip Gx Touch 24 (Perkin Elmer). Total RNA (50 ng)

was used as input for ribosomal depletion with RiboGone-Mammalian (Clontech) followed by library preparation using SMARTer Stranded Total RNA Sample Prep Kit (Clontech). Sequencing was performed on the NextSeq500 instrument (Illumina) using v.2 chemistry, resulting in average of 44 M reads per library with 1 × 75bp single-end setup. Raw reads were assessed for quality, adapter content and duplication rates with FastQC v.0.11.8 (http://www.bioinformatics.babraham.ac.uk/projects/fastqc). Trimmomatic v.≥0.36 was employed to trim reads after a quality drop below a mean of Q15 in a window of five nucleotides[41]. Only reads of at least 15 nucleotides were cleared for subsequent analyses. Trimmed and filtered reads were aligned versus mouse genome v.mm10 (GRCm38.p5) using STAR ≥2.5.4b with the parameters '–outFilterMismatchNoverLmax 0.1–alignIntronMax 200000[42]. The number of reads aligning to genes was counted with featureCounts ≥1.6.0 from the Subread package[43]. Only reads mapping at least partially inside exons were admitted and aggregated per gene. Reads overlapping multiple genes or aligning to multiple regions were excluded. Differentially expressed genes were identified using DESeq2 v. ≥1.14.0 (ref. [44]). The annotation was enriched with UniProt data (release March 24, 2017) based on Ensembl gene identifiers (Activities at the Universal Protein Resource (UniProt)).

## Cell culture, plasmid transfection and Co-IP

HEK293, HEK293T and HeLa cells were grown in DMEM (Sigma) supplemented with 10% FCS (Sigma), 2 mM L-glutamine, 100 U penicillin and 100 μg ml⁻¹ ptreptomycin at 37 °C, 5% CO₂. HEK293T cells (2 × 10⁶ per 10 cm dish) were transfected with 8 μg Flag-HA-*TET3*-pEF (catalog no. 49446, Addgene) using calcium phosphate precipitation method. At 48 h after transfection, HEK293T cells were collected and washed twice with ice-cold PBS. Cells were resuspended in 300 μl lysis buffer (20 mM Tris-HCl pH 8.0, 200 mM NaCl, 1 mM EDTA, 1 mM EGTA, 1% Triton X-100) and sonicated with a bioruptor for 15 min. Lysates were supplemented with 500 μl lysis buffer and incubated on a rotating wheel at 4 °C for 30 min. Cell debris was removed by centrifugation at 12,000g for 20 min at 4 °C. Protein lysate (800 μg) protein lysate was incubated with HA or SETD2 antibody overnight at 4 °C followed by incubation with Protein A-agarose beads (Roche) antibodies at 4 °C for 4 h. After washing three times with lysis buffer, precipitated proteins were eluted from beads in 2× SDS loading buffer and analyzed by western blot.

## Chromatin immunoprecipitation

Chromatin immunoprecipitation (ChIP) was performed following published protocols[45]. Briefly, FACS-purified SMCs (300,000) were first cross-linked with 1% formaldehyde for 10 min and then quenched using the truChIP Chromatin Shearing Kit (COVARIS) for 10 min at RT. Chromatin was sheared to an average size of 200–500 bp by sonication (Diagnode Biorupter). Protein–DNA complexes were immunoprecipitated with IgG or antibodies listed in (Supplementary Table 3), followed by incubation with Protein A/G magnetic beads (Dynabeads, Invitrogen). For ChIP–qPCR, beads were washed and protein–DNA complexes were eluted and purified using 10% (w/v) chelex-100 (Bio-Rad Laboratories) in Tris-EDTA[46]. Immunoprecipitated chromatin was analyzed by qPCR using SYBR Green quantitative real-time analysis with primers listed in Supplementary Table 1. For Pol II pSer5 ChIP–seq, 3 × 10⁶ FACS-purified SMCs were treated with DRB (100 μmol) at 4 °C for 1 h. Sheared genomic DNA (50 μg) was subjected to immunoprecipitation with 4 μg Pol II Ser5 antibody according to established protocols. Protein–DNA complexes were eluted from beads by incubation with 50 μl elution buffer (10 mM Tris-HCl pH 7.4, 5 mM EDTA, 300 mM NaCl, 0.5% SDS) at RT for 5 min and treated with 1 μg DNase-free RNase (Roche) at 37 °C for 30 min. After incubation with 25 μg proteinase K (10 mg ml⁻¹), 1 μg glycogen at 37 °C for 2 h, samples were heated at 65 °C with constant shaking at 1,350 rpm overnight. DNA was purified with a PCR purification Kit (MinElute PCR Purification Kit).

## hMeDIP–qPCR

Genomic DNA (1 μg) was extracted from control and *Tet3[smKO:T]* SMCs by using the AllPrep DAN/RNA Micro Kit (Qiagen). hMeDIP was done following instructions provided with the hMeDIP kit (Diagenode). IgG antibodies were employed as a control. Input and hMeDIP products were used as templates for quantitative real-time PCR. Relative 5hmC enrichment was calculated as follows: %recovery (specific locus) =2^[(Ct(10%input) – 3.32) – Ct(hmeDNA-IP)] × 100%; enrichment = %recovery (specific locus)/%recovery (IgG). Primers are listed in Supplementary Table 1.

## ChIP–seq and data analysis

Purified ChIP DNA (0.5–10 ng) was used for TruSeq ChIP Library Preparation Kit (Illumina) with modifications. Briefly, libraries were size selected by SPRI-bead based approach after final PCR with 18 cycles: samples were first cleaned at a 1× bead:DNA ratio, followed by two-sided-bead cleanup step with a 0.6× bead:DNA ratio. Supernatant was transferred to a new tube and incubated with additional beads at a 0.2× bead:DNA ratio. Bound DNA samples were washed with 80% ethanol, dried and resuspended in Tris-EDTA buffer. Sequencing was performed on the NextSeq500 instrument (Illumina) using v.2 chemistry with 1× 75 bp single-end setup. Quality assessment was performed via FastQC (https://www.bioinformatics.babraham.ac.uk/projects/fastqc/) and reads where trimmed using Reaper[47]. Reads were further deduplicated using Picard v.2.17.10. The Macs2 peak caller v.2.1.0 was employed to accommodate for the range of peak widths as typically expected[48]. The minimum *Q* value was set to –1.5 and false discovery rate (FDR) was changed to 0.001. Peaks overlapping ENCODE blacklisted regions (known misassemblies, satellite repeats) were excluded. To determine thresholds for significant peaks per IP, data were inspected manually in Integrated Genome Viewer (IGV) v.2.3.52 (ref. [49]). For comparison of peaks in different samples, significant peaks were overlapped and unified to represent identical regions and recounted. Background-correction was performed to correct read counts from different regions (unified peaks, promoters, genes). Treatment and Input samples were normalized for sequencing depth, before subtracting reads of the input sample from reads of the respective treatment sample in windows of 50 nt length[50]. All windows with negative values (Input > Treatment) were set to zero[51]. Background-corrected counts for regions were calculated using BigWigAverageOverBed (UCSC Tools) and normalized with DESeq2 v.≥1.14.0 (ref. [52]). Peaks were annotated with the promoter (TSS ±5,000 nt) of genes closely located to the center of the peak based on reference data from GENCODE v.25. To permit comparative display of samples in IGV, raw BAM files were scaled with DESeq2 size factors based on all unified peaks using bedtools genomecov resulting in normalized BigWig files[53]. Finally, DESeq2 was used to identify significantly differentially modified peaks based on background-corrected read counts from recounted unified peak regions.

## Nano-5hmC-seal (Nano-seal)

Genomic DNA was isolated following standard protocols and suspended in nuclease-free water. The Illumina sequencing library was generated from 75 ng genomic DNA using the NxSeq UltraLow DNA Library Kit (Lucigen) following the manufacturer's instruction, but without PCR amplification. Glucosylation of half of the purified DNA library was performed in a 21 μl reaction containing 1× Thermo Epi Buffer, 100 μM N3-UDP-Glc and 2 μl T4 beta-glucosyltransferase (Thermo Fisher) at 37 °C for 1 h. After glucosylation, 2.1 μl Biotin-PEG4-DBCO (Click Chemistry Tools, 20 mM stock) was added directly to the reaction mixture and incubated at 37 °C for 2 h. Biotinylated DNA was purified by paramagnetic DNA binding beads (1.8× volume; Omega Bio-Tek) following standard procedures. Purified DNA was incubated with 5 μl C1 Streptavidin beads (Life Technologies) in B&W buffer (B&W buffer: 5 mM Tris pH 7.5, 0.5 mM EDTA and 1 M NaCl) for 40 min at RT with rotation,

according to the manufacturer's instructions. Beads were subsequently subjected to six 5-min washes with B&W buffer before elution in 40 μl water. Eluted enriched DNA libraries were PCR-amplified with index primers and sequenced on an Illumina NextSeq 2000. Trimmomatic v.0.39 was employed to trim reads below a mean of Q15 in a window of five nucleotides[41]. Only reads longer than 15 nucleotides were used for further analyses. Trimmed and filtered reads were aligned to the mouse genome v.mm10 (ensemble release 101) using STAR v.2.7.10a with the parameters '–outFilterMismatchNoverLmax 0.2 –outFilterMatchNmin 20 –alignIntronMax 1 –outFilterMultimapNmax 1' (ref. [42]), retaining only unique alignments and excluding reads of uncertain origin. Reads were further deduplicated using Picard v.2.18.16 (Picard: A set of tools (in Java) for working with next generation sequencing data in the BAM format) to mitigate PCR artefacts leading to multiple copies of the same original fragment. Reads aligning to the mitochondrial chromosome were removed.

## CAGE–seq

Total RNA was isolated using the miRNeasy micro Kit (Qiagen) combined with on-column DNase digestion (DNase-Free DNase Set, Qiagen) to avoid contamination of genomic DNA. CAGE library preparation, sequencing, mapping and motif discovery analysis were performed by DNAFORM (Life Science Research Center, Japan). In brief, RNA quality was assessed by Bioanalyzer (Agilent) to ensure a RIN (RNA integrity number) greater than 7.0, and A260/280 and 260/230 ratios greater than 1.7. First-strand cDNA was transcribed to the 5′ end of capped RNAs, attached to CAGE 'bar code' tags. Sequenced CAGE tags were mapped to the mouse mm10 genome using BWA software (v.0.5.9) and HISAT2 after discarding ribosomal or non-A/C/G/T base-containing RNAs. Mapped CAGE tags with mapping quality higher than ten were retained, separated by the strand and trimmed to the length of one nucleotide at the 5′ end as CAGE tag start sites (CTSSs)[54]. CTSS numbers at gene bodies were calculated by excluding exon 1 and by normalizing to total tags. To identify strand-specific TSS, only sense CTSS were used. Genes with 1.3-fold higher intragenic CTSS signal in *Tet3*-deficient cells compared to controls were selected. Average CTSS coverage at single-base nucleotide was around eight in control samples, based on which single nucleotide sites with CTSS less than eight were defined as low-expressed. For visualization, only CTSS with tag greater than eight were employed. Regions 50 bp upstream or downstream of TSSs specific to TET3 KO were used for motif enrichment analysis by HOMER[55].

## Laser capture microdissection and DNA-microarray analysis

Cyrosections (7–10 μm) mounted on glass microscope slides were successively immersed into the 70% ethanol fixative solution (10 s); dH$_2$O (10 s); Mayer's hematoxylin (45 s); ddH$_2$O (10 s); tap water 10 s; 70% ethanol (10 s); 95% ethanol (10 s); 95 % ethanol (10 s); 100% ethanol (60 s); 100% ethanol (60 s). Slides were air-dried before bronchi/bronchioles and vasculature were microdissected using the Laser Microbeam System (P.A.L.M.), and collected into a tube containing 200 μl RNA lysis buffer for RNA extraction using Rneasy Micro Kit (QIAGEN). RNA (0.8 ng) was used for DNA-microarray analysis using the GeneChip WT Pico Reagent Kit (P/N 703262 Rev.1); the GeneChip WT Pico Kit, P/N: 902622; the GeneChip Hybridization, Wash and Stain Kit P/N 900720 and the Mouse transcriptome array 1.0 ST (ClariomD, Ref: 520851) according to the Affymetrix protocol User Guide. DNA-microarray data were analyzed based on published protocols[56].

## RNA transfection

RNA from 1 million freshly sorted SMCs was extracted using the TRIzol reagent (Invitrogen). HeLa or HEK293 cells at 70% confluence were mock-transfected or transfected with 1 μg total RNA using Lipofectamin MessengerMAX (LipoMAX) (Thermo Fisher) according to the manufacturer's instructions. At 18 h after transfection, cells were collected and RNA was extracted for gene expression analysis.

## Proximity ligation assay

Cryosections of lung tissues or FACS-sorted SMCs after 6 days cultivation with SmBM Smooth Muscle Cell Growth Basal Medium (LONZA) in 6-well plates (around 200,000 SMCs per well) were fixed with 4% PFA for 10 min, permeabilized with 0.3% Triton-100X in PBS for 15 min and washed twice with PBS. PLAs were performed following the Duolink PLA Fluorescence Protocol (Merck) and antibodies listed in Supplementary Table 3. Tissue sections and cells were mounted using Duolink In situ Mounting Media with DAPI (Sigma). Image acquisition were performed by confocal confocal microscopy using a Leica SP8.

## Statistical analysis

For all quantitative analyses, a minimum of two biological replicates were analyzed. For the usage of statistical tests, it was assumed that sample data are derived from a population following a probability distribution based on a fixed set of parameters; $t$-tests were used to determine the statistical significance of differences between two groups. For multiple comparisons, one-way analysis of variance (ANOVA) with Tukey's post hoc test for correction of multiple testing was performed. The following values were considered to be statistically significant: $*P < 0.05$; $**P < 0.01$; $P*** < 0.001$; $P**** < 0.0001$; NS, not significant. Calculations were done using the GraphPad Prism 9 software and R v.4.1.0. Data are represented as mean ± s.e.m. unless indicated otherwise. The boxplot displays the median with 25% (bottom value) and 75% quantiles (top value) unless indicated otherwise. No statistical method was used to predetermine sample size.

## Reporting summary

Further information on research design is available in the Nature Portfolio Reporting Summary linked to this article.

## Data availability

Data have been deposited in public databases. RNA-seq data are available at https://www.ncbi.nlm.nih.gov/geo/query/acc.cgi?acc=GSE166816, Pol II pSer5 ChIP–seq data at https://www.ncbi.nlm.nih.gov/geo/query/acc.cgi?acc=GSE166815, CAGE–seq data at https://www.ncbi.nlm.nih.gov/geo/query/acc.cgi?acc=GSE168206, Nano-seal data at https://www.ncbi.nlm.nih.gov/geo/query/acc.cgi?acc=GSE202201 and H3K36me3 ChIP–seq data at https://www.ncbi.nlm.nih.gov/geo/query/acc.cgi?acc=GSE201924. Microarray data were deposited at www.ebi.ac.uk/arrayexpress/ under the accession number E-MTAB-10144. Source data are provided with this paper.

## Code availability

All scripts used to analyze datasets described in this study are available at https://github.com/loosolab/Wu_et_al_2022_Spurious_transcription/blob/main/README.md (https://doi.org/10.5281/zenodo.7248507). Published software were used for all calculations and visualizations. Separate folders are provided, according to the data type. Commands in *base_processing.sh should be performed first, followed by those in *downstream_processing.sh, if necessary. Remaining scripts contain helper functions. Further information is provided in Methods and Reporting summary.

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

## Acknowledgements

We thank K. Richter, G. Kripp, S. Krüger, S. Thomas and S. Kreutzer for technical help and Y. Zhang for providing reagents for Nano-seal. This work was supported by the Deutsche Forschungsgemeinschaft (DFG) SFB-TRR 267 TP A05 to T. Braun and T. Boettger; SFB-TRR81 A02 to X.Y. and T. Braun; SFB1213-Project-ID 268555672-TP A02 to T. Braun and -TP B02 to X.Y. and T. Braun; SFB1531-TP B08 to T. Braun; the LOEWE project iCANx to T. Braun. The BioMaterialBank Nord, member of the Popgen 2.0 Network (P2N), is supported by the German Center for Lung Research, funded by the German Ministry for Education and Research (01ER1103). Human samples were provided by the UGMLC

Giessen Biobank, member of the DZL Platform Biobanking, which also is supported by DFG grant SFB1213-Project-ID 268555672-TP CP01. The funders had no role in study design, data collection and analysis, or decisions to prepare or publish the manuscript.

## Author contributions

X.Y. and T. Braun conceived and designed experiments. F.W. performed most of the experiments, analyzed the data and prepared figures. D.D. performed Nano-seal. S.G. performed next-generation deep sequencing. M.L., S.G. and C.K. performed bioinformatics analysis of RNA-seq and ChIP–seq. H.L. analyzed CAGE–seq and Nano-seal data. T. Boettger and N.W. contributed to microarray analysis. S.L., S.O. and M.P. provided transgenic mouse lines. S.K. and H.R. provided the mouse asthma models and N.W. the chronic hypoxia mouse model. Y.Z., X.L., A.A. and U.G. participated in data analysis and discussions and provided advice. T. Braun, X.Y. and F.W. wrote the manuscript.

## Funding

## Competing interests

The authors declare no competing interests.

## Additional information

**Extended data** is available for this paper at https://doi.org/10.1038/s41588-022-01252-3.

**Correspondence and requests for materials** should be addressed to Xuejun Yuan or Thomas Braun.

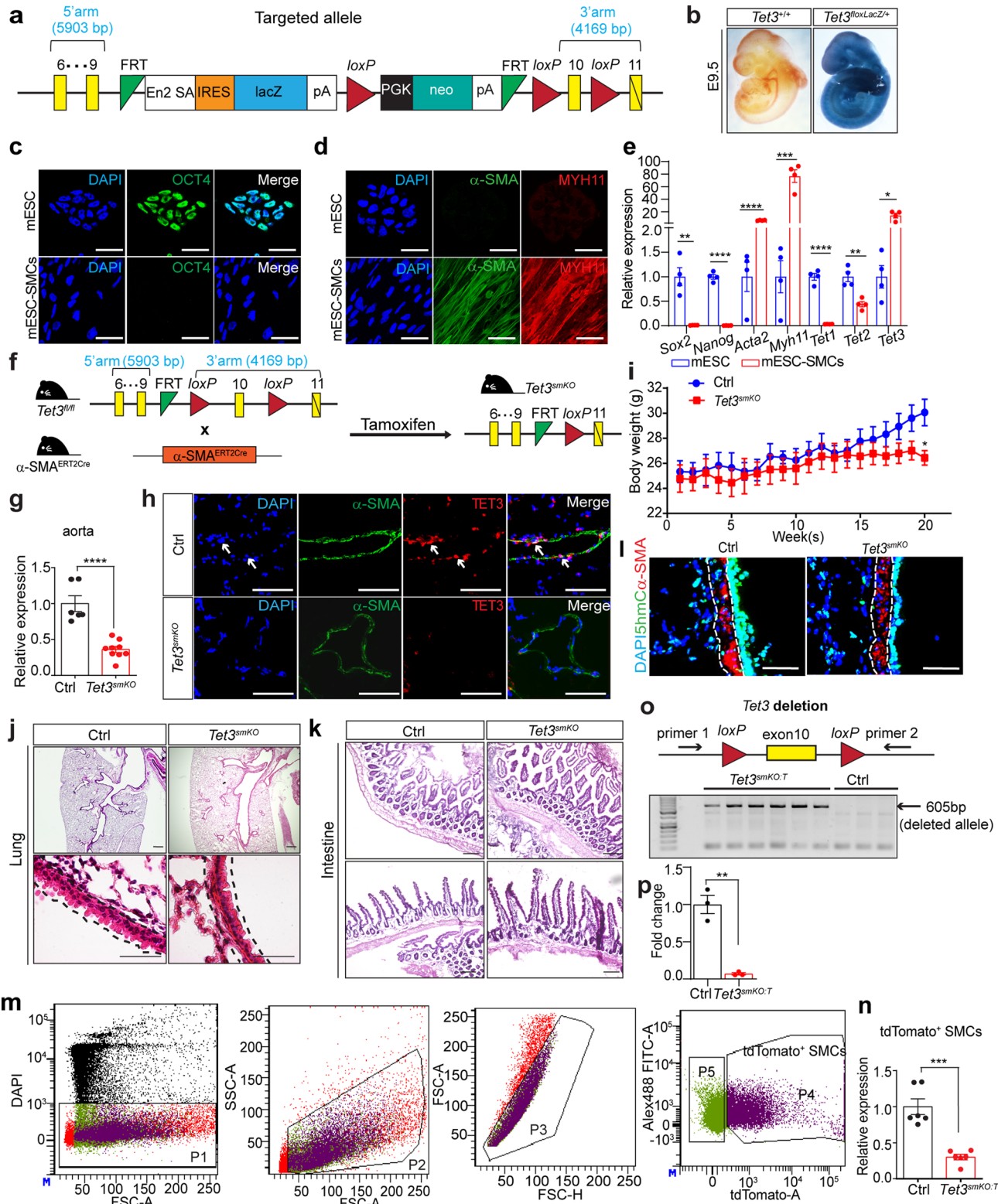

**Extended Data Fig. 1 | See next page for caption.**

**Extended Data Fig. 1 | SMC-specific inactivation of *Tet3* induces pathological changes in airways. a**, Schematic depiction of the conditional *Tet3* allele. **b**, Whole-mount LacZ staining of *Tet3*⁺/⁺, *Tet3*^floxLacZ/+ embryos at E9.5 (n = 10). **c**, Immunofluorescence analysis of OCT4 in mESC and mESC derived SMCs (mESC-SMCs) (n = 3). Scale bar: 50 μm. DNA was stained by DAPI. **d**, Immunofluorescence analysis of α-SMA and MYH11 in mESC and mESC-SMCs (n = 3). Scale bar: 50 μm. DNA was stained by DAPI. **e**, RT-qPCR analysis of *Sox2*, *Nanog*, *Acta2*, *Myh11*, *Tet1*, *Tet2*, *Tet3* in mESC and mESC-SMCs (n = 4 independent experiments; two-tailed unpaired t-test: *$P$ < 0.05; **$P$ < 0.01; ***$P$ < 0.001; ****$P$ < 0.0001). **f**, Outline of the strategy to generate inducible SMC-specific *Tet3* knockout mice (mice were 8 weeks old at first tamoxifen injection). **g**, RT-qPCR analysis of *Tet3* expression in aortas from control and *Tet3*^smKO mice, 8 weeks after tamoxifen injection. *β-actin* was used for normalization (control n = 6; *Tet3*^smKO n = 9; two-tailed unpaired t-test: ****$P$ < 0.0001). **h**, Immunofluorescence staining for α-SMA and TET3 on cryosections from control and *Tet3*^smKO lungs, 8 weeks after tamoxifen injection (n = 3). DNA was stained by DAPI. Scale bar: 50 μm. **i**, Body weights of

control and *Tet3*^smKO mice at different time points after tamoxifen injection (n = 6 independent animals). Tamoxifen injections started when mice were 8-weeks-old. **j**, **k**, H&E staining of cryosections from lungs (**j**) and intestines (**k**) of control and *Tet3*^smKO mice 8 weeks after tamoxifen injection to demonstrate normal tissue architecture (n = 3). Scale bar: 50 μm. **l**, Immunofluorescence staining for α-SMA and 5hmC on cryosections from control and *Tet3*^smKO lungs, 8 weeks after tamoxifen injection (n = 3). DNA was stained by DAPI. Scale bar: 50 μm. **m**, FACS gating strategy to isolate SMCs from control and *Tet3*^smKO:T lungs. **n**, RT-qPCR analysis of *Tet3* expression in sorted SMCs from control and *Tet3*^smKO:T mice, 8 weeks after tamoxifen injection (n = 6 independent animals; two-tailed unpaired t-test: ***$P$ = 0.0001). *β-actin* was used for normalization. **o**, PCR of genomic DNA using primers located at 5′ upstream and 3′-downsteam of floxed exon 10 of the *Tet3* gene. Genotyping was performed with 5 litters of animals. **p**, Quantification of the Western blot analysis of TET3 in sorted lung control and *Tet3*^smKO:T SMCs (n = 3 independent animals; two-tailed unpaired t-test: **$P$ < 0.01). Data (in e, g, i, n, p) are presented as mean values ± s.e.m.

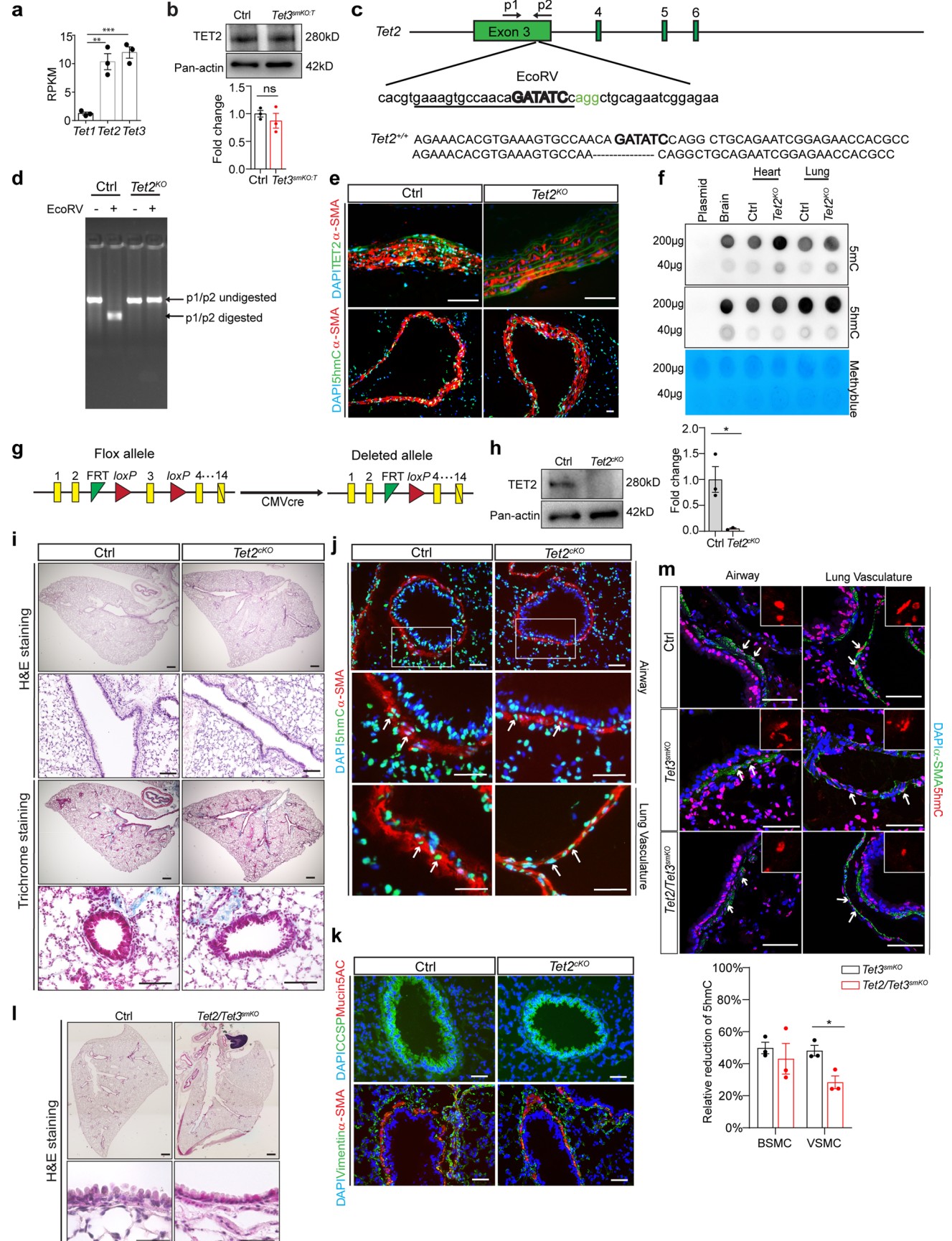

**Extended Data Fig. 2 | See next page for caption.**

**Extended Data Fig. 2 | Inactivation of *Tet2* has no impact on 5hmC levels and tissue morphology. a**, Expression of *Tet1*, *Tet2* and *Tet3* genes by RNA-seq in sorted control lung SMCs (n = 3 independent animals; one-way ANOVA with Tukey's post hoc test: \*\*$P$ = 0.0017; \*\*\*$P$ = 0.0007). **b**, Western blot analysis of TET2 in sorted lung SMCs (n = 3 independent animals; two-tailed unpaired t-test). The pan-actin loading control was run on a different gel due to differences in molecular weights. **c**, Localization of the sgRNA-targeting site in the *Tet2* gene. The sgRNA-targeting sequence is underlined, the protospacer-adjacent motif (PAM) sequence is labeled in green. The restriction site at the target region is in bold and capitalized. PCR primers (p1, p2) used for genotyping are indicated. **d**, Genotyping of *Tet2*$^{+/+}$ (Ctrl) and *Tet2*$^{-/-}$ (*Tet2*$^{KO}$) mice with or without EcoRV digestion. Genotyping was performed with 5 litters of animals. **e**, Immunofluorescence staining for TET2 & α-SMA and 5hmC & α-SMA on cryosections of 8-weeks-old *Tet2*$^{+/+}$ (Ctrl) and *Tet2*$^{KO}$ aortas (n = 3). DNA was stained by DAPI. Scale bar: 50 μm. **f**, Dot blot analysis of 5mC and 5hmC levels using genomic DNA from 8-weeks-old *Tet2*$^{+/+}$ (Ctrl) and *Tet2*$^{KO}$ hearts and lungs (n = 3). Plasmid and brain tissue DNA from *Tet2*$^{+/+}$ (Ctrl) mice were used as negative and positive controls, respectively. Methylene blue (MB) staining served as loading control. **g**, Schematic depiction of the excision of floxed exon 3 in the *Tet2* gene by CMV-Cre. **h**, Western blot analysis of TET2 in lung tissues (n = 3 independent animals; two-tailed unpaired t-test: \*$P$ = 0.0189). **i**, H&E staining (upper panel) and trichrome staining (lower panel) of paraffin sections from 8-weeks-old *Tet2*$^{fl/fl}$ (Ctrl) and *CMV-Cre*$^{pos}$*Tet2*$^{fl/fl}$ (*Tet2*$^{cKO}$) lungs (n = 3). Scale bar: 200μm, 50 μm. **j, k**, Immunofluorescence staining for 5hmC & α-SMA (h), CCSP & Mucin5AC and Vimentin & α-SMA on cryosections from 8-weeks-old *Tet2*$^{fl/fl}$ (Ctrl) and *Tet2*$^{cKO}$ lungs (n = 3). DNA was stained with DAPI. Arrows indicate 5hmC-positive SMCs. Scale bar: 50 μm. **l**, H&E staining of paraffin sections from control or *Tet2/Tet3*$^{smKO}$ lungs (n = 5). Scale bar: 200μm, 50 μm. **m**, Immunofluorescence staining for 5hmC and α-SMA on cryosections from *Tet2/Tet3*$^{fl/fl}$ (Ctrl), *Tet3*$^{smKO}$, *Tet2/Tet3*$^{smKO}$ lungs (n = 3 independent animals). DNA was stained with DAPI. Arrows indicate 5hmC-positive SMCs. Inserts depict enlarged images of 5hmC-stained SMCs from the respective panels. Mean fluorescence intensity (MFI) of 5hmC signals was quantified by Image J and is shown in the lower panel (n = 3; two-tailed unpaired t-test: \*$P$ = 0.0187). Scale bar: 50 μm. The enlarged image was scaled to 150×150 pixels. Data in (a, b, h, m) are presented as mean values ± s.e.m.

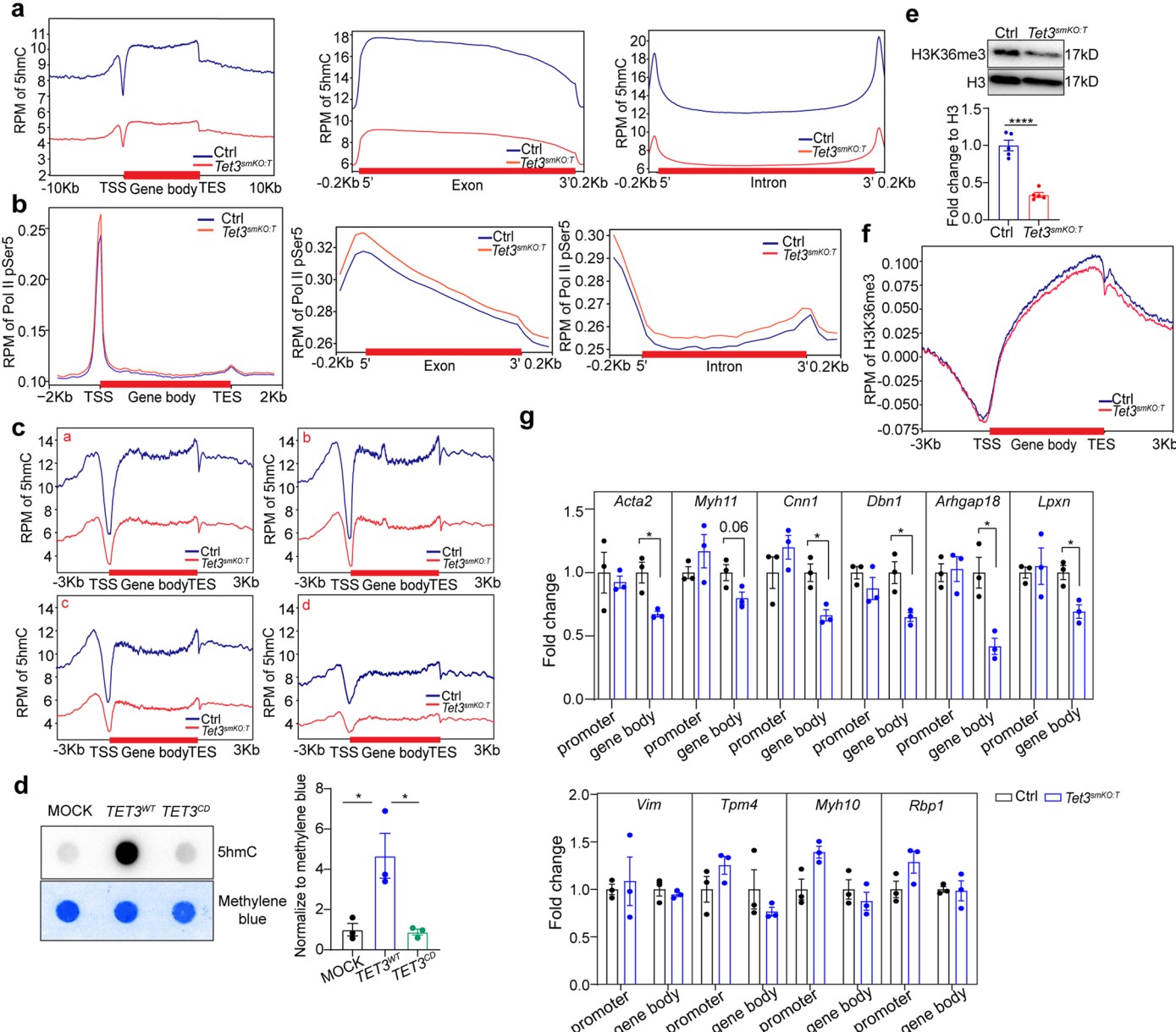

**Extended Data Fig. 3 | Loss of *Tet3* in SMCs leads to reduction of 5hmC and aberrant intragenic entry of Pol II. a,** Distribution of 5hmC analyzed by Nano-seal-seq within gene body (intragenic), promoter, exon, and intron regions in SMCs sorted from control and *Tet3smKO:T* lungs 8 weeks after tamoxifen injection (n = 2). **b,** Distribution of pSer5 Pol II at gene body, exon, and intron regions in sorted SMCs from control and *Tet3smKO:T* lungs after DRB treatment (n = 2). **c,** Distribution of 5hmC in subgroups of genes defined in Fig. 2b (n = 2). **d,** Dot blot analysis of 5hmC levels in HEK293T cells after transduction with mock, HA-tagged *TET3WT* or *TET3CD* overexpressing lentiviruses (n = 3 independent experiments). Quantification of 5hmC levels is shown in the lower panel (n = 3; one-way ANOVA with Tukey's post hoc test: *P = 0.0194, *P = 0.0170). **e,** Western blot analysis of H3K36me3 in sorted SMCs (n = 5 independent animals; two-tailed unpaired t-test: ****P = <0.0001). **f,** Distribution of H3K36me3 ChIP–seq signals within gene body (intragenic) in SMCs sorted from control and *Tet3smKO:T* lungs, 8 weeks after tamoxifen injection (n = 2). **g,** ChIP–qPCR analysis of H3K36me3 enrichment within promoters and exons of selected genes in SMCs isolated from control and *Tet3smKO:T* lungs 8 weeks after tamoxifen injection (n = 3 independent animals; two-tailed unpaired t-test: *P = 0.0181, *P = 0.0148, *P = 0.0194, *P = 0.0134, *P = 0.0161). ChIP for H3 was used for normalization. Data in (d, e, g) are presented as mean values ± s.e.m.

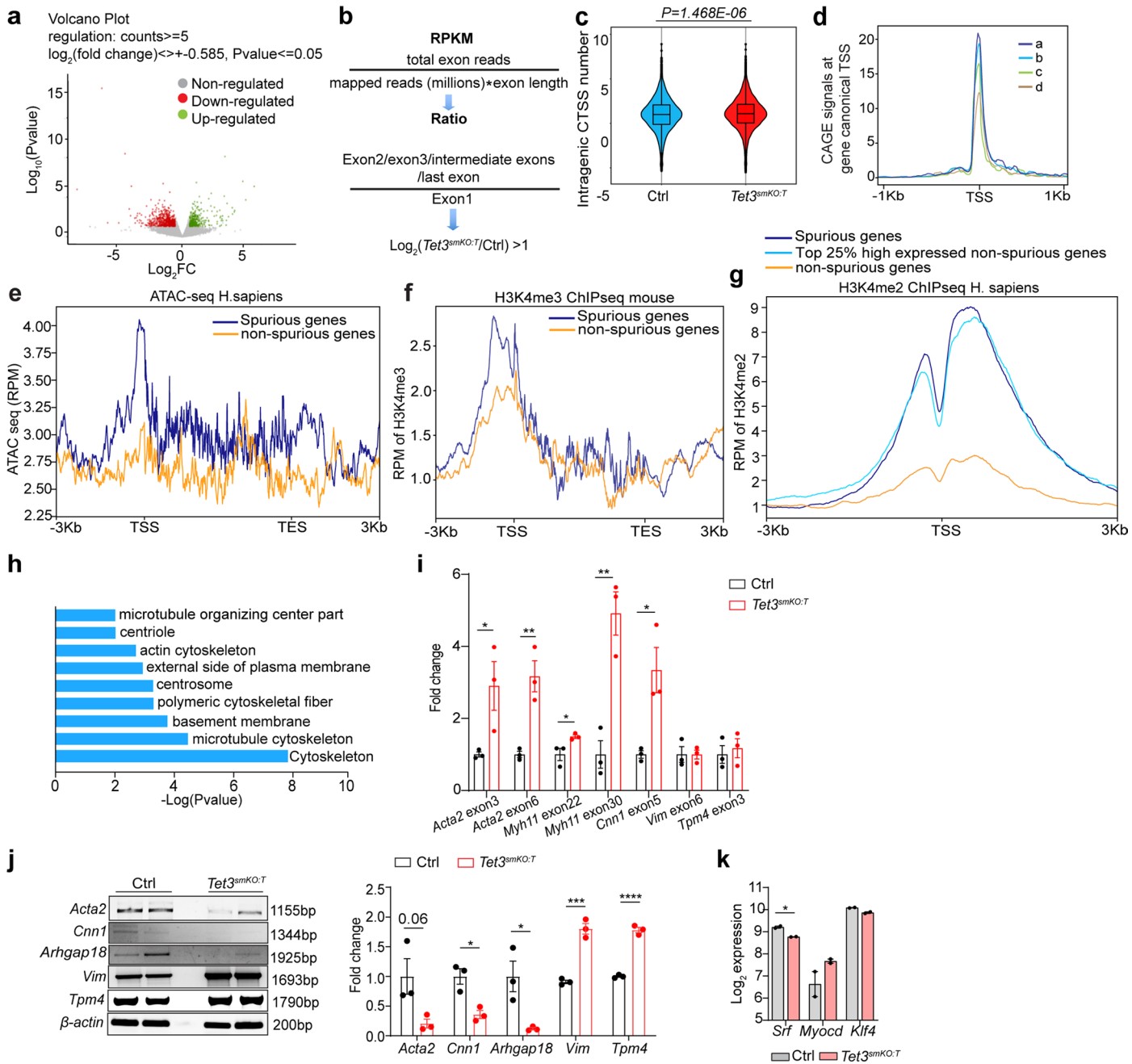

**Extended Data Fig. 4 | Aberrant initiation of intragenic transcription after *Tet3* inactivation occurs in highly transcribed SMC genes. a**, Volcano plot based on RNA-seq of sorted SMCs from control and *Tet3^smKO:T^* lungs, 8 weeks after tamoxifen injection (n = 2 independent animals) showing deregulated genes (fold change log₂ > 0.585, two sided, Wald-test with Benjamini-Hochberg correction: *P* ≤ 0.05). **b**, Formulas for calculating the ratio of normalized RNA-seq RPKM of downstream exon versus first exon between control and *Tet3^smKO:T^* SMCs. **c**, Violin plot showing CTSS numbers at gene bodies (without exon1) in SMCs from control and *Tet3^smKO:T^* lungs (n = 2 independent animals). The *P* value was calculated using the likelihood-ratio test. The violin plot displays the median with 25% and 75% quartiles. **d**, Distribution of CAGE-signals at canonical TSSs in different quartiles (a–d) defined in (Fig. 2b), based on differential intragenic binding signals of Pol II pSer5. **e**, DNA accessibility of spurious and nonspurious genes based on ATAC-seq data in human SMCs (GSM4558338). **f**, Profile of H3K4me3 enrichment (GSM112417) within gene body regions of 515 spuriously expressed genes (defined in Fig. 3a) in SMCs and randomly selected 515

nonspuriously expressed genes. **g**, Density profile of H3K4me2 enrichment over annotated TSS (GSE96375) of 515 spuriously expressed genes, randomly selected 515 nonspuriously expressed genes, and 515 randomly selected highly expressed genes. **h**, Gene Ontology enrichment analysis of spuriously transcribed genes shown in Fig. 3a (n = 2 independent animals). *P* value was calculated with two-sided, Fisher's exact test. **i**, RT-qPCR analysis of indicated genes in lung SMCs from control and *Tet3^smKO:T^* lungs (n = 3 independent animals; two-tailed unpaired t-test:*P* = 0.0495,; **P* = 0.008, *P* = 0.0491, **P* = 0.0053,*P* = 0.0213). **j**, Semi-quantitative RT-PCR analysis of *Acta2, Cnn1, Arhgap18, Vim, Tpm4* full-length mRNA in SMCs from control and *Tet3^smKO:T^* lungs (*Acta2, Cnn1, Arhgap18, Tpm4*: n = 3; *Vim*: n = 5) two-tailed unpaired t-test:*P* = 0.0121, **P* = 0.0272;***P* = 0.0007; ****P* < 0.0001). *β-actin* was used for normalization. Quantification of expression levels was performed by Image J and is shown on the right. **k**, Expression of *Srf, Myocd, Klf4* genes based on RNA-seq in sorted control and *Tet3^smKO:T^* lung SMCs (n = 2 independent animals; two-tailed unpaired t-test:*P* = 0.0118). Data in (i-k) are presented as mean values ± s.e.m.

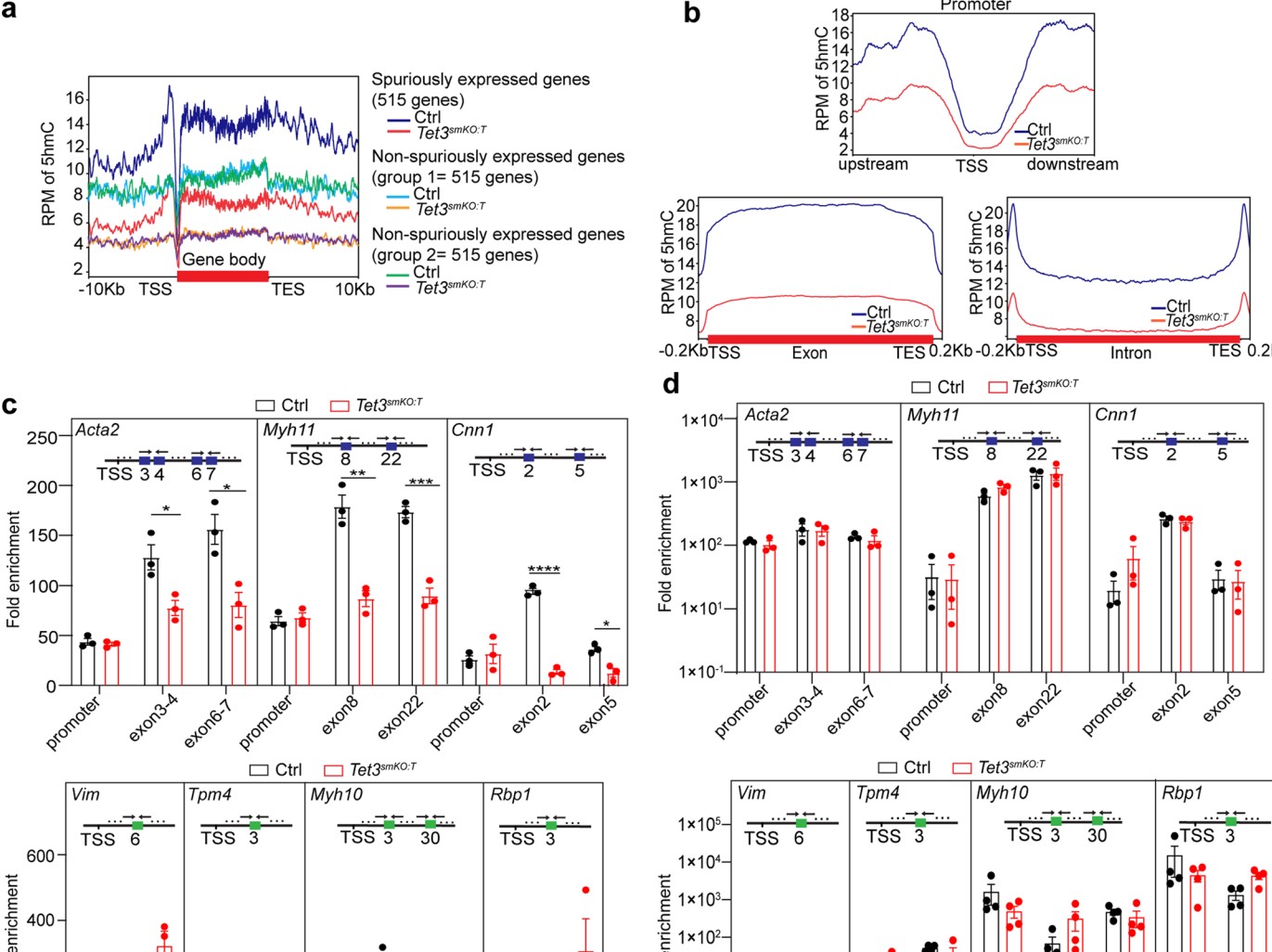

**Extended Data Fig. 5 | Inactivation of *Tet3* reduces genome-wide 5hmC levels, particularly within gene bodies. a**, Distribution of Nano-seal-seq signals within gene bodies of spuriously expressed genes (defined in Fig. 3a) and randomly selected nonspuriously expressed genes (gene Number=515) (n = 2). **b**, Distribution of 5hmC signals assessed by Nano-seal-seq within promoter, exon and intron regions of spuriously expressed genes (defined in Fig. 3a) in SMCs from control and *Tet3^{smKO:T}* lungs (n = 2). **c**, hMeDIP–qPCR analysis of 5hmC enrichment at promoters and exons of indicated genes in SMCs from control and *Tet3^{smKO:T}* lungs relative to IgG-IP, 8 weeks after tamoxifen injection (n = 3 independent animals; two-tailed unpaired t-test: *$P < 0.05$; **$P < 0.01$; ***$P < 0.001$; ****$P < 0.0001$). **d**, MeDIP qPCR analysis of 5mC enrichment at promoters and exons of indicated genes in SMCs from control and *Tet3^{smKO:T}* lungs relative to IgG-IP, 8 weeks after tamoxifen injection (*Acta2*, *Myh11*, *Cnn1*: n = 3; *Vim*, *Tpm4*, *Myh10*, *Rbp1*: n = 4). Data (c, d) are presented as mean values ± s.e.m.

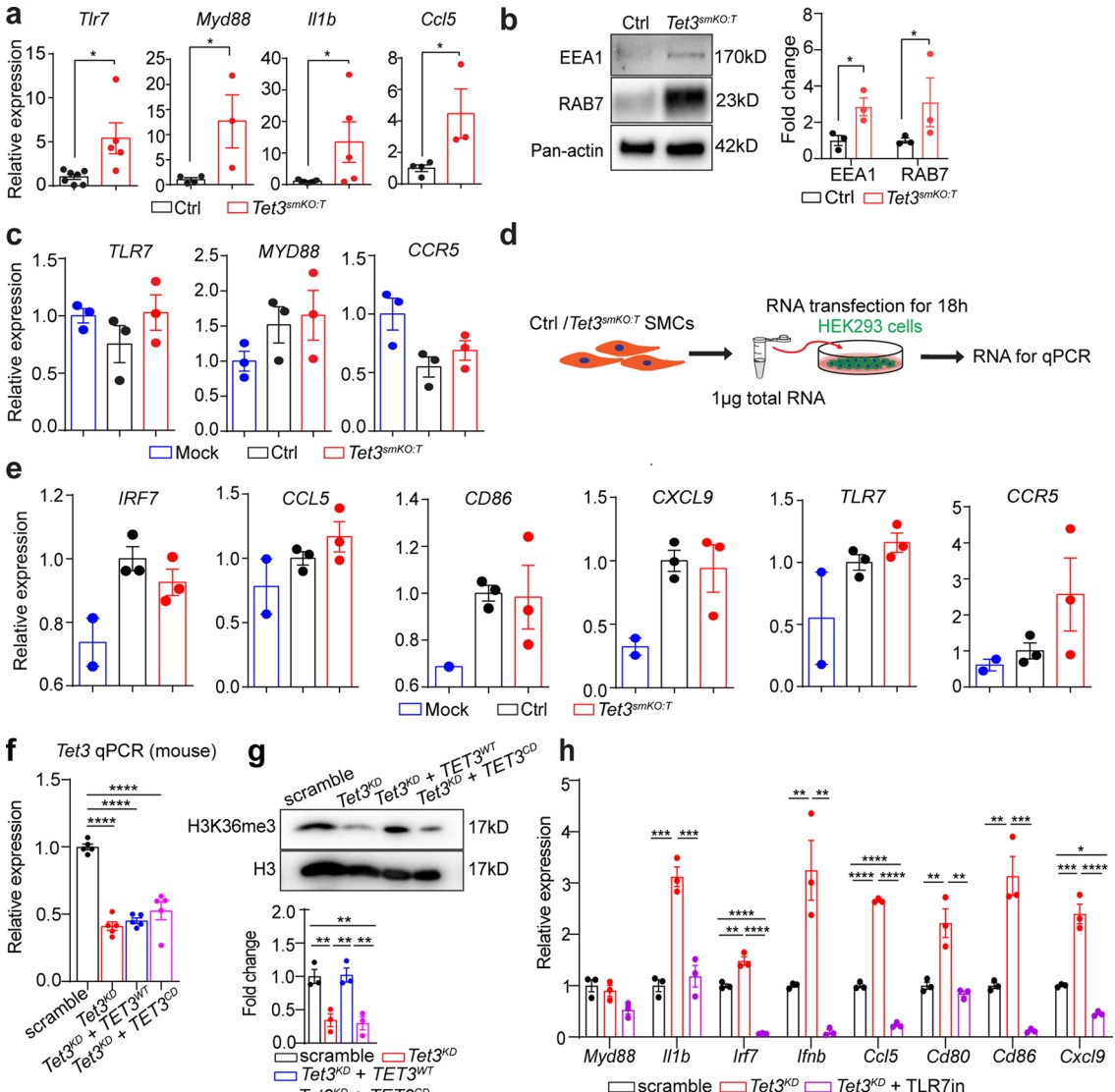

**Extended Data Fig. 6 | Spurious transcripts activate endosomal TLR signaling in SMCs. a**, RT-qPCR analysis to monitor expression of *Tlr7* (control n = 7; *Tet3smKO:T* n = 5), *Myd88* (control n = 4; *Tet3smKO:T* n = 3), *Il1b* (control n = 7; *Tet3smKO:T* n = 5), *Ccl5* (control n = 4; *Tet3smKO:T* n = 3) in SMCs from control and *Tet3smKO:T* lungs, 8 weeks after tamoxifen injection (two-tailed unpaired t-test:*P = 0.0149, *P = 0.0482, *P = 0.0414,*P = 0.0484). **b**, Western blot analysis of EEA1 and RAB7 in sorted SMCs (n = 3 independent animals; two-tailed unpaired t-test:*P = 0.0398, *P = 0.0449). Pan-actin was used as loading control. **c**, RT-qPCR analysis of HeLa cells mock-transfected and transfected with RNA isolated from control and *Tet3smKO:T* SMCs (n = 3 independent experiments; one-way ANOVA with Tukey's post hoc test:*P > 0.05). **d**, Experimental strategy for RNA transfection experiments. **e**, RT-qPCR analysis of HEK293 cells mock-transfected (Mock) and transfected with total RNAs from control (Ctrl) or *Tet3smKO:T* lung SMCs, 8 weeks

after tamoxifen injection (n = 3 independent experiments; one-way ANOVA with Tukey's post hoc test: *P > 0.05*). **f**, RT-qPCR analysis of mouse *Tet3* in mESC-SMCs after transduction with scramble, *Tet3KD*, *Tet3KD* + *TET3WT*, *Tet3KD* + *TET3CD* lentivirus (n = 5 independent experiments; one-way ANOVA with Tukey's post hoc test: ****P < 0.0001). **g**, Western blot analysis of H3K36me3 in mESC-SMCs after transduction with scramble, *Tet3KD*, *Tet3KD* + *TET3WT*, *Tet3KD* + *TET3CD* lentiviruses (n = 3 independent experiments; one-way ANOVA with Tukey's post hoc test: **P < 0.01). H3 was used as loading control. **h**, RT-qPCR analysis of indicated genes in mESC-SMCs after transduction with scramble or *Tet3KD* lentiviruses with or without the TLR7 inhibitor (TLR7in) E6446 (n = 3 independent experiments; one-way ANOVA with Tukey's post hoc test: *P < 0.05; **P < 0.01; ***P < 0.001; ****P < 0.0001). Data (a–c, e-h) are presented as mean values ± s.e.m.

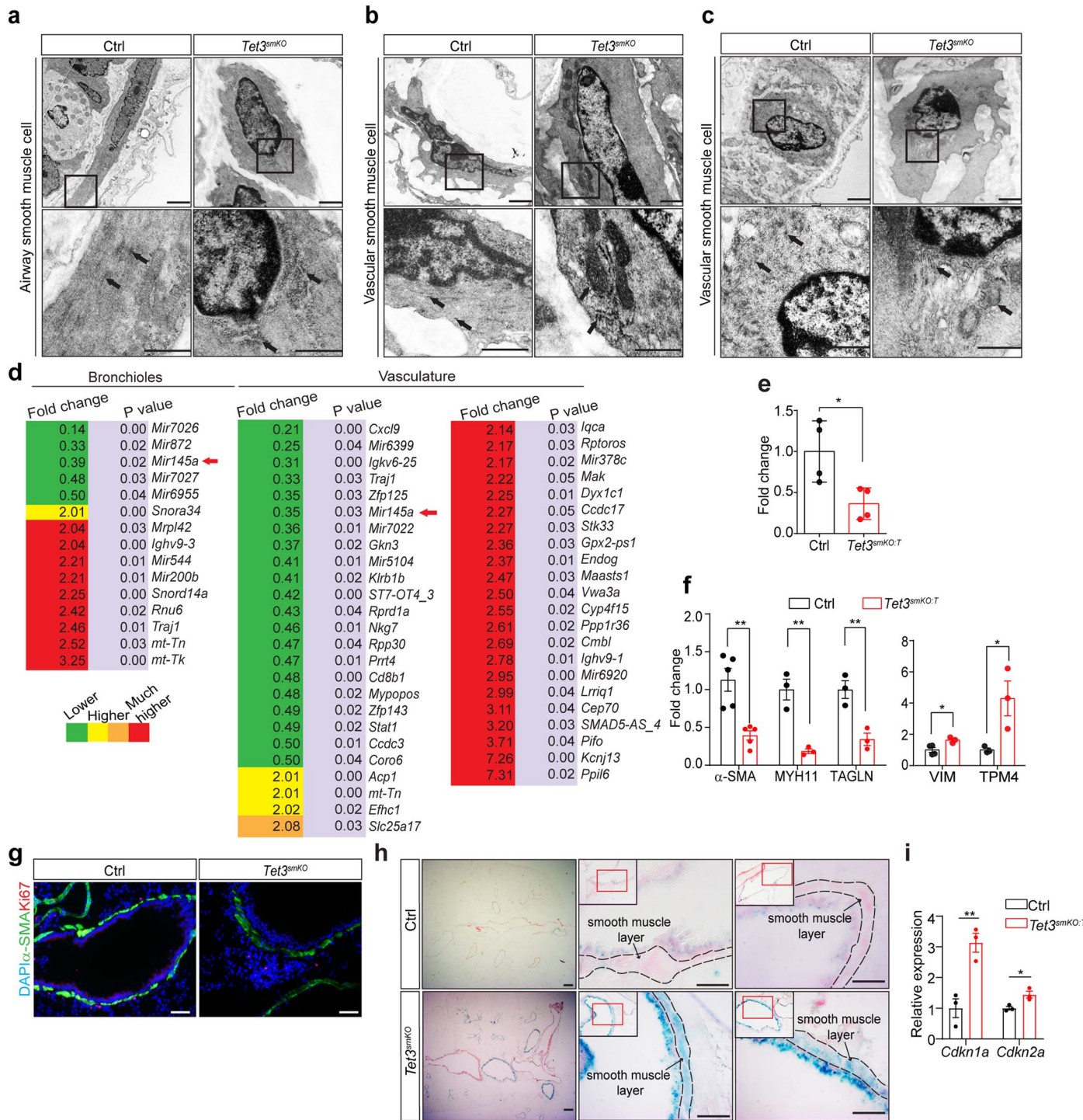

**Extended Data Fig. 7 | Reduced formation of 5hmC causes acquisition of a synthetic phenotype in lung SMCs. a–c**, EM images of bronchial SMCs, 6 months after tamoxifen injection (**a**) and of lung vascular SMCs 8 weeks (**b**) or 6 months (**c**) after tamoxifen injection in control and *Tet3^smKO* mice (n = 4). Arrows point to dense bodies (control) and rough ER (*Tet3^smKO*). Scale bar: 1000 nm. **d**, Microarray expression analysis of laser capture micro-dissected bronchiolar (left panel) or vasculature (middle and right panels) regions from control and *Tet3^smKO* lungs, 8 weeks after tamoxifen injection (n = 3; two-sided moderated t-test with Benjamini-Hochberg correction for multiple test). Fold change refer to the *Tet3^smKO*/control. **e**, RT-qPCR analysis of miR-145 expression in SMCs from control and *Tet3^smKO:T* lungs (n = 4 independent animals; two-tailed unpaired t-test: *P = 0.0235). **f**, Quantification of protein levels of α-SMA (n = 5),

MYH11 (n = 3), TAGLN (n = 3), VIM (control n = 4, *Tet3^smKO:T* n = 3), TPM4 (n = 3) in SMCs from control and *Tet3^smKO:T* lungs (two-tailed unpaired t-test: **P = 0.002, **P = 0.0045, **P = 0.0097, *P = 0.0283, *P = 0.0426). **g**, Immunofluorescence staining for α-SMA and Ki67 on cryosections from control and *Tet3^smKO* lungs, 8 weeks after tamoxifen injection (n = 3). DNA was stained by DAPI. Scale bar: 50 μm. **h**, Senescence-associated β-galactosidase staining of cryosections from control and *Tet3^smKO* lungs, 8 weeks after tamoxifen injection (n = 3). Scale bar: 50 μm. **i**, RT-qPCR analysis of *Cdkn1a* (*p21*), *Cdkn2a* (*p16*) in SMCs isolated from control and *Tet3^smKO:T* lungs 8 weeks after tamoxifen injection (n = 3 independent animals; two-tailed unpaired t-test: **P = 0.008, *P = 0.0246). Data in (e, f, i) are presented as mean values ± s.e.m.

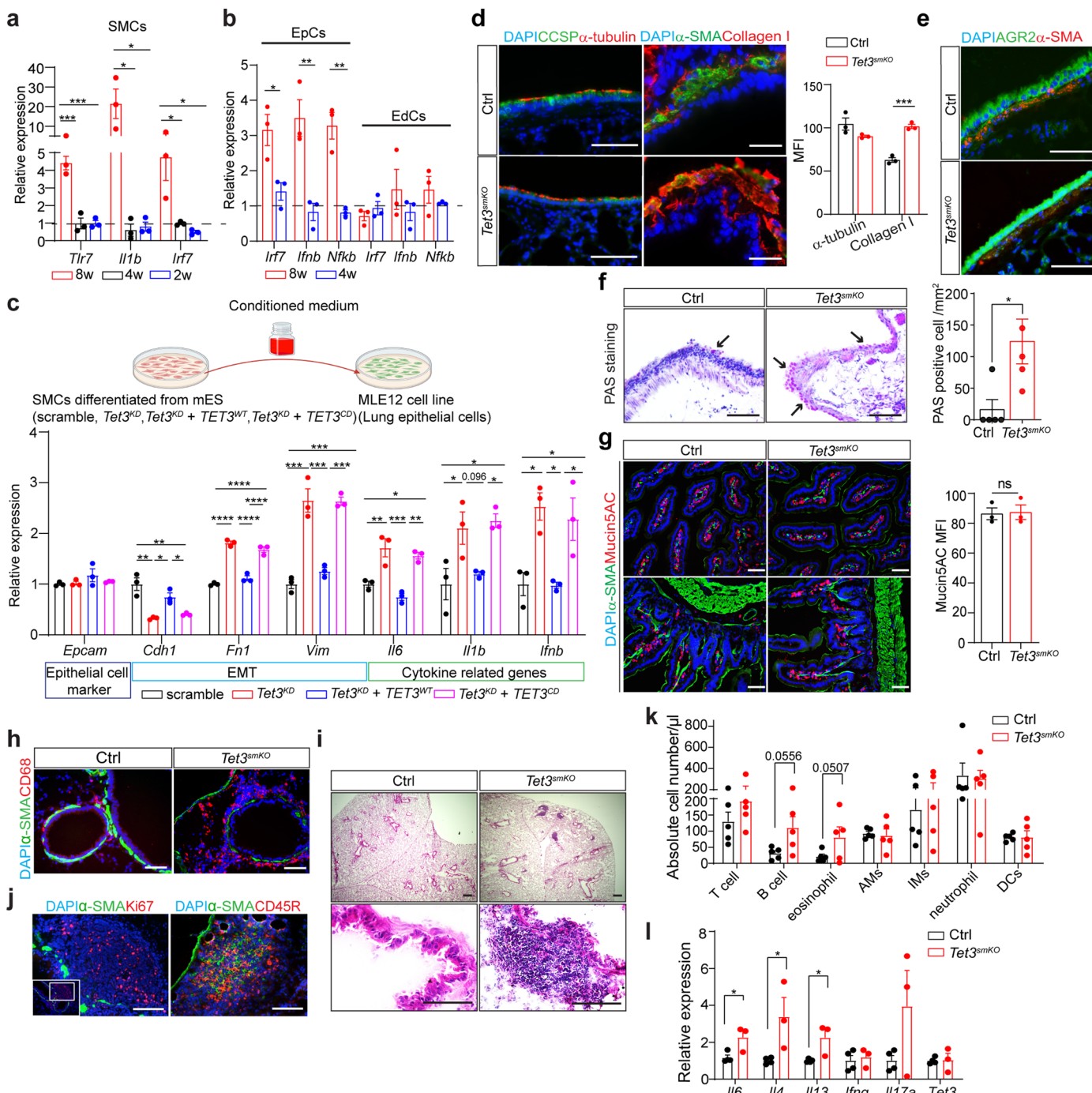

**Extended Data Fig. 8 | See next page for caption.**

**Extended Data Fig. 8 | Persistent reduction of 5hmC in SMCs causes a Th2 cell-based immune response in the lung. a**, RT-qPCR analysis of *Tlr7*, *Il1b*, *Irf7* in SMCs from *Tet3*^smKO:T^ lungs at different time points after tamoxifen injection. Relative expression levels are presented as fold-changes of *Tet3*^smKO:T^ versus control SMCs with the dashed horizontal line representing no change in expression (n = 3 independent animals; one-way ANOVA with Tukey's post hoc test: *P < 0.05; ***P < 0.001). **b**, RT-qPCR analysis of *Irf7*, *Ifnb*, *Nfkb* in epithelial cells (EpCs) and endothelial cells (EdCs) from *Tet3*^smKO:T^ lungs. Relative expression levels are presented as fold-changes of *Tet3*^smKO:T^ versus control SMCs with the dashed horizontal line representing no change in expression (n = 3 independent animals; two-tailed unpaired t-test: *P < 0.05; **P < 0.01). **c**, Upper panel: MLE12 cells were cultured with conditioned medium from mESC-SMCs after transduction with scramble, *Tet3*^KD^, *Tet3*^KD^ + TET3^WT^, *Tet3*^KD^ + TET3^CD^ lentiviruses. Lower panel: RT-qPCR analysis of *Epcam*, *Cdh1*, *Fn1*, *Vim*, *Il6*, *Il1b*, *Ifnb* in MLE12 cells, 1 day after co-culture (n = 3 independent experiments; one-way ANOVA with Tukey's post hoc test: *P < 0.05; **P < 0.01; ***P < 0.001; ****P < 0.0001). **d**, Immunofluorescence staining for CCSP and α−tubulin (upper panel), α-SMA and Collagen I (lower panel) on sections from control and *Tet3*^smKO:T^ lungs, 8 weeks after tamoxifen injection (n = 3 independent animals). Scale bar: 50 μm. DNA was stained by DAPI. MFIs of α−tubulin and Collagen I were quantified by Image J and are shown on the right (n = 3; two-tailed unpaired t-test: ***P = 0.0005). **e**, Immunofluorescence staining for AGR2 and α-SMA on sections from control

and *Tet3*^smKO^ lungs, 8 weeks after tamoxifen injection (n = 3). Scale bar: 50 μm. DNA was stained by DAPI. **f**, Periodic Acid-Schiff staining (PAS) of cryosections from control and *Tet3*^smKO^ lungs. Quantification of mucus producing cell (PAS positive) was performed by Image J software and is shown in the right panel (n = 5 independent animals; two-tailed unpaired t-test: *P = 0.0239). Scale bar: 50 μm. **g**, Immunofluorescence staining for α-SMA and Mucin5AC on cryosections from control and *Tet3*^smKO^ intestines (n = 3 independent animals). DNA was stained by DAPI. Scale bar: 50 μm. MFI of Mucin5AC was quantified by Image J and is shown on the right (n = 3; two-tailed unpaired t-test). **h**, Immunofluorescence staining for α-SMA & CD68 on paraffin sections from control and *Tet3*^smKO^ lungs, 8 weeks after tamoxifen injection (n = 3). DNA was stained by DAPI. Scale bar: 50 μm. **i**, H&E staining of cryosections from control and *Tet3*^smKO^ lungs, 6 months after tamoxifen injection (n = 5). Scale bar: 50 μm. **j**, Immunofluorescence staining for Ki67 & α-SMA (left panel) and CD45R & α-SMA (right panel) on cryosections from control and *Tet3*^smKO^ lungs, 6 months after tamoxifen injection (n = 3). DNA was stained with DAPI. Scale bar: 50 μm. **k**, FACS analysis of different immune cells in control and *Tet3*^smKO^ lungs, 6 months after tamoxifen injection (n = 5 independent animals; two-tailed unpaired t-test). **l**, RT-qPCR analysis of cytokine gene expression in FACS-sorted T cells (CD3⁺), 6 months after tamoxifen injection. *β-actin* was used for normalization (control n = 4, *Tet3*^smKO:T^ n = 3; two-tailed unpaired t-test: *P = 0.0322, *P = 0.0425, 0.0315). Data in (a–d, f, k, l) are presented as mean values ± s.e.m.

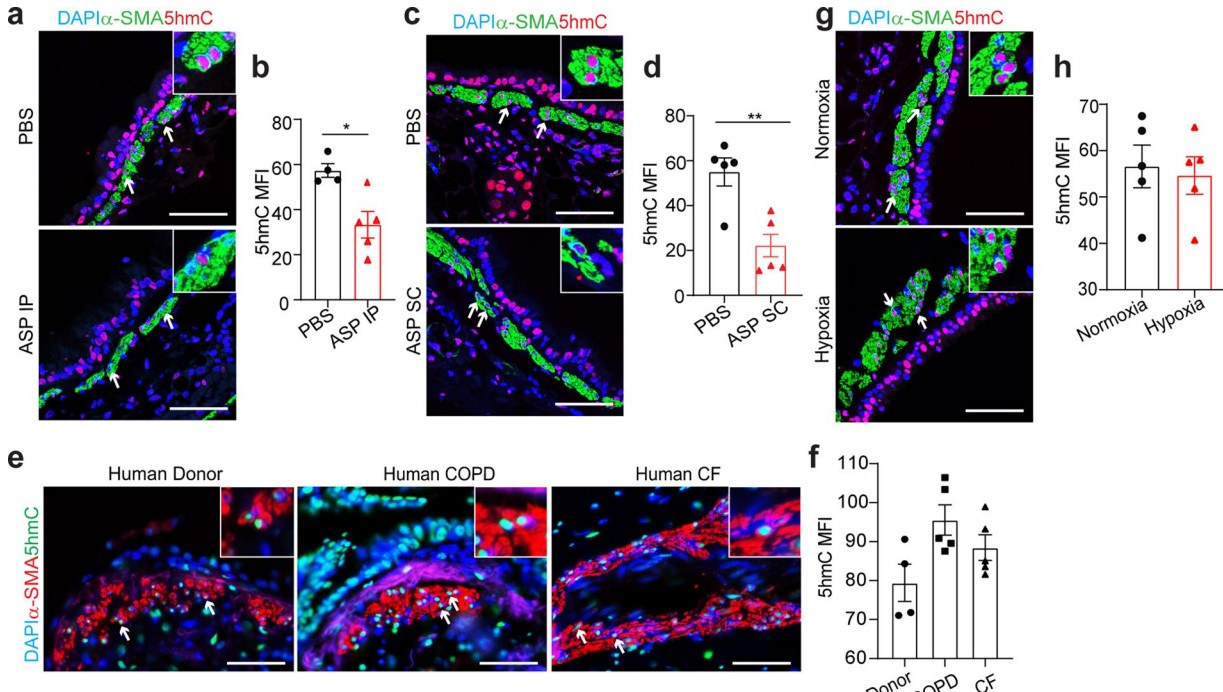

**Extended Data Fig. 9 | 5hmC levels are reduced in bronchial smooth muscle cells of mice with asthma. a&b,** Immunofluorescence staining for α-SMA and 5hmC on paraffin sections of mouse lungs after intraperitoneal injections (IP) and intranasal (IN) challenges of PBS or *Aspergillus fumigatus* (ASP) (a). Scale bar: 50 μm. DNA was stained by DAPI. Arrows indicate 5hmC-positive SMCs. Inserts depict enlarged images of 5hmC-stained SMCs from the respective panels. MFIs of 5hmC signals in lung SMCs were quantified by Image J and are shown in b (PBS: n = 4, ASP IP: n = 5; two-tailed unpaired t-test: *$P$ = 0.0126). **c&d** Immunofluorescence staining for α-SMA and 5hmC on paraffin sections of mouse lung after subcutaneous injections (SC) and IN challenges with ASP or PBS (c). Scale bar: 50 μm. DNA was stained by DAPI. Arrows indicate 5hmC-positive SMCs. Inserts depict enlarged images of 5hmC-stained SMCs from the respective panels. MFIs of 5hmC signals in lung SMCs were quantified by Image J and are shown in d (n = 5 independent animals; two-tailed unpaired t-test: **$P$ = 0.0035).

**e&f,** Immunofluorescence staining for α-SMA and 5hmC on lung paraffin sections from human donor (n = 4), COPD (n = 5) or cystic fibrosis (CF) patients (n = 5). Scale bar: 50 μm. DNA was stained by DAPI. Arrows indicate 5hmC-positive SMCs. Inserts depict enlarged images of 5hmC-stained SMCs from the respective panels. MFIs of 5hmC signals were quantified by Image J and are shown on the right (one-way ANOVA with Tukey's post hoc test). **g&h,** Immunofluorescence staining for α-SMA and 5hmC on lung paraffin sections from mice exposed to normoxia and chronic hypoxia (10%) for 4 weeks (n = 5 independent animals). Scale bar: 50 μm. DNA was stained by DAPI. Arrows indicate 5hmC-positive SMCs. Inserts depict enlarged images of 5hmC-stained SMCs from the respective panels. MFIs of 5hmC signals were quantified by Image J and are shown on the right (n = 5; two-tailed unpaired t-test). The enlarged images in (a, c, e, g) were scaled to 150×150 pixels. Data in (b, d, f, h) are presented as mean values.

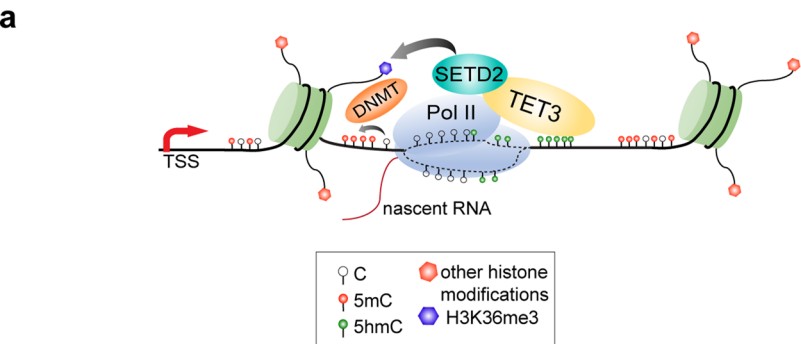

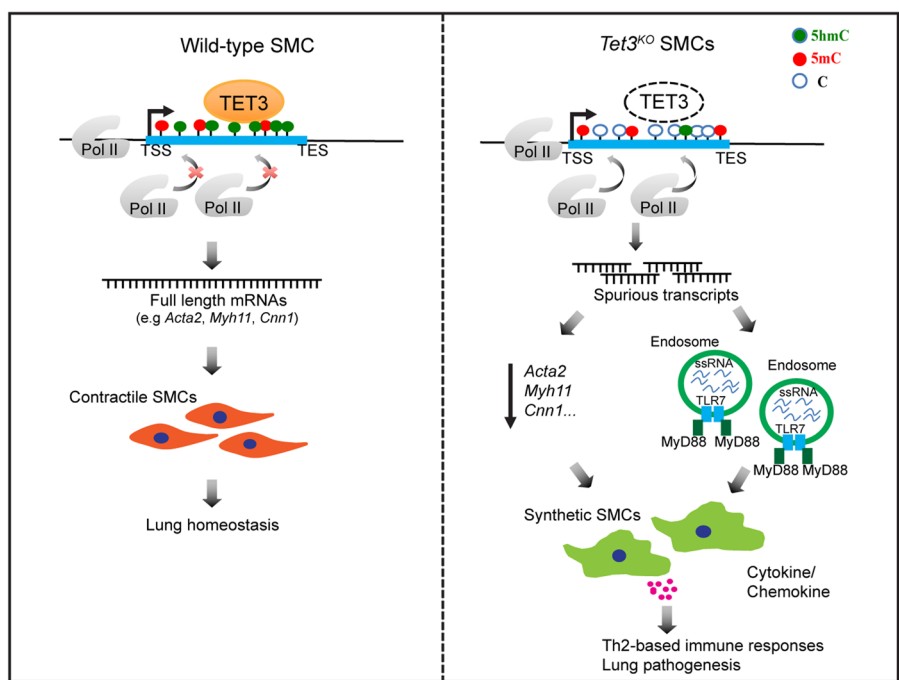

**Extended Data Fig. 10 | Reduction of 5hmC in SMCs leads to formation of spurious transcripts, causing phenotype switching, TLR7 activation, and airway inflammation. a**, Proposed model of the role of TET3 and 5hmC in transcriptional regulation. TET3 interacts with Pol II during transcriptional elongation and deposits 5hmC in actively transcribed genes when methylated cryptic starts sites (TSSs) or transposons are encountered. TET3 and/or 5hmC stabilize recruitment of SETD2 to the Pol II containing elongation machinery, which leads to increased formation of H3K36me3, preventing ectopic entry of Pol II and thereby enforcing usage of canonical, upstream promoters. Complete oxidation of 5hmC leading to demethylation might result in *de-novo* formation of 5mC, essentially resetting the chromatin for another round of transcriptional elongation. **b**, Depiction of the function of TET3 in SMCs. TET3-mediated formation of 5hmC in SMCs prevents intragenic entry of Pol II pSer5, aberrant transcriptional initiation within highly transcribed genes, and activation of TLR7 signaling. Reduction of 5hmC due to the absence of TET3 causes intragenic entry of Pol II that reduces production of contractile proteins and initiates formation of spurious transcripts, which activate endosomal TLR7 signaling and results in a phenotype switch in SMCs from the contractile to the synthetic state. Ongoing activation of innate immune responses and the phenotype switch in SMCs will provoke a T helper type 2 (Th2) cell-based immune response that accounts for the massive inflammation and pathology of *Tet3*-deficient airways.

# nature research

# Reporting Summary

Nature Research wishes to improve the reproducibility of the work that we publish. This form provides structure for consistency and transparency in reporting. For further information on Nature Research policies, see our Editorial Policies and the Editorial Policy Checklist.

## Statistics

For all statistical analyses, confirm that the following items are present in the figure legend, table legend, main text, or Methods section.

| n/a | Confirmed | |
|---|---|---|
| ☐ | ☒ | The exact sample size ($n$) for each experimental group/condition, given as a discrete number and unit of measurement |
| ☐ | ☒ | A statement on whether measurements were taken from distinct samples or whether the same sample was measured repeatedly |
| ☐ | ☒ | The statistical test(s) used AND whether they are one- or two-sided<br>*Only common tests should be described solely by name; describe more complex techniques in the Methods section.* |
| ☐ | ☒ | A description of all covariates tested |
| ☐ | ☒ | A description of any assumptions or corrections, such as tests of normality and adjustment for multiple comparisons |
| ☐ | ☒ | A full description of the statistical parameters including central tendency (e.g. means) or other basic estimates (e.g. regression coefficient) AND variation (e.g. standard deviation) or associated estimates of uncertainty (e.g. confidence intervals) |
| ☐ | ☒ | For null hypothesis testing, the test statistic (e.g. $F$, $t$, $r$) with confidence intervals, effect sizes, degrees of freedom and $P$ value noted<br>*Give P values as exact values whenever suitable.* |
| ☒ | ☐ | For Bayesian analysis, information on the choice of priors and Markov chain Monte Carlo settings |
| ☒ | ☐ | For hierarchical and complex designs, identification of the appropriate level for tests and full reporting of outcomes |
| ☒ | ☐ | Estimates of effect sizes (e.g. Cohen's $d$, Pearson's $r$), indicating how they were calculated |

*Our web collection on statistics for biologists contains articles on many of the points above.*

## Software and code

Policy information about availability of computer code

Data collection
Zeiss Z1 Imager - software Zen2 6.1.7601
Leica SP8 - software LAS X 3.5.7.23225
FACS Aria TM III (BD Biosciences) - BD FACS Diva v8 Software
ChemiDoc™ MP Imaging System - Image Lab Version 6.1.0 Standard Edition
All the sequencing data were collected using Illumina Nextseq500 platform

| Data analysis | GraphPad Prism 9 |
|---|---|
| | R 4.1.0 |
| | Macs2 peak caller version 2.1.0 |
| | IGV 2.3.52 |
| | BigWigAverageOverBed (UCSC Tools) |
| | DESeq2 package from Bioconductor (https://bioconductor.org/packages/release/bioc/html/DESeq2.html) |
| | GENCODE v25 |
| | FastQC tool http://www.bioinformatics.babraham.ac.uk/projects/fastqc. |
| | Trimmomatic version 0.39, (http://www.usadellab.org/cms/?page=trimmomatic) |
| | STAR 2.6.0c |
| | Picard package http://broadinstitute.github.io/picard/, Broad Institute 2.17.10 |
| | MUSIC peak caller (https://github.com/gersteinlab/MUSIC) |
| | UROPA (https://github.com/loosolab/UROPA) using the GENCODE annotation (V.16). |
| | BWA software (v0.5.9) |
| | HOMER |

For manuscripts utilizing custom algorithms or software that are central to the research but not yet described in published literature, software must be made available to editors and reviewers. We strongly encourage code deposition in a community repository (e.g. GitHub). See the Nature Research guidelines for submitting code & software for further information.

## Data

Policy information about availability of data

All manuscripts must include a data availability statement. This statement should provide the following information, where applicable:
- Accession codes, unique identifiers, or web links for publicly available datasets
- A list of figures that have associated raw data
- A description of any restrictions on data availability

GEO accession number: GSE166816. Figure 2a, Figure 2e, Figure 4a, b, e, g, Extended Data Figure 4a, b, Extended Data Figure 5a, b, are associated with this data.
GEO accession number: GSE 166815. Figure 2d, e, f, Figure 4a, Extended data Figure 3b, c, Extended data Figure 4d, are associated with this data.
GEO accession number: GSE 168206. Figure 4a, c, d, e, f, g, Extended data Figure 4c, d, are associated with this data.
Accession number: E-MTAB-10144 at www.ebi.ac.uk/arrayexpress/ (Password: dwmfpwxk). Extended data Figure 6d is associated with this data.
GEO accession number (Nano-seal): GSE 202201. Figure 4g, Extended data Figure 3b, c, Extended data Figure 5b are associated with this data.
GEO accession number (H3K36me3 ChIPseq): GSE 201924. Figure 3d, e, Extended data Figure 3h are associated with this data.
All data were already released to the public.

# Field-specific reporting

Please select the one below that is the best fit for your research. If you are not sure, read the appropriate sections before making your selection.

☒ Life sciences      ☐ Behavioural & social sciences      ☐ Ecological, evolutionary & environmental sciences

For a reference copy of the document with all sections, see nature.com/documents/nr-reporting-summary-flat.pdf

# Life sciences study design

All studies must disclose on these points even when the disclosure is negative.

| Sample size | Sample size were determined based on established practice and applicable standards. We opted for sample sizes which are commonly used sample size in the field. For in vivo studies, a minimum of three biological replicates were analyzed. For In vitro studies, experiments in which data were not quantified were performed with at least two replicates. Each experiment in which data were quantified was performed with at least 3 replicates. |
|---|---|
| Data exclusions | No data were excluded. |
| Replication | All in vivo studies were performed once with indicated number of animals. The in vitro study were performed at least 3 times independently and each replicate was successful. Sample sizes, statistical analyses and significance levels are all indicated in the figure legends or the method part. |
| Randomization | All animals were numbered and experiments were performed in a blinded pattern. After data collection, genotypes were revealed and animals assigned to groups based on genotype for data analysis. For in vitro study, the cultured cell plates were randomly assigned to each group for respective treatment. |
| Blinding | For in vivo study, all animals were numbered and experiments were performed in a blinded pattern. After data collection, genotypes were revealed and animals assigned to groups based on genotype for data analysis. In vitro experiments were not blinded during data collection or analysis since we know the treatment of each group before data collection. Positive controls, negative controls and samples were analyzed in exactly the same manner. |

# Reporting for specific materials, systems and methods

We require information from authors about some types of materials, experimental systems and methods used in many studies. Here, indicate whether each material, system or method listed is relevant to your study. If you are not sure if a list item applies to your research, read the appropriate section before selecting a response.

## Materials & experimental systems

| n/a | Involved in the study |
|---|---|
| ☐ | ☒ Antibodies |
| ☐ | ☒ Eukaryotic cell lines |
| ☒ | ☐ Palaeontology and archaeology |
| ☐ | ☒ Animals and other organisms |
| ☐ | ☒ Human research participants |
| ☒ | ☐ Clinical data |
| ☒ | ☐ Dual use research of concern |

## Methods

| n/a | Involved in the study |
|---|---|
| ☐ | ☒ ChIP-seq |
| ☐ | ☒ Flow cytometry |
| ☒ | ☐ MRI-based neuroimaging |

## Antibodies

**Antibodies used**

5hmC (1:5000, Active motif, #39769), 5mC (1:5000, Eurogentec, #81103), 5fC (1:5000, Active motif, #61223), 5caC (1:5000, Active motif, #61225), TET3 (1:1000, GeneTex, #GTX121453), TET3 (1:500, Active motif, #61395), TET2 (1:1000, BETHYL, #A303-604A), RNA Pol II Pan (1:1000, Active motif, #39097), RNA Pol II ser2 (1:1000, Abcam, #ab5095), RNA Pol II ser5 (1:1000, Abcam, #ab5408), SETD2 (Abclonal, #A3194), NSD3 (1:1000, Proteintech, #11345-1-AP), H3K36me3 (1:1000, Abcam, #ab9050), TLR7 (1:1000, Proteintech, #17232-1-AP), Myd88 (1:1000, Santa Cruz, #sc-74532), EEA1 (1:1000, Cell signaling, #3288T), RAB7 (1:1000, Cell signaling, #9367T), alpha-SMA-FITC (1:2000, Sigma, #F3777), alpha-SMA-Cy3 (1:2000, Sigma, #C6198), CCSP (1:1000, Millipore, #07-623), alpha-Tubulin (1:1000, Sigma, #T6074), Mucin-5AC (1:1000, Abcam, #ab3649), Ki67 (1:1000, Abcam, #ab92742), AGR2 (1:1000, Abcam, #ab76473), CD68 (1:1000, eBioscience, #E24592), Collagen I (1:1000, Proteintec, #14695-1-AP), alpha-SMA (1:1000, Sigma, #A2547), MYH11 (1:1000, Abcam, #ab5319), TAGLN (1:1000, Proteintec, #10493-1-AP), Vimentin (1:1000, Abcam, #ab92547), TPM4 (1:1000, Millipore, #AB5449), Pan actin (1:1000, Cell signaling, #4968), Histone H3 (1:5000, Abcam, #ab1791), CD3 (BD, #560590), CD45R (1:1000, Invitrogen, #4329548), Siglec-F (1:100, Miltenyi Biotec, #5161104265), CD11C (1:100, eBioscience, #4300054), CD11B (1:100, eBioscience, #4289817), F4/80, (1:100, Bio-Rad, #MCA497FT), LY6G (1:100, BD, #563005), MHCII (1:100, eBioscience, #4271684), CD4 (1:100, BD, #561830), CD3 (1:100, BD, #561798), CD196 (1:100, Biolegend, #129803), CD194 (1:100, Biolegend, #131213), CD183 (1:100, Biolegend, #126511).

**Validation**

5hmC (1:5000, Active motif, #39769), validated by dot blot with 27 base oligonucleotide containing 5-hydroxymethylcytosine (1.2 ng); validated by Methyl DNA immunoprecipitation with 25ng DNA, this antibody was validated with mouse genomic DNA for Dot blot from publications, such as Shu,L and et al, BMC genomic, 2016. 5mC (1:5000, Eurogentec, #81103), validated by Ito et al., 2010, Nature. 5fC (1:5000, Active motif, #61223), validated by dot blot with oligo containing 5-formylcytidine. This antibody has been validated with mouse tissue by previous publications, such as Fang, S, et al, Nat Common, 2019. 5caC (1:5000, Active motif, #61225), validated by dot blot with oligo containing 5-carboxylcytidine. This antibody has been validated with mouse tissue by previous publications, such as Fang, S, et al, Nat Common, 2019. TET3 (1:1000, GeneTex, #GTX121453), validated by immunoprecipitates TET3 protein with TET3 gene transfected 293T cells; validated by immunofluorescent analysis by HeLa cells. TET2 (1:1000, BETHYL, #A303-604A), validated by immunofluorescent analysis by human ovarian carcinoma sections. RNA Pol II Pan (1:1000, Active motif, #39097), validated by western blot analysis by HeLa nuclear extracts. RNA Pol II ser2 (1:1000, Abcam, #ab5095), validated by western blot with HeLa whole cell lysate. RNA Pol II ser5 (1:1000, Abcam, #ab5408), validated by chromatin immunoprecipitation with U2OC cells. SETD2 (Abclonal, #A3194), validated by western blot with mouse brain, mouse kidney and mouse liver. NSD3 (1:1000, Proteintech, #11345-1-AP), validated by immunofluorescent analysis by HeLa cells. H3K36me3 (1:1000, Abcam, #ab9050), validated by chromatin immunoprecipitation with U2OS cells. TLR7 (1:1000, Proteintech, #17232-1-AP), validated by immunohistochemical analysis with paraffin-embedded human small intestine tissue slide. MYD88 (1:1000, Santa Cruz, #sc-74532), validated by immunohistochemical analysis with paraffin-embedded human small intestine tissue. EEA1 (1:1000, Cell signaling, #3288T), validated by western blot with HeLa cell extracts. RAB7 (1:1000, Cell signaling, #9367T), validated by western blot with HeLa cell extracts. alpha-SMA-FITC (1:2000, Sigma, #F3777), validated by immunohistochemical analysis with paraffin-embedded tissue section of human appendix. This antibody has been validated with mouse tissue with previous publications, such as Dai, Z and et al, Circulation, 2016.alpha-SMA-Cy3 (1:2000, Sigma, #C6198), validated by immunohistochemical analysis with paraffin-embedded tissue section of human appendix. CCSP (1:1000, Millipore, #07-623), validated by immunohistochemical analysis with paraffin-embedded murine lung section. alpha-Tubulin (1:1000, Sigma, #T6074), validated by immunofluorescent analysis by CFB cells. Mucin-5AC (1:1000, Abcam, #ab3649), validated by immunohistochemical analysis with human gastric carcinoma tissue. Ki67 (1:1000, Abcam, #ab92742), validated by immunohistochemical analysis with human colorectal adenocarcinoma cell line. AGR2 (1:1000, Abcam, #ab76473), validated by immunofluorescent analysis with human breast adenocarcinoma epithelial cell. CD68 (1:1000, eBioscience, #E24592), validated by immunofluorescence analysis with THP-1 cells differentiated into macrophage. Collagen I (1:1000, Proteintec, #14695-1-AP), validated by immunohistochemical analysis with paraffin-embedded human skin cancer tissue slide. alpha-SMA (1:1000, Sigma, #A2547), validated by western blot with C2C12 whole cell lysate. MYH11 (1:1000, Abcam, #ab5319), validated by immunohistochemical analysis with frozen mouse small intestine tissue section. TAGLN (1:1000, Proteintec, #10493-1-AP), validated by immunofluorescent analysis with fixed mouse heart tissue. Vimentin (1:1000, Abcam, #ab92547), validated by immunohistochemical analysis with paraffin embedded mouse kidney. TPM4 (1:1000, Millipore, #AB5449), validated by Kramann et al., 2016, Cell Stem Cell. Pan actin (1:1000, Cell signaling, #4968), validated by western blot with HeLa whole cell lysate. Histone H3 (1:5000, Abcam, #ab1791), validated by western blot with HEK293 whole cell lysate. CD3 (BD, #560590), validated by flow cytometric analysis with mouse splenocytes. CD45R (1:1000, Invitrogen, #4329548), validated by flow cytometric analysis with mouse splenocytes. Siglec-F (1:100, Miltenyi Biotec, #5161104265), validated by flow cytometry analysis with bone marrow cells. CD11c (1:100, eBioscience, #4300054), validated by flow cytometric analysis with mouse splenocytes. CD11b (1:100, eBioscience, #4289817), validated by flow cytometric analysis with mouse bone marrow cells. F4/80, (1:100, Bio-Rad, #MCA497FT), validated by flow cytometric analysis with mouse peritoneal macrophages. LY6G (1:100, BD, #563005), validated by flow cytometric analysis with mouse bone marrow leukocytes. MHCII (1:100, eBioscience, #4271684), validated by flow cytometric analysis with mouse splenocytes. CD4 (1:100, BD, #561830), validated by flow cytometric analysis with mouse splenocytes. CD3 (1:100, BD, #561798),

validated by flow cytometric analysis with mouse splenocytes. CD196 (1:100, Biolegend, #129803), validated by flow cytometric analysis with mouse splenocytes. CD194 (1:100, Biolegend, #131213), validated by He W et al., 2016, Immunity. CD183 (1:100, Biolegend, #126511), validated by flow cytometric analysis with mouse splenocytes.

# Eukaryotic cell lines

Policy information about cell lines

| | |
|---|---|
| Cell line source(s) | HEK293T, HEK293, HeLa and MLE12 cells were purchased from ATCC. |
| Authentication | All human cell lines were obtained from ATCC and maintained as instructed. The cell was checked for morphology for cell authentication. |
| Mycoplasma contamination | tested negative for mycoplamsa |
| Commonly misidentified lines (See ICLAC register) | No commonly misidentified cell lines were used |

# Animals and other organisms

Policy information about studies involving animals; ARRIVE guidelines recommended for reporting animal research

| | |
|---|---|
| Laboratory animals | All mice were maintained on a C57BL/6 background, and littermates were used as controls in all experiments. All experiments were performed on balanced cohorts of male and female mice as our initial data did not indicate significant differences in lung airway morphological changes between females and males. ROSA26tdTomato mice were obtained from The Jackson Laboratory. To facilitate the lineage tracing experiment for IF and FACS sorting, we crossed the ROSA26tdTomasto with Tet3 flox mice to label smooth muscle cell specifically. The Tamoxifen injection was performed when the animals were 8 weeks old. |
| Wild animals | Studies did not involve wild animals. |
| Field-collected samples | Studies did not involve samples collected in the field. |
| Ethics oversight | All animal experiments were done in accordance with the Guide for the Care and Use of Laboratory Animals published by the US National Institutes of Health (NIH Publication No. 85-23, revised 1996) and were approved by the responsible Committee for Animal Rights Protection of the State of Hessen (Regierungspraesidium Darmstadt, Wilhelminenstr. 1-3, 64283 Darmstadt, Germany) with the project number B2/1125, B2/1137 and B2/1056. |

Note that full information on the approval of the study protocol must also be provided in the manuscript.

# Human research participants

Policy information about studies involving human research participants

| | |
|---|---|
| Population characteristics | Human donor samples, human COPD samples, human cystic fibrosis samples were provided by the DZL Biobank (Deutsches Zentrum für Lungenforschung). |
| Recruitment | Human samples were obtained during lung transplantation of human COPD and cystic fibrosis patients. Donor lung material was obtained as a result of atypical resections undertaken to adjust the donor organ to the recipient's thoracal cavity. |
| Ethics oversight | The study protocol for tissue donation was approved by the Ethics Committee of the Department of Human Medicine of Justus Liebig University Hospital, Giessen, Germany, in accordance with national law and with the 'Good Clinical Practice/ International Conference on Harmonization' guidelines. Approval code: (Az. 58/15 and 111/08).<br>Human asthma samples were obtained from the BioMaterialBank Nord, Clinical and Experimental Pathology Medicine, Research Center Borstel. Approval to use human samples for research was granted by the Ethics Committee of the University of Lübeck (Az 12-220 and 14-225). |

Note that full information on the approval of the study protocol must also be provided in the manuscript.

# ChIP-seq

## Data deposition

☒ Confirm that both raw and final processed data have been deposited in a public database such as GEO.

☒ Confirm that you have deposited or provided access to graph files (e.g. BED files) for the called peaks.

| | |
|---|---|
| Data access links<br>*May remain private before publication.* | https://www.ncbi.nlm.nih.gov/geo/query/acc.cgi?acc=GSE166815 |
| Files in database submission | Pol II ser5 ChIPseq:<br>GSM5086002    SMCs, ChIP, Mut_IP_1<br>GSM5086003    SMCs, ChIP, Mut_IP_2<br>GSM5086004    SMCs, ChIP, Mut_Input_1 |

GSM5086005    SMCs, ChIP, Mut_Input_2
GSM5086006    SMCs, ChIP, WT_IP_1
GSM5086007    SMCs, ChIP, WT_IP_2
GSM5086008    SMCs, ChIP, WT_Input_1
GSM5086009    SMCs, ChIP, WT_Input_2

H3K36me3 ChIPseq:
GSM6081162   H3K36me3 ChIPSeq [Mut_2]
GSM6081163   H3K36me3 ChIPSeq [Mut_3]
GSM6081164   Input DNA [Mut_2]
GSM6081165   Input DNA [Mut_3]
GSM6081166   H3K36me3 ChIPSeq [WT_2]
GSM6081167   H3K36me3 ChIPSeq [WT_3]
GSM6081168   Input DNA [WT_2]
GSM6081169   Input DNA [WT_3]

**Genome browser session**
(e.g. UCSC)

N/A

## Methodology

**Replicates**

Two biological replicates for RNA Pol II pSer5 ChIP-seq and H3K36me3 ChIP-seq.

**Sequencing depth**

Pol II ser5 ChIPseq:
Single end.
Length: 75bp
WT1 21M 69% aligned
WT2 23M 72% aligned
MUT1 16M 68% aligned
MUT2 24M 73% aligned

H3K36me3 ChIPseq:
WT2 21M 95% aligned
WT3 23M 94% aligned
MUT2 16M 94% aligned
MUT3 24M 94% aligned

**Antibodies**

RNA Pol II ser5 (Abcam, #ab5408), H3K36me3 (Abcam, #ab9050)

**Peak calling parameters**

Peaks were annotated with the promoter (TSS +- 5000 nt) of genes closely located to the centre of the peak based on reference data from GENCODE v25. The degree of reproducibility between samples was assessed by Spearman correlations. To permit comparative display of samples in IGV, raw BAM files were scaled with DESeq2 size factors based on all unified peaks using bedtools genomecov resulting in normalized BigWig files. Finally, DESeq2 was used to identify significantly differentially modified peaks based on background-corrected read counts from recounted unified peak regions.

**Data quality**

Pol II ser5 ChIPseq: Reproducible peaks in WT1 and WT2: 2949; promoter intersected peaks: 2325
Reproducible peaks in MUT1 and MUT2: 2935; promoter intersected peaks: 2312
H3K36me3 ChIPseq: Reproducible peaks in WT2 and WT3: 15886;
Reproducible peaks in MUT2 and MUT3: 15835

**Software**

Macs2 peak caller version 2.1.0
IGV 2.3.52
BigWigAverageOverBed (UCSC Tools)
DESeq2 package from Bioconductor
GENCODE v25
FastQC tool http://www.bioinformatics.babraham.ac.uk/projects/fastqc.

## Flow Cytometry

### Plots

Confirm that:

☒ The axis labels state the marker and fluorochrome used (e.g. CD4-FITC).

☒ The axis scales are clearly visible. Include numbers along axes only for bottom left plot of group (a 'group' is an analysis of identical markers).

☒ All plots are contour plots with outliers or pseudocolor plots.

☒ A numerical value for number of cells or percentage (with statistics) is provided.

## Methodology

Sample preparation

After sacrificing experimental mice, blood was removed by perfusion with cold PBS through the right ventricle prior to lung dissection. Lung tissues were dissected and minced into small pieces before incubation in 3 ml digestion buffer (DPBS containing Collagenase type 2 (2mg/ml, Worthington), Elastase (0.04mg/ml, Worthington) and DNase (5U/ml, Roche) with frequent agitation at 37o C for 10min. Immediately afterwards, 10-times the volume of cold DMEM supplemented with 10% FBS was added to single-cell suspensions. Cells were mechanically dissociated by passing 4-5 times through a 30 ml syringe and consecutive filtering through 100-, 70- and 40-µm cell strainers (BD Biosciences). The filtrate was centrifuged at 300 g, room temperature (RT) for 10min. Pellets were re-suspended in 1ml pre-cooled MACS buffer (Cat# A9576, Miltenyi Biotec) with 1% BSA. After 5min centrifugation at 300g, cell pellets were re-suspended in 90µL MACS buffer and incubated with 10 µL CD45 MicroBeads (Cat#130-052-301) and anti-Ter-119 MicroBeads (Cat#130-049-901) at 4o C for 15min to remove hematopoietic cells. After washing with MACS buffer, cells were loaded into pre-conditioned LS columns (Miltenyi Biotec) on a MACS separator and the flow-through containing unlabeled cells was collected.

Instrument

FACS sorting for smooth muscle cells: FACSAriaTM III (BD Biosciences)
FACS analysis for lung inflammatory cells: LSR Fortessa (BD Biosciences)

Software

BD FACS Diva v8 software

Cell population abundance

Sorted cells were reanalyzed to assess purity. A 80% purity was achieved. Data related to reanalysis of sorted cells is shown in Supplementary Figure 1.

Gating strategy

The gating strategy is shown in Extended data Fig. 1j, Supplementary Figure 1, Supplementary Figure 2.

☒ Tick this box to confirm that a figure exemplifying the gating strategy is provided in the Supplementary Information.

