## [Peer Review File · Nature Genetics]

Peer Review Information

Manuscript Title: Spurious transcription causing innate immune responses is prevented by 5-hydroxymethylcytosine

Corresponding author name(s): Thomas Braun, Xuejun Yuan

Reviewer Comments & Decisions:

Decision Letter, initial version:
--

21st May 2021

Dear Professor Braun,

Your Article entitled "Spurious transcription causing innate immune responses is prevented by 5hmC" has now been seen by 3 referees, whose comments are attached. While they find your work of potential interest, they have raised serious concerns which in our view are sufficiently important that they preclude publication of the work in Nature Genetics, at least in its present form.

While the referees find your work of some interest, they raise concerns about the strength of the novel conclusions that can be drawn at this stage.

In brief, all three reviewers found the central finding of your study to be of potentially broad interest. Concerns are raised regarding the strength of support for your model, based on the data presented; but the reviewers also provide clear and constructive guidance on how these concerns could be experimentally addressed.

Reviewer #1 (who studies lung diseases) thinks that the data support the proposed model; their major request is, in our reading, that a link to human disease be demonstrated.

Reviewer #2 is more critical of the experiments and the conclusions drawn. They make a number of detailed comments regarding many of the analyses performed and request specific experiments to address these concerns.

Reviewer #3 questions whether it has been robustly demonstrated that it is the loss of 5hmC - rather than Tet3 protein acting as a scaffold - that underlies the proposed mechanism. They suggest rescue experiments using a WT and catalytic-dead mutant that would definitively answer this point.

Our opinion is that the suggested further work - while substantial - would, in sum, significantly improve the confidence placed in your study. In particular, we thought that Reviewer #1's comment on the disease relevance of your findings, and Reviewer #3's suggestion for the rescue experiments, should be comprehensively addressed in a revision.

Should further experimental data allow you to fully address these criticisms we would be willing to consider an appeal of our decision (unless, of course, something similar has by then been accepted at Nature Genetics or appeared elsewhere). This includes submission or publication of a portion of this work someplace else.

The required new experiments and data include, but are not limited to those detailed here. We hope you understand that until we have read the revised manuscript in its entirety we cannot promise that it will be sent back for peer review.

If you are interested in attempting to revise this manuscript for submission to Nature Genetics in the future, please contact me to discuss a potential appeal. Otherwise, we hope that you find our referees' comments helpful when preparing your manuscript for resubmission elsewhere.

Sincerely,

Michael Fletcher, PhD
Associate Editor, Nature Genetics

ORCID: 0000-0003-1589-7087

Referee expertise:

Referee #1: lung disease and genetics.

Referee #2, #3: epigenetics.

Reviewers' Comments:

Reviewer #1:

Remarks to the Author:

The paper by Wu et al reveals that loss of TET3 in smooth muscle cells(SMC) results in aberrant transcripts produced by faulty methylation in the gene body. The aberrant transcripts that are produced by inappropriate de-methylation stimulate an innate immune response via release of single strand (ss) RNA and sensing by TLR7/8. This leads to de-differentiation of airway smooth muscle cells (SMC) and an inflammatory response that alters airway epithelial cells. The concept is novel related to the mechanism of stimulation of innate immunity as well as TET3 mediated gene regulation. The experiments are logical and sound in reinforcing the concept and its sequelae. However there are several questions that should be addressed and it is important to relate these observations to human disease.

(i)Are specific transcripts particularly vulnerable to loss of TET3 and why?

(ii)Under what pathophysiological conditions could TET3 be reduced. For example, hypoxia has been

implicated and should be tested.

(iii) Is there evidence of an interferon response stimulated initially by production of ssRNA and does this change over time as epithelial cells are impacted.

(iv) What lung disease does this simulate? Are the smooth muscle cells that are de-differentiated proliferative or are they senescent? Proliferative SMC are seen in asthma. Epithelial mesenchymal transition is linked to fibrosis in the airways. Does this occur?

(v) Why are no changes seen in vascular smooth muscle cells that presumably should use the same mechanism?

Specific Comments:

Results: line 103. Here the authors should indicate the age of the mice when the knockout was induced

line 108: The epithelial morphological changes are presumably related to some TLR7/8 activity but is this substantiated immediately after tamoxifen?

It is curious that the only data on vascular smooth muscle cells relates to TET2. When SMC were isolated from the murine lungs there was no distinction. It is therefore important to show parallel photomicrographs of distribution of 5HMC and 4 HC in the pulmonary artery SMC, unless TET3 is uniquely present in airway SMC which would mean that in the artery TET2 does compensate. This needs to be clarified.

Lin 231: The discussion on TLR-7 and macrophages seems to come out of the blue. Is the same paradigm present in macrophages, i.e. do they also use TET3? This needs more in depth assessment. Alternatively is there evidence that SMC are becoming macrophages as is discussed later in the Results section. Is TET3 the switch that keeps SMC from being synthetic or from being macrophages? This is not given extensive enough experimentation and may be a very important concept. Were there alterations in transcription factors that caused the switch in phenotype? Which ones? What confers specificity of TET3 at sites of H3K27trimethyl marks?

Line 289:

Is there heightened proliferation in bronchial smooth muscle as in asthma? The photomicrographs don't appear to show either an increase in SMC or in fibrosis so most of the impact of the inflammatory response appears to be paracrine on epithelial cells presumably via cytokines.

In general more data are needed to establish the model and why airway not arterial SMC are affected and whether there is a morphogenesis to a macrophage like cell or to a proliferating SMC .

We presume that the general elevation in cytokines are inducing the changes in epithelial cells but this should be studied in greater depth.

Reviewer #2:

Remarks to the Author:

Wu et al. used a conditional knockout strategy to target exon 10 of the Tet3 gene in a mouse model, aimed at removing the catalytic domain. This enzyme creates 5-hydroxymethylcytosine (5hmC) from 5mC in gene bodies and other regions and may cause DNA demethylation. They observed that Tet3 inactivation produced a phenotype in smooth muscle cells, in particular in the lung. They claim that loss of TET3 allows the formation of spurious intragenic transcripts in more highly expressed genes and activates the endosomal nucleic acid-sensing TLR7/8 signaling pathway. The immune response induces inflammation in the lung and airway remodeling. Although such findings would be of interest,

both from an epigenetic regulatory and lung biology perspective, the conclusions of the authors are in many instances not supported by the data. I have the following major concerns:

- 1) The knockout has not been verified properly. Although there is an RT-PCR assay for Tet3 (Fig. Ext Data 1), no Western blot and not even a PCR spanning exon 10 in the targeted cell population has been shown.
- 2) The Tet3-LacZ expression data is strange. The authors state that expression is confined to smooth muscle cells (SMCs) in adult mice. According to ENCODE and GTEX (human) data, however, this gene is expressed in almost all adult tissues or cell types.
- 3) The authors state that the concentration of functional full-length mRNAs dropped after Tet3 inactivation. How was this determined or verified?
- 4) The causality of airway inflammation due to aberrant spurious transcription is not proven.
- 5) Figure 1f:
5hmC is not really increased in gene bodies.
Is the dip near the TSS related to high GC-content, which is causing a reduced PCR efficiency over these regions?
- 6) Figure 2e:
It is hard to interpret the data without an IgG control in the IP.
Figure 2g is not convincing.
- 7) Figure 2h shows some Q-PCR data for H3K36me3 but it is hard to make a case for H3K36me3 changes based on this limited information.
- 8) The combined data in Figure 3 do not make a strong case for the conclusion that TET3 prevents intragenic transcription initiation.
- 9) Figure 4:
It is not clear that one can claim from an RNA transfection experiment that spurious transcripts would activate the TLR7 signaling pathway.
- 10) Figure 5b: There are large differences in band intensities between individual samples, even in the control groups, which makes it difficult to have confidence in the conclusions.
- 11) I am not sure that the authors performed strand-specific RNA sequencing. Finding transcripts on the opposite strand in gene bodies would provide much better and more convincing evidence for the model the authors are proposing.
- 12) hMeDIP is a procedure fraught with potential artifacts due to the binding of the antibody to certain repetitive DNA sequences. Have the authors confirmed their results with an alternative method?
- 13) I am not convinced that reduced 5hmC formation after inactivation of TET3 will lead to reduced levels of H3K36me3.

14) The authors propose that ssRNA degradation products may be recognized by TLR7/8. However, what about ds RNA, perhaps in analogy with retroelement transcripts observed after treatment of cells with 5-aza-cytidine?

Minor points:

Figure 1b is not labeled correctly. It seems to show different organs.

The Discussion section (lines 307-312) contains several unsupported speculative sentences.

Reviewer #3:

Remarks to the Author:

In this manuscript, Wu et al. propose a novel role for Tet3 in preventing spurious transcription and an innate immune response in lung SMCs. The authors show that Tet3 preserves the integrity of transcription through interaction with RNA polymerase II. In addition, authors show that loss of Tet3 stimulates an innate immune response that recruits immune cells to the lung.

Although the manuscript is conceptually intriguing, there are some concerns outlined below.

Major:

1. The title of the manuscript implies that 5hmC prevents spurious transcription. A large proportion of the work focuses on TET3's association with RNA polymerase. More work is required to determine if it is 5hmC or Tet3 that is important for spurious transcription:

- How does TET3's genomic localization change before and after DRB in SMCs.
- Use changes in 5hmC to classify spurious transcripts.
- Show that combined loss of Tet2 and Tet3 lowers 5hmC.
- Show gene tracks with RNA-seq, ChIP-seq, 5-hmC. If indeed intragenic 5hmC is altering transcription, it is important to show a version of Fig. 3g with ChIP-seq.

2. Rescue experiments

a. Re-express wildtype and catalytically inactive TET3 in SMCs to evaluate changes in 5hmC, gene expression, and spurious transcripts.

b. Show tracks of specific loci that are rescued by either wildtype or catalytically inactive TET3.

3. Authors state that there are no evident abnormalities in SMC containing organs other than the lung. A change in weight is observed after tamoxifen injection.

a. Can authors delineate the relationship between inflammation in Tet3 KO lung SMCs and weight loss?

b. If there is a relationship with enhanced TLR signaling, can the authors use a TLR inhibitor (i.e., Telratolimod, R-848, or E6446) in vivo to reduce spurious transcription?

Minor:

1. The observation that spurious transcription is clearer on highly transcribed genes is expected (Fig 1F). Can authors combine the 4 panels into one panel and add boxplots comparing the signal of 5hmC at TSS, Gene body, and TES in WT and KO cells.

2. Provide a western blot to show KO of TET3? Please also probe for TET2 and TET1. Extended Fig. 1E shows some signal overlapping between SMA and TET3 in Tet3smko cells.

3. Does loss of TET3 in SMCs cause increased cell growth/proliferation?

4. Is there any effect on RNA polymerase II after loss of Tet3 without induction of DRB?

5. Fig 2E – TET3 OE seems to increase protein levels of elongating RNA polymerase. Do the authors observe any change in the protein levels of RNA polymerase components after loss of TET3 in SMCs?
6. Quantify PLA in 2F as dots/cell. It is unclear what % of PLA/nuclei means. Please clarify.
7. Perform ChIP-seq of H3K36me3 and compare with gene expression, 5hmC, and spurious transcripts (pol2 chip and cage-seq). Could authors clarify why some genes (Myh11 and Cnn1) have increase H3K36me3 at the promoter?
8. The lack of reproducibility in Fig. 3e between the replicates is concerning. On line 185, authors state that 7761 genes have a log2 ratio >1. Is this in both replicates?
9. Fig. 3a is a GO analysis of spurious transcripts, Fig. 4a is a KEGG analysis of RNA-seq data. Can the authors perform KEGG analysis on spurious transcripts since it has already been defined in Figure 3?
10. PLA in 4C – please quantify spots/cell if possible and keep it consistent throughout the manuscript.
11. Rather than exogenously introducing RNA from Ctrl and KO SMCs in HELA and 293T cells, could the authors generate TLR7 KO (with CRISPR) or KD (with si or shRNA).
12. Lines 254-257 – Can authors show/test this?
13. No mention of Fig. 5f in the text.
14. Ex. Data Fig. 2 – Good piece of data. Please show western blot confirming ablation of TET2.
15. Ex. Data Fig. 4 –
 - a. Some of the significant qPCR plots (Myh11 and Cnn1) seem to be driven by an outlier. Please clarify. Add replicate if possible.
 - b. (h) – both control replicates have different levels of proteins. Vim ctrl #2 is like Tet3 ko #2
16. Ex. Data Fig. 5 – Add TET2 to western blot in (b).
17. Clarify in legend if the fold change is KO/Ctrl in (d) and include quantification (MFI) for (g) and (j).

Author Appeal

Responses to Reviewers' Comments:

Reviewer #1:

Remarks to the Author:

The paper by Wu et al reveals that loss of TET3 in smooth muscle cells (SMC) results in aberrant transcripts produced by faulty methylation in the gene body. The aberrant transcripts that are produced by inappropriate de-methylation stimulate an innate immune response via release of single strand (ss) RNA and sensing by TLR7/8. This leads to de-differentiation of airway smooth muscle cells (SMC) and an inflammatory response that alters airway epithelial cells. The concept is novel related to the mechanism of stimulation of innate immunity as well as TET3 mediated gene regulation. The experiments are logical and sound in reinforcing the concept and its sequelae. However, there are several questions that should be addressed and it is important to relate these observations to human disease.

Response: We thank the reviewer for pointing out the potential disease relevance of the SMCspecific *Tet3* knockout (KO) mouse model, which was very helpful for further improving the manuscript. We discovered that human asthma patients as well as mouse asthma models are characterized by a pronounced reduction of 5-hmC in bronchial smooth muscle cells. The finding that loss of TET3 causes an asthma-like phenotype strongly suggests that the reduction of 5-hmC in airways of human asthma patients is not an epiphenomenon but causally involved in the pathogenesis of asthma. We have also

fully addressed all other comments. Please find below a detailed point-by-point response to all comments.

(i) Are specific transcripts particularly vulnerable to loss of TET3 and why?

Response: Our results indicate that highly expressed genes are more vulnerable to the loss of TET3 than lowly expressed. Genes highly expressed in smooth muscle cells happen to be genes that code for smooth muscle-specific functions, such as genes responsible for formation of the contractile apparatus. This principal observation is extensively documented in the manuscript. To further characterize the specific chromatin configuration that distinguish spuriously expressed genes (515 genes defined in revised Fig. 3a) from non-spurious genes (randomly selected 515 genes), we analyzed published ATAC-seq data in human SMCs (Kim et al., 2020) and found higher chromatin accessibility for spurious compared to non-spurious genes (Fig. 1a for reviewer, revised Extended data Fig.5e). Furthermore, intragenic deposition H3K4me3, a histone marker involved in recruitment of Pol II pre-initiation complex at canonical TSSs (Lauberth et al., 2013), was higher for spurious than for non-spurious genes (Fig. 1b for reviewer, revised Extended data Fig.5f). We assume that spurious genes, most of which are cell identity related (revised Fig.3e), are more vulnerable to ectopic initiation and generation of spurious transcripts, since they are transcribed at a high level and attract Pol II more efficiently due to the enhanced DNA accessibility and H3K4me3 enrichment. In addition, we found, by analyzing the available H3K4me2 dataset for human SMCs, that spurious genes identified in our study contain higher H3K4me2 enrichment over the canonical TSSs compared to randomly selected, highly transcribed non-spurious genes (Fig. 1c for reviewer, revised Extended data Fig.5g). High H3K4me2 levels have been implicated to represent a stable “epigenetic signature” conferring lineage identity and correlate with active transcription of SMC cell identity-related genes (Liu et al., 2021). These findings imply that spuriously expressed genes (SMC cell lineage identity genes) are characterized by an “epigenetic signature”, which retains the chromatin in a more accessible state that is more vulnerable to initiation of spurious transcriptional initiation than lowly expressed genes with a less accessible chromatin state.

Figure 1 for reviewer: a, DNA accessibility of spurious and non-spurious genes based on ATAC-seq data in human SMCs (GSM4558338). b, Profile of H3K4me3 enrichment (GSM112417) within gene body regions of 515 spuriously expressed genes (defined in Fig.3a) in SMCs and randomly selected 515 non-spuriously expressed

genes. c, Density profile of H3K4me2 enrichment over annotated TSS (GSE96375) of 515 spuriously expressed genes, random selected 515 non-spuriously expressed genes, and 515 randomly selected highly expressed genes.

(ii) *Under what pathophysiological conditions could TET3 be reduced. For example, hypoxia has been implicated and should be tested.*

Response: We described in the previous version that inactivation of *Tet3* in smooth muscle cells causes an inflammatory phenotype in the lung, which is similar to asthma but also resembles COPD or cystic fibrosis (CF) to a certain degree. To study the link to human diseases, we obtained lung samples from mouse models of asthma and also clinical samples from human asthma, COPD and cystic fibrosis patients.

We observed a significant reduction of 5hmC in bronchial smooth muscle cells from human asthma patients and in two different mouse asthma models (house dust mites, aspergillus fumigatus treated) (revised Fig.6e-h; revised Extended data Fig.9a-d) but not in samples from human COPD and cystic fibrosis patients (revised Extended data Fig.9e&f). These results indicate reduced activity or reduced expression of TET3 in asthma. Unfortunately, available antibodies against TET3 do not work on sections, so that we cannot measure the concentration of TET3 proteins in human samples. We also followed the reviewer's suggestion and measured the 5hmC content of bronchial smooth muscle cells after subjecting mice to chronic hypoxia (10% O₂ for 4 weeks). Surprisingly, no reduction of 5hmC was seen in hypoxia-treated mouse lungs in vivo, indicating that chronic hypoxia does not efficiently inhibit TET3 activity in vivo (revised Extended data Fig.9g&h).

(iii) *Is there evidence of an interferon response stimulated initially by production of ssRNA and does this change over time as epithelial cells are impacted.*

Response: To analyze whether enhanced expression of interferon response-related genes after *Tet3* knockout depends on the TLR7 signaling pathway, we used the TLR7 inhibitor E6446. Endosomal TLR7 is usually activated by binding of single-stranded RNA derived from pathogenic (bacterial or viral) nucleic acid degradation products. Transfection of RNA extracted from *Tet3* KO primary SMCs stimulated the TLR7-signaling pathway including expression of cytokines in recipient cells, which was absent when recipient cells were treated with the TLR7 inhibitor E6446 (revised Fig.4d). These findings indicate that ssRNA in *Tet3*-KO SMC are recognized by TLR7 that plays a key role in expression of downstream cytokine/chemokine genes. We further validated these findings by using *Tet3*-knockdown (KD) SMCs derived from mouse ESCs. Consistently, treatment with the TLR7 inhibitor E6446 prevented augmented expression of cytokine/chemokine genes in *Tet3* KD SMCs (revised Extended data Fig.6h) and in HeLa cells transfected with RNA from *Tet3* KD SMCs, (revised Fig.4g).

To follow the reviewer's specific request to characterize the initial response, we tracked changes in SMC in which *Tet3* is inactivated and the impact of these change on neighboring epithelial (airway) and endothelial cells (vasculature) over time. We sorted smooth muscle cells (SMC), endothelial cells (EdC), and epithelial cells (EpC) 2w, 4w and 8w post-TAM administration and performed RT-qPCR analysis to monitor expression of interferon response related genes such as *Irf7*, *Ifnb*, *Nfkb* in SMC, EpC, and EdC. We observed a significantly increased response of interferon-related genes in lung SMCs and epithelial cells 8 weeks but not 4 weeks after TAM treatment (revised Extended Fig.8a&b). Interestingly, we did not observe obvious changes of interferon-related genes in endothelial cells at any investigated time point (revised Extended Fig.8b). To examine whether cytokines/chemokines produced in *Tet3* KO SMCs elicit paracrine effects on neighboring epithelial cells and thereby enhance expression of interferon response-related genes, we cultured epithelial cells (MLE12 cells) with conditioned medium from ESC-derived control and *Tet3* KD SMCs. Conditioned medium from *Tet3* KD SMCs strongly increased expression of pro-inflammatory genes such as *Il6*, *Il1b* and *Ifnb* as well as of EMT related genes such as *Fn1*, *Cdh1*, *Vim* (revised Extended Fig.8c). Taken together, these findings indicate that airway epithelial cells are more vulnerable than vascular endothelial cells to respond to paracrine effects of cytokines/chemokines that initially occur in *Tet3* KO SMCs but then differentially affect neighboring cells.

(iv) What lung disease does this simulate? Are the smooth muscle cells that are de-differentiated proliferative or are they senescent? Proliferative SMC are seen in asthma. Epithelial mesenchymal transition is linked to fibrosis in the airways. Does this occur?

Response: Inactivation *Tet3* in SMCs causes pathological changes similar to human asthma. We observed hyperplasia of mucus-producing cells, Th2-cell based immune responses, enhanced fibrosis in *Tet3*-deficient lungs and deregulated expression of EMT-related genes in epithelial cells cultured with conditional medium from *Tet3* KD cells (revised Fig.6a, c&d, revised Extended data Fig. 8c). Importantly, we found a dramatic reduction of 5hmC in human asthma samples and in mouse asthma models but not in CF and COPD patients, indicating that loss of TET3 and subsequent reduction of 5hmC cause an asthma-like pathology (revised Fig.6e-h; revised Extended data 9a-e).

We did not detect an increase of SMC numbers or an increase of SMCs expressing the proliferation marker Ki67 in *Tet3*-mutant lungs (revised Extended data Fig.7g). Expression of cell cycle-related genes were not altered in sorted *Tet3* KO SMC as well (Figure 2 for reviewer). However, we detected a striking increase of the number of senescent cells (SA-b-Gal positive) in the bronchial smooth muscle layer of *Tet3*-mutant lungs (revised Extended data Fig. 7h). Moreover, expression levels of the senescence marker genes *p16* (*Cdkn2a*) and *p21* (*Cdkn1a*) were significantly elevated in mutant SMCs (revised Extended data Fig. 7i). Cellular senescence and subsequent acquisition of a senescence-associated secreted phenotype (SASP) has been suggested to play an important role in subgroups of asthma patients e.g. adult-onset asthma (Wang et al., 2020). Since *Tet3*-deficient SMCs produce several

cytokines characteristic for SASP, including IL1b, we conclude that loss of TET3 causes senescence but not proliferation of bronchial SMCs.

Figure 2 for reviewer: Expression of *Plk1*, *Ccne1*, *Ccnd1*, *Ccnb1*, *Cdk7*, *Cdk9*, *Ki67* genes based on RNA-seq of sorted control and Tet3^{smKO:T} lung SMCs (n=2; two-tailed unpaired t-test).

(v) *Why are no changes seen in vascular smooth muscle cells that presumably should use the same mechanism?*

Response: The reviewer raises an interesting question that also puzzles us. The main pathological phenotype after inactivation of *Tet3* is clearly in bronchial smooth muscle cells (BSMCs) but not in VSMCs, although 5hmC levels in WT lungs are similar in bronchial and vascular smooth muscle cells. Similarly, depletion of 5hmC was detected in both bronchial and vascular SMCs after knockout of *Tet3* (revised Extended data Fig. 2m). Since *Tet2* has been implicated in regulation of vascular SMC plasticity, we generated double mutant mice, in which both *Tet2* and *Tet3* are absent in SMCs. Interestingly, loss of *Tet2* in *Tet3* KO mice led to a further decline of 5hmC in lung VSMCs but not in BSMCs, indicating that TET2 might exert specific functions and partially compensates for TET3 in VSMCs (revised Extended data Fig. 2m). The reduction of α -SMA expression in *Tet2/Tet3* compound mutant SMCs in the aorta, outside the lung, supports this conclusion (see below, Figure 3 for reviewer). However, concomitant loss of *Tet2* and *Tet3* did not cause a major pathology of lung vessels, which matches our findings in the aorta (see below, Figure 3 for reviewer).

Figure 3 for reviewer: **a**, RT-qPCR analysis to monitor expression of *Tet2* and *Tet3* in aortas and isolated SMCs from control and *Tet3*^{smKO:T} lungs, 8 weeks after tamoxifen injection (n=3; two-tailed unpaired t-test: *p<0.05). **b**, Western blot analysis of α -SMA, TAGLN in aortas from control and *Tet3*^{smKO} lungs, 8 weeks after tamoxifen injection. Pan-actin was used as loading control. **c**, Immunofluorescence staining for α -SMA and MYH11 on paraffin sections from control and *Tet2/Tet3*^{smKO} aortas, 8 weeks after tamoxifen injection (n=3). DNA was stained by DAPI. Scale bar: 50 μ m. **d**, Immunofluorescence staining for α -SMA and 5hmC on paraffin sections from control, *Tet3*^{smKO} and *Tet2/Tet3*^{smKO} aortas, 8 weeks after tamoxifen injection (n=3). Quantification of mean fluorescence intensity (MFI) of 5hmC was performed by Image J and is shown in the right panel (n=3; one-way ANOVA with Tukey's post hoc test: **p<0.01). DNA was stained by DAPI. Scale bar: 50 μ m. **e**, Immunofluorescence staining for α -SMA and 5hmC on paraffin sections from control, *Tet3*^{smKO} and *Tet2/Tet3*^{smKO} intestines, 8 weeks after tamoxifen injection (n=3). Quantification of mean fluorescence intensity (MFI) of 5hmC was performed by Image J and is shown in the right panel (n=3; one-way ANOVA with Tukey's post hoc test: *p<0.05). DNA was stained by DAPI. Scale bar: 50 μ m. **f**, RT-qPCR analysis to monitor expression of *Tlr7*, *Il1b*, *Irf7* in aortas, bladders, and intestines from control and *Tet3*^{smKO} lungs, 8 weeks after tamoxifen injection (n=3; two-tailed unpaired t-test). **g**, RT-qPCR analysis of HeLa cells mock-transfected and transfected with RNA isolated from control and *Tet3*^{smKO} aortas and bladders (n=3; one-way ANOVA with Tukey's post hoc test).

In lung VSMCs the presence of *Tet2* does not prevent the phenotype switch from contractile to synthetic gene expression caused by *Tet3* inactivation (revised Extended data Fig. 7b&c). In our view,

the most likely possibility for the relatively normal morphology and function of the vasculature is that BSMCs are more vulnerable to the loss of 5hmC than VSMCs. We do not know whether this is due to a lower rate of spurious transcription in VSMCs compared to BSMCs or a lower threshold of BSMCs to activate innate immune responses as an evolutionary selected mechanism against viral infections. We assume that the phenotype switch from contractile to synthetic gene expression in *Tet3*-deficient VSMC is the consequence of the massive lung inflammation caused by *Tet3*-deficient BSMC, since only compound *Tet2/Tet3* mutant but not *Tet3*-mutant VSMCs in the aorta showed a reduction in contractile genes expression (Figure 3b-d for reviewer). It will be interesting to investigate whether the extent of spurious transcription is lower in VSMCs than in BSMCs, which may be attributed to an additional mechanism for suppression of ectopic transcription that has emerged in VSMCs.

Another possibility is a lower threshold in BSMCs to activate innate immune responses, which biologically makes sense, since cells in the airways are more exposed to the environment and have to be ready to activate innate immune responses. To investigate such possibilities, a reliable way to separate VSMCs from BSMCs in the lung is necessary, which is presently not available.

We strongly believe that the responsiveness of cells next to *Tet3*-deficient SMCs is crucial for the pathological changes observed in mutant lungs. We followed this idea by culturing epithelial lung cells in medium derived from *Tet3* KD SMCs and detected elevated expression levels of EMT marker genes (revised Extended data Fig. 8c). In addition, interferon response related genes (i.e. *Irf7*, *Ifnb*, *Nfkb*) and EMT or epithelial cell identity marker genes (*Epcam*, *Cdh1*, *Muc5ac*, *Vim*, *Fn1*, *Cdh2*, *Snail1*, *Snail2*) were deregulated in bronchial epithelial cells but endothelial cells identity marker genes (*Vwf*, *Cdh5*, *Pecam1*) were not altered (revised Extended data Fig. 8a&b; Figure 4 for reviewer).

Figure 4 for reviewer: qPCR analysis to monitor expression of *Epcam*, *Cdh1*, *Muc5ac*, *Vim*, *Fn1*, *Cdh2*, *Snail1*, *Snail2* in epithelial cells isolated from control and *Tet3^{smKO:T}* lungs; *Vwf*, *Cdh5*, *Pecam1* in endothelial cells isolated from control and *Tet3^{smKO:T}* lungs, 8 weeks after tamoxifen injection (n=3; two-tailed unpaired t-test: *p<0.05, **p<0.01)

We also investigated whether loss of *Tet3* in VSMCs outside the lung also results in activation of TLR-signaling. We observed a lower expression of *Tet3* in the aorta compared to the lung and no impact on the levels of proteins involved in cellular contractility (e.g. α -SMA and TAGLN) in *Tet3* single KO mice, whereas compound *Tet2/Tet3* mutant VSMCs in the aorta showed significantly lower levels of proteins characteristic for the contractile SMC phenotype (Figure 3a-c for reviewer). 5hmC levels were not changed after *Tet3* single knockout but significantly reduced in compound *Tet2/Tet3* mutant SMCs isolated from both aorta and intestine (Figure 3d&e for reviewer). In contrast to SMCs from *Tet3* KO

lungs, expression of interferon response related genes was not increased in SMCs from aortas, bladders and intestines, which are SMC enriched tissues out of lung (Figure 3f for reviewer). Furthermore, transfection of RNAs isolated from FACS-sorted mutant SMCs from aortas, bladders and intestines did not stimulate cytokine/chemokine production in recipient cells as RNAs from mutant lung SMCs (Figure 3g for reviewer). Taken together, our data indicate that loss of *Tet3* induces a switch from the contractile to synthetic phenotype only in SMCs of the lung but not in SMCs outside the lung. Concomitant inactivation of *Tet2* and *Tet3* further reduced the 5hmC content in *Tet3* KO VSMCs but not in *Tet3* KO BSMCs (revised Extended Data Fig. 2m). Substantial morphological changes were only present in the airways but not the vessels of *Tet3* and *Tet2/Tet3* mutant mice, which may be caused by a different threshold for TLR activation in VSMCs compared to BSMCs or different responses by neighboring cells. We reason that the switch from a contractile to a synthetic phenotype in lung VSMCs of *Tet3* mutants is a secondary effect caused by airway inflammation. The moderate reduction of proteins such as MYH11 and α -SMA, characteristic for the contractile phenotype, in VSMCs of the aorta in *Tet2/Tet3* mutant animals is apparently not sufficient to cause major vascular pathologies (Figure 3c for reviewer). We sincerely think that an in-depth analysis and description of the different possibilities is far beyond the scope of the current study and therefore included only parts of the new results in the current manuscript.

Specific Comments:

Results: line 103. Here the authors should indicate the age of the mice when the knockout was induced

Response: 8-week-old mice were used to induce *Tet3* knockout. We have indicated the age of the mice in revised manuscript (Figure legend of revised Extended data Fig.1f).

line 108: The epithelial morphological changes are presumably related to some TLR7/8 activity but is this substantiated immediately after tamoxifen?

Response: We characterized molecular/cellular changes of epithelial cells in *Tet3* mutant mice 4 or 8 weeks after TAM administration. We found significantly increased expression of interferon response related genes (i.e. *Irf7*, *Ifnb*) in epithelial cells 8 weeks but not 4 weeks after TAM treatment, which corresponds to changes in *Tet3* KO SMCs (revised Extended data Fig.8b), indicating paracrine effects resulting from activation of TLR7/8 signaling. It always takes some time for deletion to happen after activation of the Cre-recombinase. Thus, analysis “immediately” after tamoxifen administration will yield ambiguous results due to incomplete or low recombination rates. Moreover, increased expression of interferon response related genes were only observed 8 weeks but not 4 weeks after TAM administration, which deterred us from investigating even earlier time points.

It is curious that the only data on vascular smooth muscle cells relates to TET2. When SMC were isolated from the murine lungs there was no distinction. It is therefore important to show parallel photomicrographs of distribution of 5HmC and 4 HC in the pulmonary artery SMC, unless TET3 is uniquely present in airway SMC which would mean that in the artery TET2 does compensate. This needs to be clarified.

Response: We already presented EM images of VSMC in *Tet3* mutant mice in revised Extended Data Fig. 7b&c, which the reviewer may have overlooked. These data demonstrate a phenotype switch of both BSMCs and VSMCs to synthetic SMCs in *Tet3* mutant lungs. To follow the reviewer's specific request, we also monitored 5hmC in BSMC and VSMC by immunofluorescence and detected a significant reduction of 5hmC in both BSMC and VSMC after *Tet3* inactivation. Furthermore, we generated SMC-specific *Tet2/Tet3* compound mutant mice to address the question about a potential compensation of *Tet3* deficiency by *Tet2*. We found that *Tet2* depletion further reduced the 5hmC content in *Tet3* KO VSMCs but not in *Tet3* KO BSMCs. (revised Extended Data Fig. 2m). Despite the phenotype switch in VSMCs of the lung, we did not observe a major vascular pathology, which may be caused by a different threshold for TLR activation in VSMCs compared to BSMCs or different responses by neighboring cells (Please also see the response for major comment (v) above).

Additionally, we monitored 5hmC levels in SMCs in other tissues including the aorta and intestine. 5hmC levels in aortic VSMCs did not decline significantly after inactivation of *Tet3* alone, but was clearly reduced by concomitant loss of *Tet2* and *Tet3* (Figure 3d for reviewer). Similarly, a reduction of 5hmC was only present in intestinal SMCs of compound *Tet2/3* but not in single *Tet3* mutants (Figure 3e for reviewer). The reduction of 5hmC levels corresponded to reduced expression of α -SMA in aortic and intestinal VSMCs, which was only observed in *Tet2/Tet3* compound but not in *Tet3* single mutant mice (Figure 3c-e for reviewer), arguing for overlapping functions of *Tet2* and *Tet3* for maintaining the contractile phenotype of SMCs. Increased expression of innate immune response-related genes was only observed in SMCs of the lung but not in the aorta, bladder and intestine (Figure 3f for reviewer).

We are not sure what the reviewer wants to know about 4 HC. Presumably, 5mC is meant. As shown in the manuscript (revised Fig.1d, revised Extended data Fig.5d), we did not detect a significant change of the total 5mC content or at gene bodies of selected contractile genes in SMCs after inactivation of *Tet3*.

Lin 231: The discussion on TLR-7 and macrophages seems to come out of the blue. Is the same paradigm present in macrophages, i.e. do they also use TET3? This needs more in depth assessment. Alternatively, is there evidence that SMC are becoming macrophages as is discussed later in the Results section. Is TET3 the switch that keeps SMC from being synthetic or from being macrophages? This is not given extensive enough experimentation and may be a very important concept. Were there alterations in transcription factors that caused the switch in phenotype? Which ones? What confers specificity of TET3 at sites of H3K27trimethyl marks?

Response: SMC can be converted into macrophage-like cells in arteriosclerotic lesions. In a study published in Nature Medicine, it was reported that SMCs acquire properties of macrophages in a model of arteriosclerosis (Apoe^{-/-} mice)(Shankman et al., 2015). In our study, we not only detected a switch of SMCs to a more synthetic phenotype but also similar molecular changes as in the study by Shankman et al., including reduced expression of SMC marker proteins such as TAGLN, α -smooth muscle actin (α -SMA), and myosin heavy chain (MYH11). Furthermore, we saw increased expression of macrophage markers such as *Cd68*, *Lgals3* and *F4/80* in *Tet3*-deficient primary SMCs (revised Fig.4b). We also observed increased levels of EEA1 and RAB7, proteins regulating endosome trafficking that co-localize with TLR7 in macrophage-like *Tet3*-deficient SMCs (revised Extended data Fig. 6b). Any effects on bona fide macrophages in our model, described in (revised Fig. 6b, revised Extended data Fig. 8h), are most likely secondary, since inactivation of *Tet3* occurred specifically in SMCs and not in macrophages. We have changed 'stimulated macrophage' to 'macrophage-like *Tet3*-deficient SMCs' in the manuscript to clarify this issue.

Our RNA-seq data revealed that key transcription factors for contractile or synthetic gene transcription such as *Klf4* and *Myocd* remain unchanged, although the transcription factor *Srf* was slightly reduced in *Tet3*-mutant SMCs (revised Extended data Fig. 4l). The transcriptional activity of contractile genes was NOT reduced in *Tet3*-mutant SMCs, which clearly indicates that the reduced presence of contractile PROTEINS is not caused by down-regulation of the transcription of contractile genes, but by increased spurious transcription, which reduces generation of full-length transcripts and subsequent translation.

In the present study, we did not investigate the correlation of TET3 recruitment and H3K27me3. We assume the reviewer means H3K36me3. TET3 interacts with elongating Pol II and SETD2. SETD2 is found at gene bodies and mediates local deposition of H3K36me3 (Revised Fig.2e &g).

Line 289:

Is there heightened proliferation in bronchial smooth muscle as in asthma?

Response: We performed Ki67 staining and did not detect an increase of BSMC proliferation (revised Extended data Fig. 7g). However, we detected increased number of senescent cells in airway SMC layer and enhanced expression of senescence marker genes *Cdkn1a* and *Cdkn2a* in mutant lungs (revised Extended Fig. 7h&i). Cellular senescence and subsequent acquisition of a senescence-associated secreted phenotype (SASP) has been suggested to play an important role for a specific subgroups of asthma patients e.g. adult-onset asthma (Wang et al., 2020). Since *Tet3*-deficient SMCs produce several cytokines characteristic for SASP, including IL1b, we reason that loss of *Tet3* in bronchial SMCs will cause senescence instead of proliferation.

The photomicrographs don't appear to show either an increase in SMC or in fibrosis so most of the impact of the inflammatory response appears to be paracrine on epithelial cells presumably via cytokines.

Response: The reviewer is absolutely right. We only detected an increase in fibrosis 6 months after *Tet3* inactivation, suggesting paracrine effects on neighboring cells.

In general, more data are needed to establish the model and why airway not arterial SMC are affected and whether there is a morphogenesis to a macrophage like cell or to a proliferating SMC. We presume that the general elevation in cytokines are inducing the changes in epithelial cells but this should be studied in greater depth.

Response: As described in the response to major comment (v), we made major efforts to analyze the consequences not only of a loss of *Tet3* but also of a combined loss of *Tet2* and *Tet3* on airway and arterial SMCs. We hope the reviewer agrees that a detailed analysis of why airway SMCs react differently than arterial SMCs is beyond the scope of the current study. We discuss several possibilities but can certainly not follow them up right now. We agree that the Th2-cell based immune response is a secondary phenomenon. In the revised manuscript, we describe changes of epithelial cells in mutant lung over time and detected increased expression of EMT-related genes and elevated production of cytokine/chemokine by epithelial cells (Revised Extended data figure 8a&b). We also found that conditional medium from *Tet3* KO SMCs stimulates expression of EMT related genes and production of cytokines/chemokines in epithelial cells. Furthermore, we demonstrate that the paracrine effects of mutant SMCs on epithelial cells are TLR7-dependent (Revised Extended data fig.8c). All our new results point to the fact that the innate immune responses initiate from *Tet3*-deficient BSMCs, which is supported by the RNA transfection experiments and the RNA-seq data, demonstrating increased expression of a numerous of cytokine genes in SMCs after inactivation of *Tet3*.

As pointed out above, we do not see an increase of SMC proliferation after *Tet3* inactivation (Revised Extended data fig.7g). We think that we already provide compelling evidence that SMCs acquire a macrophage-like phenotype after *Tet3* inactivation.

Reviewer #2:*Remarks to the Author:*

Wu et al. used a conditional knockout strategy to target exon 10 of the Tet3 gene in a mouse model, aimed at removing the catalytic domain. This enzyme creates 5-hydroxymethylcytosine (5hmC) from 5mC in gene bodies and other regions and may cause DNA demethylation. They observed that Tet3 inactivation produced a phenotype in smooth muscle cells, in particular in the lung. They claim that loss of TET3 allows the formation of spurious intragenic transcripts in more highly expressed genes and activates the endosomal nucleic acid-sensing TLR7/8 signaling pathway. The immune response induces inflammation in the lung and airway remodeling. Although such findings would be of interest, both from an epigenetic regulatory and lung biology perspective, the conclusions of the authors are in many instances not supported by the data. I have the following major concerns:

Response: We thank the reviewer for careful evaluation of the manuscript.

1) The knockout has not been verified properly. Although there is an RT-PCR assay for Tet3 (Fig. Ext Data 1), no Western blot and not even a PCR spanning exon 10 in the targeted cell population has been shown.

Response: We have tried extensively to detect endogenous TET3 protein by western blot analysis but the available commercial antibodies are not of sufficient quality. However, in addition to the RT-PCR mentioned by the reviewer, we detected depletion of TET3 signals by immunofluorescence staining in SMC layers in *Tet3* knockout (KO) lungs and aortas compared to controls (revised extended data Fig. 1i; Fig. 1a for reviewer). We also provide a snapshot view of RNA-seq from *Tet3*-deficient SMCs, demonstrating efficient downregulation of *Tet3* transcription (Fig. 1b for reviewer) and results from genomic PCRs using primers spanning the floxed exon 10, showing efficient excision of floxed exon 10 in *Tet3* KO SMCs (revised extended data Fig. 1g). These data clearly demonstrate successful inactivation of the *Tet3* gene in SMCs. Furthermore, we respectfully would like to point out the strong loss of 5hmC in SMCs, which is also a clear indicator of inactivation of the *Tet3* gene (revised Fig. 1d).

Figure 1 for reviewer: a, Immunofluorescence staining for α -SMA and TET3 on cryosections from control and Tet3^{smKO} aortas, 8 weeks after tamoxifen injection (n=3). DNA was stained by DAPI. Scale bar: 50 μ m. b, Integrated Genome Viewer (IGV) tracks displaying the RNA-seq peaks over *Tet3* exon 10 (n=2).

2) The *Tet3-lacZ* expression data is strange. The authors state that expression is confined to smooth muscle cells (SMCs) in adult mice. According to ENCODE and GTEX (human) data, however, this gene is expressed in almost all adult tissues or cell types.

Response: We understand the reviewer's concern. We do not want to claim that expression of *Tet3* is exclusively restricted to SMCs. We apologize if we were not precise enough to convey this message. The *Tet3-lacZ* reporter indicates that expression of *Tet3* in adult organs of mice is the highest in SMCs, which does not exclude expression in other cell types. Other methods, such as RT-PCR or deep RNA-sequencing are certainly more sensitive. We performed RTqPCR expression analysis of *Tet3*, using purified SMCs, epithelial cells, endothelial cells and inflammatory cells in the lung. The results demonstrate that *Tet3* has the highest expression level in lung SMCs, which confirms the results from the *Tet3-lacZ* reporter (Fig. 2 for reviewer). Furthermore, the *Tet3-lacZ*-signals were more widespread (essentially ubiquitous) during embryonic development but then became more restricted to SMCs (revised Extended data Fig.

1b). We do not have any reason to question the validity of the *Tet3-lacZ* reporter.

Figure 2 for reviewer: RT-qPCR analysis to monitor expression of *Tet3* in epithelial cells (EpC), endothelial cells (EdC), inflammatory cells (IC) and isolated SMCs in the lung tissues (n=3; one-way ANOVA with Tukey's post hoc test: *p<0.05, **p<0.01, ***p<0.001).

3) The authors state that the concentration of functional full-length mRNAs dropped after *Tet3* inactivation. How was this determined or verified?

Response: We apologize that we did not make this clearer. We performed RT-PCR analysis of full-length transcripts and detected reduced concentrations of full-length mRNA from highly transcribed genes such as genes coding for the contractile apparatus (*Acta2*, *Cnn1* and *Arhgap18*). The results were compared to the number of RNAseq-reads obtained with an Illumina machine (revised Fig. 3e; Fig. 5d and revised Extended data Fig. 4j&k). Furthermore, we found decreased concentrations of proteins characteristic for the contractile phenotype of SMCs by western blot analysis (revised Fig. 5b). We changed the description to make that clearer.

4) The causality of airway inflammation due to aberrant spurious transcription is not proven.

Response: We agree that causality of spurious transcription and airway inflammation in vivo is circumstantial. However, we provide clear experimental evidence that inactivation of *Tet3* in SMCs activates innate immune responses leading to airway inflammation, that activation of innate immune responses is conferred by RNAs, and that activation of innate immune responses is tightly linked to spurious transcription. We have expanded our efforts to establish causality and now present five different lines of evidence supporting the conclusion that spurious transcripts in *Tet3*-mutant SMCs activate innate immune responses and thereby initiate airway inflammation: 1.) Transfection of RNA extracted from *Tet3*-mutant but not from WT SMCs into recipient cells induces production of various cytokines in recipient cells. 2.) RNA extracted from *Tet3* deficient SMCs failed to activate interferon responses after treatment of recipient cells with an TLR7 inhibitor (E6446) (revised Fig. 4d). 3.) TLR7 inhibitor treatment prevented augmented expression of downstream cytokine/chemokine genes in *Tet3* knock down (KD) SMCs (revised Extended data Fig. 6h). 4.) TLR7 inhibitor treatment prevented augmented expression of downstream cytokine/chemokine genes after transfection of RNA extracted from *Tet3* KD SMCs (revised Fig. 4g). 5.) Conditioned medium from *Tet3* KO SMCs stimulated expression of EMT related genes and production of cytokines/chemokines in airway epithelial cells (revised

Extended data Fig. 8c). We also show that the paracrine effects of mutant SMCs on epithelial cells are TLR7-dependent (Fig. 3 for reviewer). Overall, these findings strongly demonstrate that aberrant spurious transcripts (ssRNAs) in *Tet3* KO/KD SMC are recognized by TLR7-signaling, which plays a key role in facilitating expression of downstream cytokine/chemokine genes. Synthetic/secretory *Tet3*-deficient SMCs elicit paracrine effects on the neighboring airway epithelial cells, leading to EMT, enhanced cytokine production and airway remodeling in mutant lung, followed by Th2-based immune responses (revised Fig. 6a&b; revised Extended data Fig. 8b-e).

Figure 3 for reviewer: Upper panel: Outline of the experiment. MLE12 cells were cultured with conditioned medium from mES-derived SMCs, collected after treatment and subsequent removal of the inhibitor. Lower panel: RT-qPCR analysis of *Epcam*, *Cdh1*, *Fn1*, *Vim*, *Il6*, *Il1b*, *Ifnb* expression in MLE12 cells cultured with conditioned medium from scramble, *Tet3*^{KD}, *Tet3*^{KD}+*Tet3*^{WT}, *Tet3*^{KD}+*Tet3*^{CD} mES-derived SMCs that had been previously treated or not with the TLR7 inhibitor (TLR7in) E6446 (n=3; one-way ANOVA with Tukey’s post hoc test: *p<0.05, **p<0.01, ***p<0.001, ****p<0.0001).

We indeed do not know whether spurious transcripts directly activate TLR7 or whether additional steps are involved but that does not diminish the value of our observation that aberrant transcripts from *Tet3*-mutant SMCs activate inflammatory reactions. Further studies will reveal the molecular mechanisms by which aberrant transcripts in *Tet3*-mutant SMCs activate TLR7 but we sincerely think that this is beyond the scope of the current study.

5) Figure 1f: *5hmC* is not really increased in gene bodies. Is the dip near the TSS related to high GC-content, which is causing a reduced PCR efficiency over these regions?

Response: It has been reported that the 5hmC content is lower at transcriptional start sites (TSSs) in mammalian genomes compared to promoters and gene bodies (Cui et al., 2020; Pastor et al., 2011; Song et al., 2011). In our study, we wanted to emphasize that 5hmC is enriched in gene bodies relative to intergenic regions, particularly in the top 25% of highly transcribed genes. To obtain a better view of the genome-wide distribution of 5hmC, we extended the intergenic regions in the coverage profiles of 5hmC to \pm 10kb of TSS and TES, respectively. In addition, we integrated the coverage profiles of 5hmC in actively transcribed genes (top 75% genes based on RNA-seq data) into one plot. We observed a clear accumulation of 5hmC within gene bodies specifically in the group of top 25% highly transcribed genes when we evaluated the ratios but not at absolute levels of 5hmC in gene bodies compared to either the intergenic regions in the same gene group or to other gene groups with lower transcription activity (revised Fig. 1f; revised Extended data Fig. 3c). In our view, it is not meaningful to evaluate the enrichment without a reference point.

In addition to our study, several other reports described depletion of 5mC and 5hmC in TSS regions of actively transcribed genes, e.g. in human and mouse ES cells, mouse neural progenitor cells, neurons and the cerebellum (Greco et al., 2016; Pastor et al., 2011; Tan et al., 2013). Since the extend of the dip within control SMCs follows transcriptional activity from Top25%>25%-50%>50%-75%>Bottom 25%, we assume that actively transcribed genes are hypo-methylated in the TSS region, which is required to form nucleosome-free region allowing initiation of transcription, but also show low 5hmC levels. To comply with the reviewer's request, we analyzed the GC content for gene groups according to the quartile based of RNAseq expression levels in control lung SMCs. We detected a positive correlation of the depth of the dip with GC content, unveiling that the degree of the dip near the TSS is related to the degree of the GC-content (Fig. 4 for reviewer). This observation is consistent with a previous study, in which the nucleosome free regions bound by Pol II in actively transcribed gene promoters are GC/CpG-enriched and nucleosome density in CGI-containing promoters is negatively correlated to GC/CpG content (Fenouil et al., 2012). Fenouil et al. also sequenced sonicated genomic DNA and demonstrated that the sequencing bias in promoters containing the highest CpG content is rather low.

Figure 4 for reviewer: a, Overlap of hMeDIP-seq and RNA-seq datasets from sorted lung SMCs of control and $Tet3^{smKO:T}$ mice ($n=2$). b, GC content calculated for different gene sets based on RNA-seq as in revised Fig. 1f ($n=2$).

6) *Figure 2e: It is hard to interpret the data without an IgG control in the IP.*

Response: We apologize for this omission. We have added the IgG control to the IP experiments in (revised Fig. 2e) as requested.

Figure 2g is not convincing.

Response: To make (Fig. 2g) more convincing, we repeated and extended the co-IP experiment by overexpressing wild-type or catalytic inactive TET3. In the new experiments, which were carefully quantified, we precipitated similar amount of endogenous SETD2 in all samples. We found that only expression of TET3 WT but not catalytically inactive TET3 enhances the interaction between SETD2 and Pol II (revised Fig. 2g), suggesting that TET3-mediated 5hmC formation but not a potential non-catalytic function TET3 stabilizes the interaction between SETD2 and Pol II in SMCs.

7) *Figure 2h shows some Q-PCR data for H3K36me3 but it is hard to make a case for H3K36me3 changes based on this limited information.*

Response: In addition to the ChIP-qPCR experiments, we now did H3K36me3 ChIP-seq, which revealed that H3K36me3 deposition is reduced in $Tet3$ -deficient primary SMCs (revised Extended data Fig.3h). Integrated analysis of H3K36me3 ChIP-seq, RNA-seq and Pol II pSer5 ChIP-seq datasets demonstrate

that H3K36me3 levels drop dramatically in highly transcribed genes, which is associated with increased intragenic Pol II entry after *Tet3* inactivation (revised Fig. 2h&i). Furthermore, gene tracks showing H3K36me3 deposition within *Acta2* and *Myh11* genes (revised Fig. 3g) demonstrate that *Tet3* inactivation concomitantly attenuates H3K36me3 signals and 5hmC deposition in the vicinity of CAGE-TSSs.

8) *The combined data in Figure 3 do not make a strong case for the conclusion that TET3 prevents intragenic transcription initiation.*

Response: To support our conclusion that TET3 prevents intragenic transcription initiation, we performed H3K36me3 ChIP-seq and analyzed the genome-wide correlation between H3K36me3 deposition and intragenic transcription initiation caused by loss of *Tet3*. Our results clearly demonstrate that inactivation of *Tet3* significantly reduce H3K36me3, a key histone mark preventing intragenic spurious transcription, particularly in gene bodies of highly transcribed genes, resulting in global drop of H3K36me3 in protein level (revised Fig. 2h; revised Extended data Fig. 3g). H3K36me3 level drop predominately in those genes, in which intragenic Pol II pSer5 entry increases after *Tet3* inactivation (revised Fig. 2i). Moreover, gene tracks showing H3K36me3 signals within gene bodies of contractile genes i.e. *Acta2* and *Myh11* reveal that genomic regions containing intragenic CTSSs (TSSs identified by CAGEseq) concomitantly lose H3K36me3 and 5hmC after *Tet3* depletion (revised Fig. 3g). Altogether, our results strongly demonstrate that TET3 prevents intragenic transcription initiation by facilitating SETD2-mediated H3K36me3 deposition, thereby precluding Pol II binding at cryptic TSSs.

Furthermore, to validate the hypothesis that *Tet3* mediated 5hmC but not a potential noncatalytic function of *Tet3* prevents spurious transcription, we performed rescue experiment by expressing wild-type or catalytic dead human TET3 in *Tet3* KD mouse embryonic stem cell (mESCs)-derived SMCs. Efficient depletion of *Tet3* in ES cell-derived SMCs led to a similar phenotype as seen in *Tet3* knockout primary SMCs, including reduction of 5hmC and H3K36me3, impaired full-length mRNA transcription but enhanced presence of transcripts from intermediate exons of highly expressed contractile (*Acta2*, *Cnn1*, *Myh11*), reduced protein levels of contractile (α -SMA, CNN1, MYH11) but not synthetic proteins, and specific binding of Pol II-pSer5 in the vicinity of CTSSs (revised Fig. 4e, f; revised Fig. 5c-f, revised Extended data Fig. 6g). mRNA transfection analysis demonstrates that RNA from *Tet3* KD but not from control SMCs activate the endosomal TLR7 signaling pathway (revised Fig. 4d). Expression of wild-type but not a catalytically dead version of TET3 revokes all aspects of the phenotype initiated by inactivation *Tet3* in SMCs (revised Fig. 4f, g; revised Fig. 5c-f; revised Extended data Fig. 6g). Gene tracks of Nano-seal sequencing discovered reduced 5hmC levels, matching the decline of H3K36me3 in the vicinity of intragenic CTSSs in *Acta2* and *Myh11* genes after *Tet3* depletion (revised Fig. 3g).

We hope the reviewer agrees that these new data provide strong experimental evidence for our conclusion that TET3 prevents intragenic initiation of transcription.

9) *Figure 4: It is not clear that one can claim from an RNA transfection experiment that spurious transcripts would activate the TLR7 signaling pathway.*

Response: RNA transfection experiments have been instrumental in the past to investigate the ability of aberrant RNAs to activate innate immune responses (Kariko et al., 2005). We therefore think that it is perfectly legitimate to use RNA transfections to study immunogenic properties of RNAs. To substantiate our claim that spurious transcripts activate the TLR7 signaling pathway, we treated ESC-derived SMCs after knockdown of *Tet3* with a TLR7 inhibitor. We observed reduced expression of downstream cytokine/chemokine genes in *Tet3* KD SMCs after TLR7 inhibitor treatment (Extended data Fig. 6h). Furthermore, transfection of RNA extracted from both *Tet3* KO and KD SMCs stimulated the TLR7-signaling pathway including expression of cytokines in recipient cells, which was absent when recipient cells were treated with a TLR7 inhibitor (revised Fig. 4d&g). These findings indicate that TLR7-signaling plays a key role in expression of downstream cytokine/chemokine genes. As pointed out above (response to comment #4), we think that it is outside the scope of this study to interrogate in detail the molecular mechanisms by which aberrant transcripts in *Tet3*-mutant SMCs activate TLR7.

10) *Figure 5b: There are large differences in band intensities between individual samples, even in the control groups, which makes it difficult to have confidence in the conclusions.*

Response: We thank the reviewer to point out this weakness. We have redone the WB of α SMA, MYH11 and TAGLN and have carefully quantified the results (revised Fig. 5b, Extended data Fig. 7f).

11) *I am not sure that the authors performed strand-specific RNA sequencing. Finding transcripts on the opposite strand in gene bodies would provide much better and more convincing evidence for the model the authors are proposing.*

Response: The CAGE-seq analysis allowed detection of RNA transcription from both sense and anti-sense strands. We identified much more spurious transcripts from the sense-strand (2114 genes) than from the antisense-strand (904 genes) in *Tet3*-deficient SMCs. Only 10% of genes contained spurious transcripts from both sense- and antisense-strands. The preference for the sense-strand is higher than in yeast, in which DNA methylation does not occur and ectopic initiation normally happens on both the sense and antisense strands (Hennig and Fischer, 2013). In mammalian cells, the distribution of 5hmC is strongly asymmetric (Yu et al., 2012) with significantly higher levels on the sense strand, suggesting an association of 5hmC accumulation with transcriptional orientation (Wen et al., 2014). The asymmetric character of 5hmC may also explain the preferential occurrence of cryptic transcription on the sense strand after loss of 5hmC in *Tet3* mutant SMCs.

GO term enrichment analysis revealed that genes with a significant increase in sense transcripts were largely correlated to SMC cell identity, while genes with antisense transcripts have no obvious GO term

enrichment in *Tet3* mutant SMCs. We speculate that the increased anti-sense RNA transcription is caused by secondary effects. Therefore, we focused on genes with spurious transcripts from the sense-strand in our study. We have clarified this in the revised manuscript.

12) *hMeDIP is a procedure fraught with potential artifacts due to the binding of the antibody to certain repetitive DNA sequences. Have the authors confirmed their results with an alternative method?*

Response: The reviewer is correct that due to the intrinsic affinity of IgG for short unmodified DNA repeats, some sequences may be falsely assigned to carry 5hmC. This is an intrinsic problem of all DIP-seq methods (discussed by Lentini et al., Nature Methods 2018: A reassessment of DNA-immunoprecipitation based genomic profiling). We do understand these concerns but would like to emphasize that 5hmC MeDIP-seq has been used in many different studies and in many different labs.

In addition, we have taken numerous quality measures to address this issue: **(i)** We have carefully re-analyzed our data by comparing input and 5hmC-MeDIP-seq data. We found that the coverage profiles of our input samples are very similar to coverage profiles from published IgG samples (Fig. 5 for reviewer). Since our 5hmC-MeDIP-seq data were normalized to input, potential artifacts resulting from repetitive sequences should be excluded. **(ii)** We exclusively focus on predefined gene sets and promoters and do not consider peaks. **(iii)** We deduplicated reads and reads mapping more than once in the genome were discarded (star – outFilterMultimapNmax parameter is set to 1). Thus, our results will not be affected by repetitive regions, since such regions were removed from the results.

Figure 5 for reviewer: **a**, Coverage profile of 5hmC enrichment in Input (hMeDIP-seq) and IgG (SRR3586799) samples. **b**, Coverage profile of 5hmC enrichment in gene groups defined in Fig. 1f in Input (hMeDIP-seq) and IgG (SRR3586799) samples.

To further address the reviewer's comments, we **(iv)** performed Nano-5hmC-Seal-sequencing (Nano-seal), a non-antibody-based technique which avoids the pitfalls inherent to all DIP-seq methods. We

obtained similar results with Nano-seal sequencing for the distribution of 5hmC when compared to 5hmC MeDIP-seq, both for gene bodies of the whole genome (revised Extended data Fig. 3a) and for genes grouped in different quartiles (revised Fig. 1f). The new results are shown in revised Extended data 3b&c. Gene tracks of Nano-seal also clearly demonstrate that 5hmC enrichment in the vicinity of CTSSs within gene bodies of *Acta2* and *Myh11* genes are substantially reduced after *Tet3* inactivation (revised Fig. 3g). Taken together, we do not think that inclusion of some short unmodified DNA repeats will change results dramatically and/or affect our conclusions, although we cannot completely rule out some contamination by repetitive regions in our 5hmC-MeDIP-seq data.

13) *I am not convinced that reduced 5hmC formation after inactivation of TET3 will lead to reduced levels of H3K36me3.*

Response: To further substantiate our conclusions, we monitored H3K36me3 levels in control and *Tet3*-mutant SMCs by western blot analysis. We found that inactivation of *Tet3* (*Tet3* KO&KD) dramatically reduced the content of H3K36me3 in SMCs (revised Extended data Fig. 3g&revised Extended Fig. 6g). Furthermore, we successfully performed H3K36me3 ChIP-seq, which indicates that H3K36me3 deposition is clearly reduced in *Tet3*-deficient primary SMCs (Extended data Fig. 3h). Integrated analysis of H3K36me3 ChIP-seq, RNA-seq and Pol II pSer5 ChIP-seq the data show that H3K36me3 levels drop dramatically in highly transcribed genes, facilitating intragenic Pol II entry after *Tet3* inactivation (revised Fig. 2h&i). Gene tracks of H3K36me3 within *Acta2* and *Myh11* genes reveal concomitant reduction of H3K36me3 signals and 5hmC deposition in the vicinity of intragenic CTSSs (revised Fig. 3g).

14) *The authors propose that ssRNA degradation products may be recognized by TLR7/8. However, what about ds RNA, perhaps in analogy with retroelement transcripts observed after treatment of cells with 5-aza-cytidine?*

Response: dsRNA is not recognized by TLR7/8 but by TLR3 and RIG1. We did not obtain evidence that these genes and their downstream effectors were affected after the loss of 5hmC (Fig. 6 for reviewer). In addition, inhibition of TLR7 prevents expression of inflammatory cytokine/chemokines in *Tet3* KD SMCs and in recipient cells transfected with RNA from *Tet3*-deficient SMCs (both *Tet3* KO and *Tet3* KD SMCs) (revised Fig. 4d&g; revised Extended data Fig. 6h), indicating that the innate immune responses triggered by inactivation of *Tet3* are dependent on TLR7.

Figure 6 for reviewer: Expression of *Ddx58*, *Ifih1*, *Traf3* (*RIG1* and their downstream effector genes) and *Tlr3* genes based on RNA-seq data of sorted control and Tet3^{smKO:T} lung SMCs (n=2; two-tailed unpaired t-test).

Minor points:

Figure 1b is not labeled correctly. It seems to show different organs.

Response: We are sorry for the confusion. Fig. 1b shows H&E staining of mouse lung, heart and aorta. The different organs were correctly indicated on the right side of images in the original figures. We now placed the labels for the different organs on the left side of the panels, which should avoid any confusion (revised Fig. 1b).

The Discussion section (lanes 307-312) contains several unsupported speculative sentences.

Response: We have changed the text of the discussion to avoid unsupported speculations.

Reviewer #3:*Remarks to the Author:*

In this manuscript, Wu et al. propose a novel role for Tet3 in preventing spurious transcription and an innate immune response in lung SMCs. The authors show that Tet3 preserves the integrity of transcription through interaction with RNA polymerase II. In addition, authors show that loss of Tet3 stimulates an innate immune response that recruits immune cells to the lung. Although the manuscript is conceptually intriguing, there are some concerns outlined below.

Response: We thank the reviewer for the careful reading of the manuscript and the helpful comments.

Major:

1. The title of the manuscript implies that 5hmC prevents spurious transcription. A large proportion of the work focuses on TET3's association with RNA polymerase. More work is required to determine if it is 5hmC or Tet3 that is important for spurious transcription:

Response: The reviewer brings up an important point. In the revised version, we performed rescue experiments using catalytically active and inactive TET3 to answer this question. First, we confirmed by dot blot analysis that expression of wild-type but not catalytic inactive TET3 enhances the overall content of 5hmC (revised extended data Fig. 3f). The dot blot results also exclude a dominant-negative effect of catalytic inactive TET3 on 5hmC deposition. To examine whether the catalytical activity of TET3 is critical for stabilizing the interaction between SETD2 and Pol II, we examined the capability of catalytically inactive TET3 to form a complex with Pol II and to enhance SETD2-Pol II interactions. Co-IP experiments revealed that the interaction of catalytically inactive TET3 with Pol II Ser2 and SETD2 is significantly lower compared to catalytically active TET3 (revised Fig. 2e). In addition, only expression of catalytically active but not inactive TET3 enhances the interaction between SETD2 and Pol II and stabilizes the complex (revised Fig. 2g).

We also investigated whether catalytically inactive TET3 prevents the cellular phenotype caused by loss of 5hmC in SMCs. Since the in vitro culture of primary lung SMCs are not suitable for such studies and commercially available human pulmonary smooth muscle cells show a predominately synthetic phenotype, we have established a system to generate differentiated SMCs from mouse embryonic stem cells. *Tet1* expression was completely absent in mESC-derived SMCs and *Tet2* expression was much lower compared to mESCs, whereas *Tet3* was robustly expressed, much stronger than in mESC (revised Extended Fig. 1c-e). We successfully knocked down expression of *Tet3* (*Tet3* KD) in mESC-derived SMCs, which dramatically reduced 5hmC levels in SMCs. Expression of human wild-type but not catalytically inactive TET3 restored 5hmC levels (revised Fig. 4e). mESC-derived WT SMCs showed typical features of contractile SMCs (revised Extended Fig. 1c-e), but depletion of *Tet3* induced a contractile to synthetic phenotype switch, similar to primary *Tet3* knockout (KO) SMCs. We observed a strong reduction of

5hmC and H3K36me3 in *Tet3* KD mESC-derived SMCs, impaired full-length mRNA transcription and enhanced intermediate exon transcription of highly expressed contractile (*Acta2*, *Cnn1*, *Myh11*), reduced protein levels of contractile (α SMA, CNN1, MYH11) but not synthetic proteins, and specific binding of Pol II-pSer5 in the vicinity of CTSSs of highly expressed contractile genes (revised Fig. 4e, f; revised Fig. 5c-f, revised Extended data Fig. 6g). mRNA transfection analysis demonstrated that RNA from *Tet3* knockdown but not from control SMCs activated the endosomal TLR7 signaling pathway (revised Fig. 4d). Importantly, expression of wild-type but not a catalytically inactive version of human TET3 prevents all effects caused by inactivation of *Tet3* (revised Fig. 4 f, g; revised Fig. 5c-f; revised Extended data Fig. 6g). We sincerely think this new suite of experiments clearly demonstrates that TET3-mediated 5hmC formation is crucial for prevention of spurious transcription, but not a non-enzymatic function of TET3.

a. How does TET3's genomic localization change before and after DRB in SMCs.

Response: We attempted to monitor the genomic localization of TET3 before and after DRB in SMCs by ChIP-qPCR using numerous different TET3 antibodies. However, we failed despite considerable efforts. Therefore, we expressed HA-tagged TET3 in ES cell-derived SMCs and performed ChIP-qPCR with HA antibodies with or without DRB treatment. DRB treatment did not affect the global level of 5hmC (Fig. 1a for reviewer). ChIP-qPCR results revealed that HATET3 was recruited to different target genes such as *Acta2*, *Myh11* and *Cnn1* but not *Rbp1* or *Vim*. Recruitments were not altered by the DRB treatment (Fig. 1b for reviewer). Of course, expression of exogenous is less convincing than analysis of endogenous TET3, but we see no other way to solve the technical difficulties. These results are shown only to the reviewer, since we do not believe that outcome has a major impact for the study.

Figure 1 for reviewer: a, Dot blot analysis of 5hmC levels using genomic DNA from *in vitro* differentiated Tet3^{KD} SMCs that overexpress human Tet3^{WT} after treatment with either DMSO or DRB (n=3). Methylene blue staining served as loading control. b, ChIP-qPCR to monitor HA enrichment within gene-bodies of indicated genes in *in vitro* differentiated Tet3^{KD} SMCs that overexpress human Tet3^{WT} after treatment with either DMSO or DRB (n=4 two-tailed unpaired t-test). Relative fold-changes compared to IgG control are indicated by a dash line.

b. Use changes in 5hmC to classify spurious transcripts.

Response: We now used changes in 5hmC to classify spurious transcripts as requested by the reviewer (revised Extended data Fig. 4h). To generate meaningful results, the analysis was limited to genes containing a high content of 5hmC in control cells, so that a clear reduction after inactivation of *Tet3* was measurable. The data demonstrate that spurious transcripts are mainly generated from genes that show a high 5hmC content under WT conditions but a strong reduction of 5hmC after *Tet3* inactivation. Furthermore, we did a similar analysis using newly obtained data from the Nano-5hmC-seal (Nano-seal) analysis, demonstrating a genome-wide reduction of 5hmC levels, which was particularly evident within gene bodies of spuriously expressed genes (revised Extended data Fig. 5b).

c. Show that combined loss of Tet2 and Tet3 lowers 5hmC.

Response: To investigate whether the combined loss of *Tet2* and *Tet3* increases the loss of 5hmC and aggravates the lung phenotype, we generated SMC-specific *Tet2/Tet3* compound mutant mice. Concomitant deletion of *Tet2* and *Tet3* further reduced 5hmC levels specifically in pulmonary vascular SMCs (VSMCs) but not in lung bronchiolar SMCs (BSMCs) compared to single *Tet3* mutant mice, indicating that *Tet2* may partially compensate for the loss of *Tet3* in VSMCs but not in BSMCs (revised Extended data Fig. 2m). Notably, the presence of *Tet2* in *Tet3*-deficient pulmonary VSMCs did not prevent the SMC phenotype switch caused by *Tet3* inactivation, which is probably the consequence of inflammatory processes initiated in BSMCs.

In our view, the most likely possibility for the relatively normal morphology and function of the vasculature in *Tet3* mutant lungs is that BSMCs are more vulnerable to the loss of 5hmC than VSMCs. We do not know whether this is due to a lower rate of spurious transcription in VSMCs compared to BSMCs or a lower threshold of BSMCs to activate innate immune responses as an evolutionary selected mechanism against viral infections.

5hmC levels did not significantly decline in SMCs of the aorta and intestine after inactivation of *Tet3*, but dropped significantly after concomitant inactivation of *Tet2* and *Tet3* (Figure 2a&b for reviewer). Consistently, reduced expression of α -SMA in aortic and intestinal SMCs was only observed in compound *Tet2/Tet3* but not in *Tet3* single mutant mice (Figure 2a-d for reviewer), arguing for overlapping functions of TET2 and TET3 for maintaining the contractile phenotype in some SMC types. Concomitant inactivation of *Tet2* and *Tet3* did not reduce the 5hmC content in BSMCs compared to *Tet3* inactivation alone (revised Extended Data Fig. 2m). The data indicate that 5hmC levels are regulated in a different manner in SMCs of different organs. Due to space limitations, we cannot not show these results in the revised manuscript.

Figure 2 for reviewer: **a**, Immunofluorescence staining for α -SMA and 5hmC on paraffin sections from control, Tet3^{smKO} and Tet2/Tet3^{smKO} aortas, 8 weeks after tamoxifen injection (n=3). Quantification of mean fluorescence intensity (MFI) of 5hmC was performed by Image J and is shown in the right panel (n=3; one-way ANOVA with Tukey's post hoc test: **p<0.01). DNA was stained by DAPI. Scale bar: 50 μ m. **b**, Immunofluorescence staining for α -SMA and 5hmC on paraffin sections from control, Tet3^{smKO} and Tet2/Tet3^{smKO} intestines, 8 weeks after tamoxifen injection (n=3). Quantification of mean fluorescence intensity (MFI) of 5hmC was performed by Image J and is shown in the right panel (n=3; one-way ANOVA with Tukey's post hoc test: *p<0.05). DNA was stained by DAPI. Scale bar: 50 μ m. **c**, Western blot analysis of α -SMA, TAGLN in aortas from control and Tet3^{smKO} lungs, 8 weeks after tamoxifen injection. Pan-actin was used as loading control. **d**, Immunofluorescence staining for α SMA and MYH11 on paraffin sections from control and Tet2/Tet3^{smKO} aortas, 8 weeks after tamoxifen injection (n=3). DNA was stained by DAPI. Scale bar: 50 μ m.

d. Show gene tracks with RNA-seq, Cage-seq, 5-hmC. If indeed intragenic 5hmC is altering transcription, it is important to show a version of Fig. 3g with CMS-seq.

Response: We thank the reviewer for this constructive suggestion. We have consulted with Prof. Anjana Rao, an eminent expert in the field. She did not really recommend CMS-seq, because rather large quantities of DNA are required, which are difficult to obtain from sorted SMCs. We therefore decided to perform Nano-Seal, and used Nano-seal 5hmC sequencing data to analyze the distribution of 5hmC both for gene bodies of the whole genome (revised Extended data Fig. 3a) and for genes grouped in different quartiles (revised Fig. 1f). The new results are shown in (revised Extended data Fig. 3b&c). As requested by the reviewer, we show gene tracks of Nano-Seal, RNA-seq and CAGE-seq in (revised Fig. 3g). We have also included gene tracks of H3K36me3 ChIP-seq, which demonstrates substantially

reduced 5hmC and H3K36me3 signals in the vicinity of CTSSs within gene bodies of *Acta2* and *Myh11* genes after *Tet3* inactivation.

2. Rescue experiments

a. *Re-express wildtype and catalytically inactive TET3 in SMCs to evaluate changes in 5hmC, gene expression, and spurious transcripts.*

Response: We welcome this suggestion. Please refer to the answer for comment #1, where we describe in detail the experimental strategy and the new results.

b. *Show tracks of specific loci that are rescued by either wildtype or catalytically inactive TET3.*

Response: We have performed Pol II pSer5 ChIP-qPCR with chromatin isolated from *Tet3* knockout down (*Tet3* KD) SMCs derived from mESCs, in which wild-type or catalytically inactive human TET3 were expressed, to monitor the binding of Pol II pSer5 within gene bodies of several selected contractile (i.e. *Acta2*, *Myh11*, *Cnn1*) and synthetic genes (i.e. *Vim*, *Tpm4*). Similar to *Tet3* knockout in primary lung SMCs, *Tet3* KD increases intragenic entry of Pol II pSer5 in contractile genes. Expression of wild-type but not catalytically inactive human TET3 prevents enhanced intragenic entry of Pol II pSer5 in the vicinity of CTSSs and normalized elevated transcription of intermediate exons within highly transcribed contractile genes in *Tet3* KD SMCs (revised Fig. 5e&f).

3. *Authors state that there are no evident abnormalities in SMC containing organs other than the lung. A change in weight is observed after tamoxifen injection.*

a. *Can authors delineate the relationship between inflammation in Tet3 KO lung SMCs and weight loss?*

Response: We only detected a significant reduction of body weight in *Tet3* mutant mice 6 months after tamoxifen injection. There was a tendency for body weight loss 2 months after TAM injection, which was not significant. Inflammation was much more severe at 6 months compared to 2 months after TAM injection in *Tet3* mutant. Thus, the aggravated lung pathology and the increased inflammation is most likely responsible for the loss of body weight. We have replaced the original Extended data Fig.1f with these new results (revised Extended data Fig. 1j) and clarified the issue in the revised manuscript.

b. *If there is a relationship with enhanced TLR signaling, can the authors use a TLR inhibitor (i.e., Telratolimod, R-848, or E6446) in vivo to reduce spurious transcription?*

Response: We apologize that we did not present our findings in a better way. Activation of TLR signaling is a consequence of spurious transcription, not the other way around. There is no reason to assume the existence of an auto-regulatory loop that further increases spurious transcription after TLR activation. Nevertheless, the use of TLR inhibitors is an excellent suggestion to validate the importance of TLR7 signaling for the initiation of innate immune responses and inflammation. Since TLR7 inhibitor (E6446) treatment indiscriminately affects different cell types in the lung and is toxic for primary *Tet3* deficient SMCs, we treated mESCderived SMCs after *Tet3* KD with a TLR inhibitor. We observed reduced expression of cytokine/chemokine genes after TLR7 inhibitor treatment of *Tet3* knock-down SMCs (revised Extended Fig. 6h). Furthermore, transfection of RNA extracted from both *Tet3* KO and *Tet3* KD SMCs did not evoke expression of cytokine/chemokine genes in recipient cells when such cells were treated with the TLR7 inhibitor (revised Fig. 4d&g). These results clearly indicate that innate immune response triggered by the loss of 5hmC is TLR7-dependent.

Minor:

1. *The observation that spurious transcription is clearer on highly transcribed genes is expected (Fig 1F). Can authors combine the 4 panels into one panel and add boxplots comparing the signal of 5hmC at TSS, Gene body, and TES in WT and KO cells.*

Response: Thank you for the suggestion. We were also not completely happy about the presentation of the data. We have optimized the presentation as requested (revised Fig. 1f). Since the 5hmC signals in the bottom 25% quartile genes are very noisy and will interfere to visualize the 5hmC coverage profiles of the other 3 groups of genes, we present them separately.

2. *Provide a western blot to show KO of TET3? Please also probe for TET2 and TET1. Extended Fig. 1E shows some signal overlapping between SMA and TET3 in Tet3smko cells.*

Response: We have extensively tried to detect endogenous TET3 protein by western blot analysis but the available commercial antibodies are not of sufficient quality. We presented a RT-qPCR analysis of *Tet3* gene expression (Extended data Fig. 1h&o). We have now also included immunofluorescence staining of SMC layers in *Tet3* KO aorta and a snapshot view of RNA-seq from *Tet3*-deficient lung SMCs (Fig. 3 for reviewer). In addition, genomic PCRs using primers spanning the floxed exon 10 verified efficient excision (revised Extended Fig. 1g). These data clearly demonstrate inactivation of the *Tet3* gene. Furthermore, we respectfully would like to point out the strong loss of 5hmC in SMCs, which is also a clear indicator of *Tet3* gene inactivation.

As requested by the reviewer, we also performed WBs to monitor TET2 levels. WB analysis indicate efficient depletion of TET2 (revised Extended Fig. 2h), but unchanged TET2 levels in *Tet3* KO (revised Extended Fig. 2b). Tet1 remains undetectable in control, *Tet2* KO, and *Tet3* KO SMCs. The low level of

TET1 is consistent with published studies, reporting very low levels of *Tet1* in differentiated cells (Lio and Rao, 2019).

Some signal overlapping between SMA and TET3 in $Tet3^{smKO}$ cells might originate from other cells infiltrating into SMC layer, since the staining was performed using 8 μ m cryosection of lung tissues instead of cultured monolayer cells. These signals are rare and do not interfere with our conclusion.

Figure 3 for reviewer: **a**, Immunofluorescence staining for α -SMA and TET3 on cryosections from control and $Tet3^{smKO}$ lungs, 8 weeks after tamoxifen injection ($n=3$). DNA was stained by DAPI. Scale bar: 50 μ m. **b**, Integrated Genome Viewer (IGV) tracks displaying RNA-seq peaks in *Tet3* exon 10 ($n=2$).

3. Does loss of TET3 in SMCs cause increased cell growth/proliferation?!

Response: We did not observe an increase of SMC numbers or an increase of SMCs expressing the proliferation marker Ki67 in *Tet3*-mutant lungs (revised Extended Fig. 7g). Expression of cell cycle related genes in sorted *Tet3* KO SMC was not altered as well (Fig. 4 for reviewer). However, we detected a clear increase of senescent cells (SA-b-Gal staining positive) in the bronchial smooth muscle layer (revised Extended Fig. 7h). Moreover, expression of the senescence marker genes *p16* (*Cdkn1a*) and *p21* (*Cdkn1a*) were significantly augmented in mutant SMCs (revised Extended Fig. 7i). Cellular senescence and subsequent acquisition of a senescence-associated secretory phenotype (SASP) has been suggested to play an important role for specific subgroups of asthma patients e.g. adult-onset asthma (Wang et al., 2020). Since *Tet3*-deficient SMCs produce several cytokines characteristic for SASP, including IL1b, we reason that loss of *Tet3* in bronchial SMCs causes senescence instead of proliferation.

Figure 4 for reviewer: Expression of *Plk1*, *Ccne1*, *Ccnd1*, *Ccnb1*, *Cdk7*, *Cdk9*, *Ki67* genes by RNA-seq in sorted control and Tet3^{smKO:T} lung SMCs (n=2; two-tailed unpaired t-test).

4. Is there any effect on RNA polymerase II after loss of *Tet3* without induction of DRB?

Response: We performed western blot analysis to measure the levels of RNA polymerase II but did not detect obvious differences in RNA Pol II between control and *Tet3*-mutant SMCs (Fig. 5a for reviewer).

5. Fig 2E – TET3 OE seems to increase protein levels of elongating RNA polymerase. Do the authors observe any change in the protein levels of RNA polymerase components after loss of TET3 in SMCs?

Response: We have redone the experiment shown in (Fig. 2e), in which we expressed not only wild-type but also catalytic inactive TET3. Quantification of western blot from three biologically independent experiments revealed no change of Pol II pSer2 levels in mocktransfected cells or cells overexpressing catalytically active (Tet3^{WT}) or inactive TET3 (Tet3^{CD}) (revised Fig. 2e, Fig. 5b for reviewer). We also determined the protein levels of Pol II complex components such as TFIID, TAF4a by western blot analysis in wild-type and *Tet3* mutant SMCs, but did not detect obvious changes as well (Fig. 5a for reviewer).

Figure 5 for reviewer: **a**, Western blot analysis of TFIID, TAF4a, Pol II in sorted control and Tet3^{smKO:T} SMCs (n=3). Pan-actin was used as loading control. Quantifications of protein levels of TFIID, TAF4a, Pol II are shown in the right panel (n=3; two-tailed unpaired t-test). **b**, Western blot analysis of Pol II pSer2 in HEK293T cells with Mock, Tet3^{WT} and Tet3^{CD} (n=3). Pan-actin was used as loading control. Quantification of protein levels of Pol II pSer2 is shown in the right panel (n=3; one-way ANOVA with Tukey’s post hoc test).

6. Quantify PLA in 2F as dots/cell. It is unclear what % of PLA/nuclei means. Please clarify.

Response: We are sorry for the unclear information. We have quantified the number of PLA dots in each nucleus in the initial figure (Fig. 2f). The Y-axis now depicts the number of dots per cell (revised Fig. 2f).

7. Perform ChIP-seq of H3K36me3 and compare with gene expression, 5hmC, and spurious transcripts (pol2 chip and cage-seq). Could authors clarify why some genes (*Myh11* and *Cnn1*) have increase H3K36me3 at the promoter?

Response: We have performed H3K36me3 ChIP-seq experiments, demonstrating substantial reduction of H3K36me3 deposition in *Tet3*-deficient primary SMCs (revised Extended data Fig. 3h). Integrated analysis of H3K36me3 ChIP-seq, RNA-seq and Pol II pSer5 ChIP-seq revealed a dramatic drop of H3K36me3 levels in highly transcribed genes as well as a strong increase of intragenic Pol II entry after *Tet3* inactivation (revised Fig. 2h&i). Furthermore, gene tracks of H3K36me3 within *Acta2* and *Myh11* genes (revised Fig. 3g) indicate concomitant reduction of H3K36me3 signals and 5hmC deposition in the vicinity of intragenic CTSSs in contractile genes after *Tet3* inactivation.

In (Fig. 2h) (now shown as revised Extended data Fig. 3i), we only detected a decline of H3K36me3 deposition in gene bodies of *Acta2*, *Myh11* and *Cnn1*. We saw a slight increase of H3K36me3 at promoters of *Myh11* and *Cnn1*, but the changes are not significant (t-test p value >0.1).

8. The lack of reproducibility in Fig. 3e between the replicates is concerning. On line 185, authors state that 7761 genes have a log₂ ratio >1. Is this in both replicates?

Response: We reported that of all genes containing more than 4 exons, 7761 had a log₂ ratio >1 of all intermediate exons from second exon onwards versus the first exon in *Tet3*-deficient SMCs. This statement indeed refers to changes concomitantly observed in both replicates. Nevertheless, the reviewer is right that there is some variability among the replicates, which is a common issue when working with FACS-sorted primary smooth muscle cells from different individual mice. RNA-seq results obtained from cells cultured under the same conditions always yield more homogenous results when compared to primary cells derived from individual mice.

To alleviate the reviewer's concerns about reproducibility, we performed additional RT-qPCR experiments to monitor expression of intermediate exons of several spuriously (*Acta2*, *Myh11*, *Cnn1*) and non-spuriously expressed genes (*Vim*, *Tpm4*). These new data are shown in (revised Extended data Fig. 4j), validating the results from RNA-seq shown in (Fig. 3e). Please note that the RT-qPCR analysis indicates clear differences between WT and *Tet3*-deficient SMCs as well as between spuriously and non-spuriously expressed genes, despite some variability among individual mice.

9. Fig. 3a is a GO analysis of spurious transcripts, Fig. 4a is a KEGG analysis of RNA-seq data. Can the authors perform KEGG analysis on spurious transcripts since it has already been defined in Figure 3?

Response: Following the reviewer's request, we also performed KEGG analysis for spurious transcripts. KEGG analysis reveals that in addition to changes related to the actin cytoskeleton, several pathways involved in the sarcoplasmic reticulum function and SMC contraction such as 'Focal adhesion' (Ribeiro-Silva et al., 2021), 'calcium signaling pathway', and 'Inositol phosphate metabolism' are enriched (revised Fig. 5d). Deregulation of these pathways are in line with our finding that loss of *Tet3* impairs contractility of SMC.

10. PLA in 4C – please quantify spots/cell if possible and keep it consistent throughout the manuscript.

Response: We now consistently quantify data as introduced in Fig.2f (i.e. as % of PLA dots/cell) (revised Fig. 4c).

11. Rather than exogenously introducing RNA from Ctrl and KO SMCs in HELA and 293T cells, could the authors generate TLR7 KO (with CRISPR) or KD (with si or shRNA).

Response: This is certainly a good suggestion but probably not necessary, since good inhibitors for TLR7 are available. We treated ESC-derived SMC after *Tet3* knock-down with the TLR7 specific inhibitor (E6446). Inhibition of TLR7 in *Tet3* knock-down SMCs prevented expression of cytokine/chemokine genes, which was seen without TLR7 inhibition (revised Extended data Fig. 6h). Moreover, expression of cytokine/chemokine genes was blocked after transfection with RNA isolated from *Tet3* knock-down or knockout SMCs when the recipient HeLa cells were treated with the TLR7 inhibitor (revised Fig. 4d&g). These findings indicate that innate immune responses triggered by *Tet3* inactivation are TLR7-dependent.

12. Lines 254-257 – Can authors show/test this?

Response: We have compared the reduction of 5hmC levels in bronchial and pulmonary vascular SMCs after inactivation of *Tet3* KO and in newly generated *Tet2/Tet3* compound mutants. The additional loss of *Tet2* in *Tet3*-deficient SMCs further reduced 5hmC in vascular but not in bronchial SMCs (revised Extended Fig. 2m), indicating that *Tet2* partially compensates for the loss of in pulmonary VSMCs. As explained in detail above, we reason that BSMCs are more vulnerable to the loss of 5hmC than VSMCs, which results in different regulatory processes and reactions in these two different types of SMCs. Further evidence for this conclusion comes from the comparison of transcriptome profiles of

bronchioles and vascular cells. We identified only a minor overlap of DEGs (differentially regulated genes) in bronchioles and vasculatures after inactivation of *Tet3* (Extended data Fig. 7d).

We now also demonstrate that differences in the responsiveness of neighboring cells matter for the pathogenic events. We observed increased expression of interferon response related genes in lung SMCs and epithelial cells 8 weeks but not 4 weeks after TAM treatment (revised Extended Fig. 8a, b). Interestingly, we did not observe corresponding expression changes in endothelial cells (revised Extended Fig. 8b). Furthermore, we cultured epithelial cells (MLE12 cells) with conditioned medium from ESC-derived control or *Tet3* KD SMCs. Conditional medium from *Tet3* KD SMCs increased expression of pro-inflammatory genes such as *Il6*, *Il1b* and *Ifnb* and EMT related genes such as *Fn1*, *Cdh1*, *Vim* in MLE12 lung epithelial cells (revised extended date Fig. 8c). These findings indicate that airway epithelial cells are more responsive than vascular endothelial cells to paracrine effects of cytokines/chemokines produced in *Tet3* KO SMCs, resulting in airway remodeling.

13. No mention of Fig. 5f in the text.

Response: We are sorry for the mistake. We now mention (Fig. 5f) (now revised Fig. 6d) in revised manuscript.

14. Ex. Data Fig. 2 – Good piece of data. Please show western blot confirming ablation of TET2.

Response: We thank the reviewer for appreciating our experimental approach. As requested by the reviewer, we performed WB analysis for detection of TET2 protein. TET2 protein was not detectable in *Tet2* knockout SMCs (revised Extended Fig. 2h).

15. Ex. Data Fig. 4 –

a. Some of the significant qPCR plots (*Myh11* and *Cnn1*) seem to be driven by an outlier. Please clarify. Add replicate if possible.

Response: We have redone these RT-qPCR analyses of *Myh11* and *Cnn1* with newly FACSorted control and *Tet3* KO SMCs and replaced the original panel (Extended Fig. 4f) with a panel showing the new results (revised Extended data Fig. 5c).

b. (h) – both control replicates have different levels of proteins. *Vim* ctrl #2 is like *Tet3* ko #2

Not proteins, but full-length mRNA. A bit variation of vimentin, so adding more replicates and show the significance.

Response: We have repeated the analysis of *Vim* and updated the results (revised Extended data Fig. 4k).

16. Ex. Data Fig. 5 – Add TET2 to western blot in (b).

Response: We have included the results of TET2 WB analysis in *Tet3* KO SMCs (revised Extended data Fig. 2b). We think the data fit better into (revised Extended data Fig. 2b) than in (Extended data Fig. 5), allowing to demonstrate that TET2 expression is not increased as a consequence of the inactivation of *Tet3*.

17. Clarify in legend if the fold change is KO/Ctrl in (d) and include quantification (MFI) for (g) and (j).

Response: We assume the reviewer refers to results shown in (Extended data Fig. 6d, g&j). The fold-change in 'd' indeed refers to KO/Ctrl. We have included this information in legend of the revised figure legend. We also provide quantification (MFI) for panels 'g&j' (revised Extended data Fig. 8d&g).

References

- Cui, X.L., Nie, J., Ku, J., Dougherty, U., West-Szymanski, D.C., Collin, F., Ellison, C.K., Sieh, L., Ning, Y., Deng, Z., *et al.* (2020). A human tissue map of 5-hydroxymethylcytosines exhibits tissue specificity through gene and enhancer modulation. *Nat Commun* *11*, 6161.
- Fenouil, R., Cauchy, P., Koch, F., Descostes, N., Cabeza, J.Z., Innocenti, C., Ferrier, P., Spicuglia, S., Gut, M., Gut, I., *et al.* (2012). CpG islands and GC content dictate nucleosome depletion in a transcription-independent manner at mammalian promoters. *Genome Res* *22*, 2399-2408.
- Greco, C.M., Kunderfranco, P., Rubino, M., Larcher, V., Carullo, P., Anselmo, A., Kurz, K., Carell, T., Angius, A., Latronico, M.V., *et al.* (2016). DNA hydroxymethylation controls cardiomyocyte gene expression in development and hypertrophy. *Nat Commun* *7*, 12418.
- Hennig, B.P., and Fischer, T. (2013). The great repression: chromatin and cryptic transcription. *Transcription* *4*, 97-101.
- Kariko, K., Buckstein, M., Ni, H., and Weissman, D. (2005). Suppression of RNA recognition by Toll-like receptors: the impact of nucleoside modification and the evolutionary origin of RNA. *Immunity* *23*, 165-175.
- Kim, J.B., Zhao, Q., Nguyen, T., Pjanic, M., Cheng, P., Wirka, R., Travisano, S., Nagao, M., Kundu, R., and Quertermous, T. (2020). Environment-Sensing Aryl Hydrocarbon Receptor Inhibits the Chondrogenic Fate of Modulated Smooth Muscle Cells in Atherosclerotic Lesions. *Circulation* *142*, 575-590.
- Lauberth, S.M., Nakayama, T., Wu, X., Ferris, A.L., Tang, Z., Hughes, S.H., and Roeder, R.G. (2013). H3K4me3 interactions with TAF3 regulate preinitiation complex assembly and selective gene activation. *Cell* *152*, 1021-1036.
- Lio, C.J., and Rao, A. (2019). TET Enzymes and 5hmC in Adaptive and Innate Immune Systems. *Front Immunol* *10*, 210.
- Liu, M., Espinosa-Diez, C., Mahan, S., Du, M., Nguyen, A.T., Hahn, S., Chakraborty, R., Straub, A.C., Martin, K.A., Owens, G.K., *et al.* (2021). H3K4 di-methylation governs smooth muscle lineage identity and promotes vascular homeostasis by restraining plasticity. *Dev Cell* *56*, 2765-2782 e2710.
- Pastor, W.A., Pape, U.J., Huang, Y., Henderson, H.R., Lister, R., Ko, M., McLoughlin, E.M., Brudno, Y., Mahapatra, S., Kapranov, P., *et al.* (2011). Genome-wide mapping of 5hydroxymethylcytosine in embryonic stem cells. *Nature* *473*, 394-397.
- Ribeiro-Silva, J.C., Miyakawa, A.A., and Krieger, J.E. (2021). Focal adhesion signaling: vascular smooth muscle cell contractility beyond calcium mechanisms. *Clin Sci (Lond)* *135*, 1189-1207.

Shankman, L.S., Gomez, D., Cherepanova, O.A., Salmon, M., Alencar, G.F., Haskins, R.M., Swiatlowska, P., Newman, A.A., Greene, E.S., Straub, A.C., *et al.* (2015). KLF4-dependent phenotypic modulation of smooth muscle cells has a key role in atherosclerotic plaque pathogenesis. *Nat Med* *21*, 628-637.

Song, C.X., Szulwach, K.E., Fu, Y., Dai, Q., Yi, C., Li, X., Li, Y., Chen, C.H., Zhang, W., Jian, X., *et al.* (2011). Selective chemical labeling reveals the genome-wide distribution of 5-hydroxymethylcytosine. *Nat Biotechnol* *29*, 68-72.

Tan, L., Xiong, L., Xu, W., Wu, F., Huang, N., Xu, Y., Kong, L., Zheng, L., Schwartz, L., Shi, Y., *et al.* (2013). Genome-wide comparison of DNA hydroxymethylation in mouse embryonic stem cells and neural progenitor cells by a new comparative hMeDIP-seq method. *Nucleic Acids Res* *41*, e84.

Wang, Z.N., Su, R.N., Yang, B.Y., Yang, K.X., Yang, L.F., Yan, Y., and Chen, Z.G. (2020). Potential Role of Cellular Senescence in Asthma. *Front Cell Dev Biol* *8*, 59.

Wen, L., Li, X., Yan, L., Tan, Y., Li, R., Zhao, Y., Wang, Y., Xie, J., Zhang, Y., Song, C., *et al.* (2014). Whole-genome analysis of 5-hydroxymethylcytosine and 5-methylcytosine at base resolution in the human brain. *Genome Biol* *15*, R49.

Yu, M., Hon, G.C., Szulwach, K.E., Song, C.X., Zhang, L., Kim, A., Li, X., Dai, Q., Shen, Y., Park, B., *et al.* (2012). Base-resolution analysis of 5-hydroxymethylcytosine in the mammalian genome. *Cell* *149*, 1368-1380.

Decision Letter, Appeal:

Dear Thomas,

Thank you for your message asking us to reconsider our decision on your manuscript "Spurious transcription causing innate immune responses is prevented by 5hmC".

I have now discussed the points of your letter with my colleagues, and we think that your revision has addressed the major concerns in the previous round to our satisfaction and that it can be sent back to the original referees for further review. We therefore invite you to (re-)submit your manuscript for that.

When preparing a revision, please ensure that it fully complies with our editorial requirements for format and style; details can be found in the Guide to Authors on our website (<http://www.nature.com/ng/>).

Please be sure that your manuscript is accompanied by a separate letter detailing the changes you have made and your response to the points raised. At this stage we will need you to upload:

1) a copy of the manuscript in MS Word .docx format.

2) The Editorial Policy Checklist:

<https://www.nature.com/documents/nr-editorial-policy-checklist.pdf>

3) The Reporting Summary:

(Here you can read about the role of the Reporting Summary in reproducible science:

<https://www.nature.com/news/announcement-towards-greater-reproducibility-for-life-sciences-research-in-nature-1.22062>)

Please use the link below to be taken directly to the site and view and revise your manuscript:

[redacted]

With kind wishes,

Michael Fletcher, PhD
Senior Editor, Nature Genetics

ORCID: 0000-0003-1589-7087

Decision Letter, first revision:

Dear Thomas,

Thank you to you and your co-authors for your patience in this round of review - I hope that the weather in Bad Nauheim is as good as here in Berlin today!

Your Article, "Spurious transcription causing innate immune responses is prevented by 5hmC" has now been seen by the original 3 referees. You will see from their comments below that while they continue find your work of interest and all appreciate the improvements presented in this revision, there are still some important points outstanding. We remain interested in the possibility of publishing your study in Nature Genetics, but would like to consider your response to these concerns in the form of a revised manuscript before we make a final decision on publication.

In very brief, Reviewers #1 and #3 are now both satisfied and support publication. Reviewer #2, on the other hand, still has a few major criticisms. Most importantly, think that the hMeDIP-seq data is of poor quality and should be removed entirely in favour of the Nano-seal. Their other points are also important, e.g. the data supporting the Tet3 KO. In our reading of these comments, we do not think that this referee is asking for substantial further experimental work and so we hope that you will be able to address the concerns relatively easily.

To guide the scope of the revisions, the editors discuss the referee reports in detail within the team, including with the chief editor, with a view to identifying key priorities that should be addressed in revision and sometimes overruling referee requests that are deemed beyond the scope of the current study. We hope that you will find the prioritized set of referee points to be useful when revising your study. Please do not hesitate to get in touch if you would like to discuss these issues further.

We therefore invite you to revise your manuscript taking into account all reviewer and editor comments. Please highlight all changes in the manuscript text file. At this stage we will need you to upload a copy of the manuscript in MS Word .docx or similar editable format.

*2) If you have not done so already please begin to revise your manuscript so that it conforms to our Article format instructions, available [here](http://www.nature.com/ng/authors/article_types/index.html). Refer also to any guidelines provided in this letter.

[redacted]

We hope to receive your revised manuscript within four to eight weeks. If you cannot send it within this time, please let us know.

Sincerely,

Michael Fletcher, PhD
Senior Editor, Nature Genetics

ORCID: 0000-0003-1589-7087

Reviewers' Comments:

Reviewer #1:

Remarks to the Author:

The authors have been highly responsive to my comments and suggestions. The additional experiments solidify the innovation and importance of the work with respect to human disease.

Reviewer #2:

Remarks to the Author:

The authors provide a revised manuscript on their study of TET3 inactivation in smooth muscle cells with a focus on lung pathology. They addressed most of the reviewers' concerns and provided several additional data sets. The manuscript is clearly improved, and some issues of uncertainty have been resolved. However, there are still a few major concerns that need to be addressed.

1) The key point they are trying to make is that 5-hydroxymethylcytosine (5hmC) prevents spurious transcription in gene bodies (see Title). However, their current data on 5hmC are insufficient. After downloading the data and inspection in a genome browser, the genomic areas enriched in 5hmC by the hMeDIP method have the following characteristics: (i) The enriched (peak) areas rarely contain CpG sites, the targets of TET activity. (ii) There is poor overlap between replicates. (iii) Almost all enriched regions contain either TG repeats or regions with strong purine enrichment on one strand, such as GA repeats. (iv) Enriched areas are not visibly enhanced in gene bodies. This data shows that their 5hmC mapping data are incorrect and cannot be used to support the main conclusions of the paper (Figure 1f,g; Figure 2d; Figure 3d; Extended Data Figure 1; Extended Data Figure 5). However, they have now also performed Nano-seal, a pulldown technique based on biotinylation of 5hmC. This data looks much better. I do see CG sequences in the peaks, no repeats, and a nice enrichment in gene bodies and some surrounding regions. Also, the replicates line up well. In my opinion, all hMeDIP data needs to be removed and replaced with the Nano-seal data, and I don't understand why the authors would want to retain the poor-quality data from the initial experiments.

2) Figure 2:

I am not fully convinced that loss of TET3 leads to a reduction of H3K36me3 as claimed from Fig. 2i. This has now become a major point of the study. The differences are only very minor. Mechanistically, it is unclear. In these types of experiments, it is difficult to be sure that the observed differences in metagene profiles are not due to small experimental variations or normalization problems. If it's not completely clear, this point doesn't really need to be made.

3) Figure 3:

Gene track panels are not fully convincing. The data for 5hmC should be shown as replicates. Why not

show a gene that has denser coverage with 5hmC, such as Myh9?

4) Figure 6:

I think the link to asthma is not clear and seems a bit premature. The image data for 5hmC in the tissue section is unclear. The data refers to $n=4$ or $n=5$, but the "n" is not defined. Are these single cells? If yes, the numbers are clearly too small. The same question applies to Extended Data Figure 9.

5) Figure 1 for reviewer and Ext. Data Figure 1:

TET3 knockout confirmation: The authors provide an RNA-seq snapshot over exon 10 (Figure 1 for reviewer). However, in the knockout with exon 10 deletion, there is no reduced signal relative to neighboring exons.

Extended Data Figure 1, panel i for TET3 is so much less convincing than Extended Data Figure 2, panel e for TET2.

Extended Data Figure 3:

There is a large discrepancy in H3K36me3 signal by Western blot (panel h) and by ChIP-seq (panel g). Most of the H3K36me3 signal should indeed be in gene bodies.

Reviewer #3:

Remarks to the Author:

I am satisfied with the efforts of the authors to address all major concerns.

The data in the revised manuscript is sufficient and does a fine job of tying in many different concepts.

Author Rebuttal, first revision:

Responses to reviewers' comments

Reviewer #1:

Remarks to the Author:

The authors have been highly responsive to my comments and suggestions. The additional experiments solidify the innovation and importance of the work with respect to human disease.

Response: We thank the reviewer for the helpful suggestions, which helped us to improve the manuscript.

Reviewer #2:

Remarks to the Author:

The authors provide a revised manuscript on their study of TET3 inactivation in smooth muscle cells with a focus on lung pathology. They addressed most of the reviewers' concerns and provided several additional data sets. The manuscript is clearly improved, and some issues of uncertainty have been resolved. However, there are still a few major concerns that need to be addressed.

1) *The key point they are trying to make is that 5-hydroxymethylcytosine (5hmC) prevents spurious transcription in gene bodies (see Title). However, their current data on 5hmC are insufficient. After downloading the data and inspection in a genome browser, the genomic areas enriched in 5hmC by the hMeDIP method have the following characteristics: (i) The enriched (peak) areas rarely contain CpG sites, the targets of TET activity. (ii) There is poor overlap between replicates. (iii) Almost all enriched regions contain either TG repeats or regions with strong purine enrichment on one strand, such as GA repeats. (iv) Enriched areas are not visibly enhanced in gene bodies. This data shows that their 5hmC mapping data are incorrect and cannot be used to support the main conclusions of the paper (Figure 1f,g; Figure 2d; Figure 3d; Extended Data Figure 1; Extended Data Figure 5). However, they have now also performed Nano-seal, a pulldown technique based on biotinylation of 5hmC. This data looks much better. I do see CG sequences in the peaks, no repeats, and a nice enrichment in gene bodies and some surrounding regions. Also, the replicates line up well. In my opinion, all hMeDIP data needs to be removed and replaced with the Nano-seal data, and I don't understand why the authors would want to retain the poor-quality data from the initial experiments.*

Response: We appreciate the reviewer's view about the value of the hMeDIP-seq data. We do not want to hide that we have a somewhat different view about the usefulness of the data, which might be due to different analytical pipelines used. Nevertheless, we agree that the quality of Nano-seal data is superior compared to the hMeDIP results. Therefore, we followed the reviewer's advice and replaced the hMeDIP data (Figure 1f, g; Figure 2d; Extended Data Figure 3a&e, Extended Data Figure 5a) by the new Nano-seal-seq results (revised Figure 1f, g; Figure 2d; Extended Data Figure 3a&c, Extended Data Figure 5b). The main conclusions of the study are fully supported by the Nano-seal results.

The reviewer requested to remove hMeDIP-seq data from (Figure 3d and Extended Data Figure 1), which is a bit confusing, since neither figure contained hMeDIP-seq results. We assume that the reviewer meant (Extended Data Figure 3) and (Extended Data Figure 5), which were both altered by showing Nano-seal instead of hMeDIP-seq data.

2) *Figure 2: I am not fully convinced that loss of TET3 leads to a reduction of H3K36me3 as claimed from Fig. 2i. This has now become a major point of the study. The differences are only very minor. Mechanistically, it is unclear. In these types of experiments, it is difficult to be sure that the observed*

differences in metagene profiles are not due to small experimental variations or normalization problems. If it's not completely clear, this point doesn't really need to be made.

Response: We are also puzzled by the discrepancies between the western blot analysis, which indicates a strong overall reduction of H3K36me3 after Tet3 inactivation in smooth muscle cells, and the ChIP-seq analysis, which indicates a moderate H3K36me3 reduction over the gene bodies. To further validate the western blot findings, we increased the sample numbers for the western blot analysis to (n=5) and carefully quantified the results. The revised (Extended data Fig. 3e) demonstrates a clear reduction of H3K36me3 to approximately 35% of wild-type levels in Tet3-null smooth muscle cells.

We assume that the -in comparison- moderate reduction of H3K36me3 at gene bodies, either assessing the whole genome (Extended data Fig. 3f) or distinct subgroups of genes as defined in Fig. 2b (Fig. 2i), is caused by normalization issues, which partially mask potential differences. Theoretically, it is also possible that the strong reduction of H3K36me3, which is evident by western blot analysis, is caused by reduction of H3K36me3 outside of gene bodies. However, the common assumption is that gene bodies constitute the major source of H3K36me3, which was also emphasized by the reviewer, who wrote “Most of the H3K36me3 signal should indeed be in gene bodies.” Therefore, we assume that limitations in the H3K36me3 ChIP-seq analysis prevented us from detecting more dramatic differences. Similar observations were made in other studies, reporting a relatively modest reduction of H3K36me3 on gene bodies when assessed by ChIP-seq, despite clear evidence for reduction of H3K36me3 by other methods (e.g., see (Leonards et al, 2020)). The paper by (Leonards et al, 2020) reported reduced formation of H3K36me3 based on western blot analysis (Suppl. Fig. 6b), but the reduction of H3K36me3 monitored by ChIP-seq (Fig. 8d) was minor. In comparison, the reduction of H3K36me3 indicated by ChIP-seq in our study was much more pronounced than in the paper by (Leonards et al, 2020).

To further confirm the reduction of H3K36me3 after inactivation of *Tet3*, we performed additional H3K36me3 ChIP-qPCR experiments of several selected genes, which confirmed the reduction of H3K36me3 over gene bodies (revised Extended data Fig. 3g).

Mechanistically, the depletion of H3K36me3 over gene bodies corresponds well to other experimental findings described in the manuscript. We found that TET3 forms a complex with Pol II and SETD2 and that *Tet3* overexpression enhances the interaction between RNA Pol II and SETD2 (the H3K36me3 methyltransferase) (Fig. 2e&g), leading to increased H3K36me3 deposition (revised Extended data Fig. 6g). Importantly, we also demonstrated that overexpression of catalytic dead TET3 does not enhance the interaction between RNA Pol II and SETD2 (Fig. 2g) and therefore does not restore the reduced H3K36me3 levels in *Tet3*depleted SMCs. These results suggest that TET3-dependent 5hmC formation, but not a noncatalytic function of TET3, is responsible for promoting the interaction between SETD2 and RNA Pol II. We reason that these findings provide a convincing mechanistic explanation for the reduction of H3K36me3 in *Tet3* mutants.

3) Figure 3:

Gene track panels are not fully convincing. The data for 5hmC should be shown as replicates. Why not show a gene that has denser coverage with 5hmC, such as *Myh9*?

Response: Due to space constraints, we only show the sum of the reads from control and mutant groups and not the replicates. We provide a figure for the reviewer presenting the different individual tracks of Nano-seq (Figure 1 for the reviewer). We do not fully understand the request for MYH9. MYH9 is not a marker for contractile smooth muscle cells and *Myh9* is not part of the group of 515 genes showing spurious transcription (defined in Fig. 3a).

Figure 1 for the reviewer: Integrated Genome Viewer (IGV) tracks displaying the first single nucleotide of CAGE-seq capture sequences (CAGE-tag>8) and RNA-seq peaks, Nano-seq, and H3K36me3 ChIP-seq signals in *Acta2* and *Myh11* genes (n=2). Bottom, schematic representation of putative transcription

factor binding sites and gene tracks views of Nano-seal, H3K36me3 ChIP-seq signals within genomic regions containing putative transcription factor binding sites for (n=2).

4) *Figure 6: I think the link to asthma is not clear and seems a bit premature. The image data for 5hmC in the tissue section is unclear. The data refers to n=4 or n=5, but the “n” is not defined. Are these single cells? If yes, the numbers are clearly too small. The same question applies to Extended Data Figure 9.*

Response: We politely disagree with the reviewer’s view, which might have been caused by insufficient description of our experimental approach. We demonstrated a reduction of 5hmC in two different mouse models of asthma AND in a cohort of human asthma patients. The “nnumbers” refer to the number of human asthma patients and mice. In each individual, hundreds of cells were analyzed and counted. In (Fig.6e&g), one dot represents the MFI (mean fluorescence intensity) of 5hmC in nuclei of 100 randomly selected α -SMA⁺ cells per human or mouse lung section. To clarify our experimental approach, we expanded the method part, which now reads: “5hmC signals were determined by quantifying the mean fluorescence intensity (MFI) of 100 randomly selected α -SMA⁺ cells per lung tissue section of individual mouse and human subjects. Images were analyzed by Image J, followed by calculation of average MFI values per nucleus. Individual dots reflect the abundance of 5hmC signals in individual mouse and human lungs. N-numbers refer to the number of individual mouse and human subjects.”

Furthermore, we observed pronounced hyperplasia of mucus-producing cells, Th2-cell based immune responses, enhanced fibrosis in *Tet3*-deficient lungs and deregulated expression of EMT-related genes in epithelial cells cultured with conditional medium from *Tet3* KD cells, all hallmarks for asthma (revised Fig.6a, c&d, revised Extended data Fig. 8c&f). In sum, we can conclude that our results provide convincing evidence that the reduction of 5-hmC in airways of human asthma patients is not an epiphenomenon but causally involved in the pathogenesis of asthma.

5) *Figure 1 for reviewer and Ext. Data Figure 1: TET3 knockout confirmation: The authors provide an RNA-seq snapshot over exon 10 (Figure 1 for reviewer). However, in the knockout with exon 10 deletion, there is no reduced signal relative to neighboring exons. Extended Data Figure 1, panel i for TET3 is so much less convincing than Extended Data Figure 2, panel e for TET2.*

Response: We appreciate the reviewer’s concerns. The reviewer is right that the number of RNA reads of the deleted exon 10 is similar to the number of reads of the neighboring *Tet3* exons. It is a common phenomenon that deletion of floxed exon leads to rapid decay of the mutated mRNA, which subsequently reduces RNA-seq signals for all exons. In fact, we observed a reduction of RNA-seq reads for all *Tet3* exons (Figure 2 for the reviewer), indicating that deletion of exon 10 of the *Tet3* gene indeed results in RNA decay. We also would like to point out that the PCR analysis of genomic PCR from *Tet3* mutant cells clearly demonstrates efficient deletion of exon 10 (Extended Data Figure 1o).

Figure 2 for reviewer: Integrated Genome Viewer (IGV) tracks displaying RNA-seq peaks in *Tet3* gene (n=2).

To unequivocally prove depletion of TET3 protein after deletion of exon 10 of the *Tet3* gene, we tested a newly purchased TET3 antibody from Active Motif, which works nicely. The new antibody revealed a strong reduction of TET protein by western blot analysis (Extended Data Figure 1p). Furthermore, we now demonstrate reduction of TET3 protein in mutant SMCs both on lung sections and in isolated lung SMCs. These results are now shown in (revised Extended data Fig. 1h & p).

Extended Data Figure 3: There is a large discrepancy in H3K36me3 signal by Western blot (panel h) and by ChIP-seq (panel g). Most of the H3K36me3 signal should indeed be in gene bodies.

Response: The comment addresses the same issue raised in comment #1. As pointed out above, we further validated the western blot findings of H3K36me3 reduction by increasing the sample numbers for western blot analysis (n=5) and carefully quantified the results. The revised (Extended data Fig. 3e) demonstrates a clear reduction of H3K36me3 to approximately 35% of wild-type levels in *Tet3*-null smooth muscle cells. We assume that the -in comparison- moderate reduction of H3K36me3 at gene bodies, either assessing the whole genome (Extended data Fig. 3f) or distinct subgroups of genes as defined in Fig. 2b (Fig. 2i), was caused by normalization issues, which mask potential differences. Therefore, we performed additional H3K36me3 ChIP-qPCR experiments for several selected genes, which confirmed the reduction of H3K36me3 over gene bodies (revised Extended data Fig. 3g). The reduction of H3K36me3 detected by western blot and the reduced formation of H3K36me3 in gene bodies uncovered by ChIP-seq after inactivation of *Tet3* are consistent with our conclusion that TET3-mediated formation of 5hmC plays an important role for the recruitment of SETD1 to target genes. The ChIP-seq analysis of lung smooth muscle cells revealed a more pronounced reduction of H3K36me3 after *Tet3* inactivation at the 3'-end of gene bodies compared to the rest of gene bodies, which is consistent with previous studies showing an enrichment of H3K36me3 towards the 3'-end of gene bodies (Liu et al., 2019).

References:

Leonards, K., Almosaillekh, M., Tauchmann, S., Bagger, F.O., Thirant, C., Juge, S., Bock, T., Mereau, H., Bezerra, M.F., Tzankov, A., et al. (2020). Nuclear interacting SET domain protein 1 inactivation impairs GATA1-regulated erythroid differentiation and causes erythroleukemia. *Nat Commun* 11, 2807.

Liu B, Liu Y, Wang B, Luo Q, Shi J, Gan J, Shen WH, Yu Y, Dong A. The transcription factor OsSUF4 interacts with SDG725 in promoting H3K36me3 establishment. *Nat Commun*. 2019 Jul 5;10(1):2999. doi: 10.1038/s41467-019-10850-5.

Reviewer #3:

Remarks to the Author:

I am satisfied with the efforts of the authors to address all major concerns.

The data in the revised manuscript is sufficient and does a fine job of tying in many different concepts.

Response: We are delighted that we were able to address all concerns of the reviewer. Thank you for the careful evaluation.

Decision Letter, second revision:

Our ref: NG-A57348R2

22nd Aug 2022

Dear Thomas,

Thank you for submitting your revised manuscript "Spurious transcription causing innate immune responses is prevented by 5hmC" (NG-A57348R2). It has now been seen by the original Referee #2 and their comments are below. The reviewer is satisfied with your revisions, and therefore we'll be happy in principle to publish it in *Nature Genetics*, pending minor revisions to satisfy the referees' final requests and to comply with our editorial and formatting guidelines.

Sincerely,

Michael Fletcher, PhD
Senior Editor, Nature Genetics

ORCID: 0000-0003-1589-7087

Reviewer #2 (Remarks to the Author):

The authors have addressed my remaining questions. The study provides important information on the functional role of 5-hydroxymethylcytosine in gene bodies.

Final Decision Letter:

Dear Thomas,

I am delighted to say that your manuscript "Spurious transcription causing innate immune responses is prevented by 5-hydroxymethylcytosine" has been accepted for publication in an upcoming issue of Nature Genetics.

Your paper will be published online after we receive your corrections and will appear in print in the next available issue. You can find out your date of online publication by contacting the Nature Press Office (press@nature.com) after sending your e-proof corrections. Now is the time to inform your

Public Relations or Press Office about your paper, as they might be interested in promoting its publication. This will allow them time to prepare an accurate and satisfactory press release. Include your manuscript tracking number (NG-A57348R3) and the name of the journal, which they will need when they contact our Press Office.

Please note that *Nature Genetics* is a Transformative Journal (TJ). Authors may publish their research with us through the traditional subscription access route or make their paper immediately open access through payment of an article-processing charge (APC). Authors will not be required to make a final decision about access to their article until it has been accepted. [Find out more about Transformative Journals](https://www.springernature.com/gp/open-research/transformative-journals)

Authors may need to take specific actions to achieve [compliance](https://www.springernature.com/gp/open-research/funding/policy-compliance-faqs) with funder and institutional open access mandates. If your research is supported by a funder that requires immediate open access (e.g. according to [Plan S principles](https://www.springernature.com/gp/open-research/plan-s-compliance)) then you should select the gold OA route, and we will direct you to the compliant route where possible. For authors selecting the subscription publication route, the journal's standard licensing terms will need to be accepted, including [self-archiving-and-license-to-publish](https://www.nature.com/nature-portfolio/editorial-policies/self-archiving-and-license-to-publish). Those licensing terms will supersede any other terms that the author or any third party may assert apply to any version of the manuscript.

Please note that Nature Portfolio offers an immediate open access option only for papers that were first submitted after 1 January, 2021.

If you have not already done so, we invite you to upload the step-by-step protocols used in this manuscript to the Protocols Exchange, part of our on-line web resource, natureprotocols.com. If you complete the upload by the time you receive your manuscript proofs, we can insert links in your article that lead directly to the protocol details. Your protocol will be made freely available upon publication of your paper. By participating in natureprotocols.com, you are enabling researchers to more readily reproduce or adapt the methodology you use. [Natureprotocols.com](http://natureprotocols.com) is fully searchable, providing your protocols and paper with increased utility and visibility. Please submit your protocol to <https://protocolexchange.researchsquare.com/>. After entering your nature.com username and password you will need to enter your manuscript number (NG-A57348R3). Further information can be found at <https://www.nature.com/nature-portfolio/editorial-policies/reporting-standards#protocols>

Sincerely,

Michael Fletcher, PhD
Senior Editor, Nature Genetics

ORCID: 0000-0003-1589-7087

Click here if you would like to recommend Nature Genetics to your librarian
<http://www.nature.com/subscriptions/recommend.html#forms>

** Visit the Springer Nature Editorial and Publishing website at http://editorial-jobs.springernature.com?utm_source=ejp_NGen_email&utm_medium=ejp_NGen_email&utm_campaign=ejp_NGen for more information about our career opportunities. If you have any questions please click [here](mailto:editorial.publishing.jobs@springernature.com).**